# Limitations of Lazy Training of
# Two-layers Neural Networks

**Behrooz Ghorbani**
Department of Electrical Engineering
Stanford University
ghorbani@stanford.edu

**Song Mei**
ICME
Stanford University
songmei@stanford.edu

**Theodor Misiakiewicz**
Department of Statistics
Stanford University
misiakie@stanford.edu

**Andrea Montanari**
Department of Electrical Engineering
and Department of Statistics
Stanford University
montanar@stanford.edu

## Abstract

We study the supervised learning problem under either of the following two models:

(1) Feature vectors $\boldsymbol{x}_i$ are $d$-dimensional Gaussians and responses are $y_i = f_*(\boldsymbol{x}_i)$ for $f_*$ an unknown quadratic function;

(2) Feature vectors $\boldsymbol{x}_i$ are distributed as a mixture of two $d$-dimensional centered Gaussians, and $y_i$'s are the corresponding class labels.

We use two-layers neural networks with quadratic activations, and compare three different learning regimes: the random features (RF) regime in which we only train the second-layer weights; the neural tangent (NT) regime in which we train a linearization of the neural network around its initialization; the fully trained neural network (NN) regime in which we train all the weights in the network. We prove that, even for the simple quadratic model of point (1), there is a potentially unbounded gap between the prediction risk achieved in these three training regimes, when the number of neurons is smaller than the ambient dimension. When the number of neurons is larger than the number of dimensions, the problem is significantly easier and both NT and NN learning achieve zero risk.

## 1   Introduction

Consider the supervised learning problem in which we are given i.i.d. data $\{(\boldsymbol{x}_i, y_i)\}_{i \le n}$, where $\boldsymbol{x}_i \sim \mathbb{P}$ a probability distribution over $\mathbb{R}^d$, and $y_i = f_*(\boldsymbol{x}_i)$. [1] We would like to learn the unknown function $f_*$ as to minimize the prediction risk $\mathbb{E}\{(f(\boldsymbol{x}) - f_*(\boldsymbol{x}))^2\}$. We will assume throughout $f_* \in L^2(\mathbb{R}^d, \mathbb{P})$, i.e. $\mathbb{E}\{f_*(\boldsymbol{x})^2\} < \infty$.

The function class of two-layers neural networks (with $N$ neurons) is defined by:

$$\mathcal{F}_{\mathsf{NN},N} = \Big\{ f(\boldsymbol{x}) = c + \sum_{i=1}^{N} a_i \sigma(\langle \boldsymbol{w}_i, \boldsymbol{x} \rangle) : c, a_i \in \mathbb{R}, \boldsymbol{w}_i \in \mathbb{R}^d, i \in [N] \Big\}. \tag{1}$$

Classical universal approximation results [9] imply that any $f_* \in L^2(\mathbb{R}^d, \mathbb{P})$ can be approximated arbitrarily well by an element in $\mathcal{F}_{\mathsf{NN}} = \cup_N \mathcal{F}_{\mathsf{NN},N}$ (under mild conditions). At the same time,

we know that such an approximation can be constructed in polynomial time only for a subset of functions $f_*$. Namely, there exist sets of functions $f_*$ for which no algorithm can construct a good approximation in $\mathcal{F}_{\mathsf{NN},N}$ in polynomial time [19, 24], even having access to the full distribution $\mathbb{P}$ (under certain complexity-theoretic assumptions).

These facts lead to the following central question in neural network theory:

> *For which subset of function $\mathcal{F}_{\mathrm{tract}} \subseteq L^2(\mathbb{R}^d, \mathbb{P})$ can a neural network approximation be learnt efficiently?*

Here 'efficiently' can be formalized in multiple ways: in this paper we will focus on learning via stochastic gradient descent.

Significant amount of work has been devoted to two subclasses of $\mathcal{F}_{\mathsf{NN},N}$ which we will refer to as the random feature model (RF) [22], and the neural tangent model (NT) [18]:

$$\mathcal{F}_{\mathsf{RF},N}(\boldsymbol{W}) = \left\{ f_N(\boldsymbol{x}) = \sum_{i=1}^{N} a_i \sigma(\langle \boldsymbol{w}_i, \boldsymbol{x} \rangle) : a_i \in \mathbb{R}, i \in [N] \right\}, \tag{2}$$

$$\mathcal{F}_{\mathsf{NT},N}(\boldsymbol{W}) = \left\{ f_N(\boldsymbol{x}) = c + \sum_{i=1}^{N} \sigma'(\langle \boldsymbol{w}_i, \boldsymbol{x} \rangle)\langle \boldsymbol{a}_i, \boldsymbol{x} \rangle : c \in \mathbb{R}, \boldsymbol{a}_i \in \mathbb{R}^d, i \in [N] \right\}. \tag{3}$$

Here $\boldsymbol{W} = (\boldsymbol{w}_1, \ldots, \boldsymbol{w}_N) \in \mathbb{R}^{d \times N}$ are weights which are not optimized and instead drawn at random. Through this paper, we will assume $(\boldsymbol{w}_i)_{i \leq N} \sim_{iid} \mathsf{N}(\boldsymbol{0}, \boldsymbol{\Gamma})$. [2]

We can think of RF and NT as *tractable inner bounds* of the class of neural networks NN:

- *Tractable.* Both $\mathcal{F}_{\mathsf{RF},N}(\boldsymbol{W})$, $\mathcal{F}_{\mathsf{NT},N}(\boldsymbol{W})$ are finite-dimensional linear spaces, and minimizing the empirical risk over these classes can be performed efficiently.

- *Inner bounds.* Indeed $\mathcal{F}_{\mathsf{RF},N}(\boldsymbol{W}) \subseteq \mathcal{F}_{\mathsf{NN},N}$: the random feature model is simply obtained by fixing all the first layer weights. Further $\mathcal{F}_{\mathsf{NT}}(\boldsymbol{W}) \subseteq \mathrm{cl}(\mathcal{F}_{\mathsf{NN},2N})$ (the closure of the class of neural networks with $2N$ neurons). This follows from $\varepsilon^{-1}[\sigma(\langle \boldsymbol{w}_i + \varepsilon \boldsymbol{a}_i, \boldsymbol{x} \rangle) - \sigma(\langle \boldsymbol{w}_i, \boldsymbol{x} \rangle)] = \langle \boldsymbol{a}_i, \boldsymbol{x} \rangle \sigma'(\langle \boldsymbol{w}_i, \boldsymbol{x} \rangle) + o(1)$ as $\varepsilon \to 0$.

It is possible to show that the class of neural networks NN is significantly more expressive than the two linearization RF, NT, see e.g. [26, 15]. In particular, [15] shows that, if the feature vectors $\boldsymbol{x}_i$ are uniformly random over the $d$-dimensional sphere, and $N, d$ are large with $N = O(d)$, then $\mathcal{F}_{\mathsf{RF},N}(\boldsymbol{W})$ can only capture linear functions, while $\mathcal{F}_{\mathsf{NT},N}(\boldsymbol{W})$ can only capture quadratic functions.

Despite these findings, it could still be that the subset of functions $\mathcal{F}_{\mathrm{tract}} \subseteq L^2(\mathbb{R}^d, \mathbb{P})$ for which we can learn efficiently a neural network approximation is well described by RF and NT. Indeed, several recent papers show that –in a certain highly overparametrized regime– this description is accurate [12, 11, 20]. A specific counterexample is given in [26]: if the function to be learnt is a single neuron $f_*(\boldsymbol{x}) = \sigma(\langle \boldsymbol{w}_*, \boldsymbol{x} \rangle)$ then gradient descent (in the space of neural networks with $N = 1$ neurons) efficiently learns it [21]; on the other hand, RF or NT require a number of neurons exponential in the dimension to achieve vanishing risk.

## 1.1 Summary of Main Results

In this paper we explore systematically the gap between RF, NT and NN, by considering two specific data distributions:

(qf) Quadratic functions: feature vectors are distributed according to $\boldsymbol{x}_i \sim \mathsf{N}(\boldsymbol{0}, \mathbf{I}_d)$ and responses are quadratic functions $y_i = f_*(\boldsymbol{x}_i) \equiv b_0 + \langle \boldsymbol{x}_i, \boldsymbol{B}\boldsymbol{x}_i \rangle$ with $\boldsymbol{B} \succeq 0$.

(mg) Mixture of Gaussians: $y_i = \pm 1$ with equal probability $1/2$, and $\boldsymbol{x}_i | y_i = +1 \sim \mathsf{N}(0, \boldsymbol{\Sigma}^{(1)})$, $\boldsymbol{x}_i | y_i = -1 \sim \mathsf{N}(0, \boldsymbol{\Sigma}^{(2)})$.

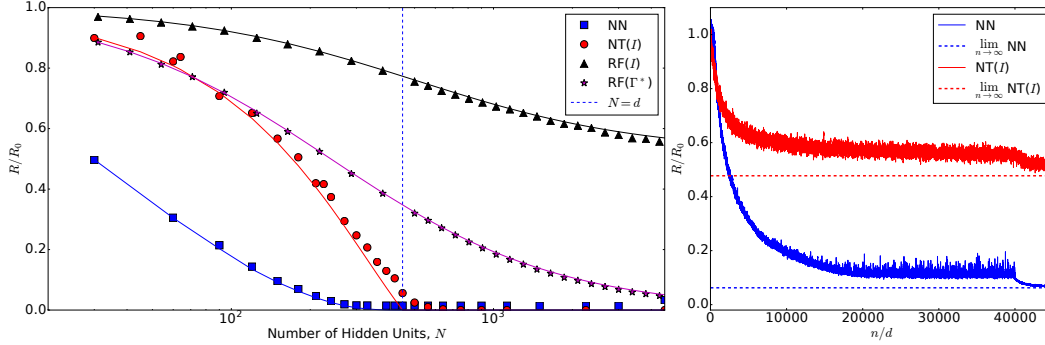

Figure 1: Left frame: Prediction (test) error of a two-layer neural networks in fitting a quadratic function in $d = 450$ dimensions, as a function of the number of neurons $N$. We consider the large sample (population) limit $n \to \infty$ and compare three training regimes: random features (RF), neural tangent (NT), and fully trained neural networks (NN). Lines are analytical predictions obtained in this paper, and dots are empirical results. Right frame: Evolution of the risk for NT and NN with the number of samples. Dashed lines are our analytic prediction for the large $n$ limit.

Let us emphasize that the choice of quadratic functions in model qf is not arbitrary: in a sense, it is the *most favorable case for* NT *training*. Indeed [15] proves that[3] (when $N = O(d)$): $(i)$ Third- and higher-order polynomials cannot be approximated nontrivially by $\mathcal{F}_{\mathsf{NT},N}(\boldsymbol{W})$; $(ii)$ Linear functions are already well approximated within $\mathcal{F}_{\mathsf{RF},N}(\boldsymbol{W})$.

For clarity, we will first summarize our result for the model qf, and then discuss generalizations to mg. The prediction risk achieved within any of the regimes RF, NT, NN is defined by

$$R_{\mathsf{M},N}(f_*) = \min_{\hat{f} \in \mathcal{F}_{\mathsf{M},N}(\boldsymbol{W})} \mathbb{E}\big\{(f_*(\boldsymbol{x}) - \hat{f}(\boldsymbol{x}))^2\big\}, \qquad \mathsf{M} \in \{\mathsf{RF}, \mathsf{NT}, \mathsf{NN}\}. \tag{4}$$

$$R_{\mathsf{NN},N}(f_*; \ell, \varepsilon) = \mathbb{E}\big\{(f_*(\boldsymbol{x}) - \hat{f}_{\mathsf{SGD}}(\boldsymbol{x}; \ell, \varepsilon))^2\big\}, \tag{5}$$

where $\hat{f}_{\mathsf{SGD}}(\,\cdot\,; \ell, \varepsilon)$ is the neural network produced by $\ell$ steps of stochastic gradient descent (SGD) where each sample is used once, and the stepsize is set to $\varepsilon$ (see Section 2.3 for a complete definition). Notice that the quantities $R_{\mathsf{M},N}(f_*)$, $R_{\mathsf{NN},N}(f_*; \ell, \varepsilon)$ are random variables because of the random weights $\boldsymbol{W}$, and the additional randomness in SGD.

Our results are summarized by Figure 1, which compares the risk achieved by the three approaches above in the population limit $n \to \infty$, using quadratic activations $\sigma(u) = u^2 + c_0$. We consider the large-network, high-dimensional regime $N, d \to \infty$, with $N/d \to \rho \in (0, \infty)$. Figure 1 reports the risk achieved by various approaches in numerical simulations, and compares them with our theoretical predictions for each of three regimes RF, NT, and NN, which are detailed in the next sections.

The agreement between analytical predictions and simulations is excellent but, more importantly, a clear picture emerges. We can highlight a few phenomena that are illustrated in this figure:

*Random features* do not capture quadratic functions. The random features risk $R_{\mathsf{RF},N}(f_*)$ remains generally bounded away from zero for all values of $\rho = N/d$. It is further highly dependent on the distribution of the weight vectors $\boldsymbol{w}_i \sim \mathsf{N}(\boldsymbol{0}, \boldsymbol{\Gamma})$. Section 2.1 characterizes explicitly this dependence, for general activation functions $\sigma$. For large $\rho = N/d$, the optimal distribution of the weight vectors uses covariance $\boldsymbol{\Gamma}^* \propto \boldsymbol{B}$, but even in this case the risk is bounded away from zero unless $\rho \to \infty$.

*The neural tangent model* achieves vanishing risk on quadratic functions for $N > d$. However, the risk is bounded away from zero if $N/d \to \rho \in (0, 1)$. Section 2.1 provides explicit expressions for the minimum risk as a function of $\rho$. Roughly speaking NT fits the quadratic function $f_*$ along random subspace determined by the random weight vectors $\boldsymbol{w}_i$. For $N \geq d$, these vectors span the

whole space $\mathbb{R}^d$ and hence the limiting risk vanishes. For $N < d$ only a fraction of the space is spanned, and not the most important one (i.e. not the principal eigendirections of $\boldsymbol{B}$).

*Fully trained neural networks* achieve vanishing risk on quadratic functions for $N > d$: this is to be expected on the basis of the previous point. For $N/d \to \rho \in (0, 1)$ the risk is generally bounded away from 0, but its value is smaller than for the neural tangent model. Namely, in Section 2.3 we give an explicit expression for the asymptotic risk (holding for $\boldsymbol{B} \succeq \boldsymbol{0}$) implying that, for some $\mathrm{GAP}(\rho) > 0$ (independent of $N, d$),

$$\lim_{t \to \infty} \lim_{\varepsilon \to 0} R_{\mathsf{NN},N}(f_*; \ell = t/\varepsilon, \varepsilon) = \inf_{f \in \mathcal{F}_{\mathsf{NN},N}} \mathbb{E}\{(f(\boldsymbol{x}) - f_*(\boldsymbol{x}))^2\} \leq R_{\mathsf{NT},N}(f_*) - \mathrm{GAP}(\rho). \quad (6)$$

We prove this result by showing convergence of SGD to gradient flow in the population risk, and then proving a strict saddle property for the population risk. As a consequence the limiting risk on the left-hand side coincides with the minimum risk over the whole space of neural networks $\inf_{f \in \mathcal{F}_{\mathsf{NN},N}} \mathbb{E}\{(f(\boldsymbol{x}) - f_*(\boldsymbol{x}))^2\}$. We characterize the latter and shows that it amounts to fitting $f_*$ along the $N$ principal eigendirections of $\boldsymbol{B}$. This mechanism is very different from the one arising in the NT regime.

The picture emerging from these findings is remarkably simple. The fully trained network learns the most important eigendirections of the quadratic function $f_*(\boldsymbol{x})$ and fits them, hence surpassing the NT model which is confined to a random set of directions.

Let us emphasize that the above separation between NT and NN is established only for $N \leq d$. It is natural to wonder whether this separation generalizes to $N > d$ for more complicated classes of functions, or if instead it always vanishes for wide networks. We expect the separation to generalize to $N > d$ by considering higher order polynomial, instead of quadratic functions. Partial evidence in this direction is provided by [15]: for third- or higher-order polynomials NT does not achieve vanishing risk at any $\rho \in (0, \infty)$. The mechanism unveiled by our analysis of quadratic functions is potentially more general: neural networks are superior to linearized models such as RF or NT, because they can learn a good representation of the data.

Our results for quadratic functions are formally presented in Section 2. In order to confirm that the picture we obtain is general, we establish similar results for mixture of Gaussians in Section 3. More precisely, our results of RF and NT for mixture of Gaussians are very similar to the quadratic case. In this model, however, we do not prove a convergence result for NN analogous to (6), although we believe it should be possible by the same approach outlined above. On the other hand, we characterize the minimum prediction risk over neural networks $\inf_{f \in \mathcal{F}_{\mathsf{NN},N}} \mathbb{E}\{(y - f(\boldsymbol{x}))^2\}$ and prove it is strictly smaller than the minimum achieved by RF and NT. Finally, Section 4 contains background on our numerical experiments.

## 1.2 Further Related Work

The connection (and differences) between two-layers neural networks and random features models has been the object of several papers since the original work of Rahimi and Recht [22]. An incomplete list of references includes [5, 2, 6, 7, 23]. Our analysis contributes to this line of work by establishing a sharp asymptotic characterization, although in more specific data distributions. Sharp results have recently been proven in [15], for the special case of random weights $\boldsymbol{w}_i$ uniformly distributed over a $d$-dimensional sphere. Here we consider the more general case of anisotropic random features with covariance $\boldsymbol{\Gamma} \not\propto \boldsymbol{I}$. This clarifies a key reason for suboptimality of random features: the data representation is not adapted to the target function $f_*$. We focus on the population limit $n \to \infty$. Complementary results characterizing the variance as a function of $n$ are given in [17].

The NT model (3) is much more recent [18]. Several papers show that SGD optimization within the original neural network is well approximated by optimization within the model NT as long as the number of neurons is large compared to a polynomial in the sample size $N \gg n^{c_0}$ [12, 11, 3, 28]. Empirical evidence in the same direction was presented in [20, 4].

Chizat and Bach [8] clarified that any nonlinear statistical model can be approximated by a linear one in an early (*lazy*) training regime. The basic argument is quite simple. Given a model $\boldsymbol{x} \mapsto f(\boldsymbol{x}; \boldsymbol{\theta})$ with parameters $\boldsymbol{\theta}$, we can Taylor-expand around a random initialization $\boldsymbol{\theta}_0$. Setting $\boldsymbol{\theta} = \boldsymbol{\theta}_0 + \boldsymbol{\beta}$, we get

$$f(\boldsymbol{x}; \boldsymbol{\theta}) \approx f(\boldsymbol{x}; \boldsymbol{\theta}_0) + \boldsymbol{\beta}^\mathsf{T} \nabla_{\boldsymbol{\theta}} f(\boldsymbol{x}; \boldsymbol{\theta}_0) \approx \boldsymbol{\beta}^\mathsf{T} \nabla_{\boldsymbol{\theta}} f(\boldsymbol{x}; \boldsymbol{\theta}_0). \quad (7)$$

Here the second approximation holds since, for many random initializations, $f(\boldsymbol{x}; \boldsymbol{\theta}_0) \approx 0$ because of random cancellations. The resulting model $\boldsymbol{\beta}^{\mathsf{T}} \nabla_{\boldsymbol{\theta}} f(\boldsymbol{x}; \boldsymbol{\theta}_0)$ is linear, with random features.

Our objective is complementary to this literature: we prove that RF and NT have limited approximation power, and significant gain can be achieved by full training.

Finally, our analysis of fully trained networks connects to the ample literature on non-convex statistical estimation. For two layers neural networks with quadratic activations, Soltanolkotabi, Javanmard and Lee [25] showed that, as long as the number of neurons satisfies $N \geq 2d$ there are no spurious local minimizers. Du and Lee [10] showed that the same holds as long as $N \geq d \wedge \sqrt{2n}$ where $n$ is the sample size. Zhong et. al. [27] established local convexity properties around global optima. Further related landscape results include [14, 16, 13].

## 2 Main Results: Quadratic Functions

As mentioned in the previous section, our results for quadratic functions (qf) assume $\boldsymbol{x}_i \sim \mathsf{N}(\mathbf{0}, \mathbf{I}_d)$ and $y_i = f_*(\boldsymbol{x}_i)$ where

$$f_*(\boldsymbol{x}) \equiv b_0 + \langle \boldsymbol{x}, \boldsymbol{B} \boldsymbol{x} \rangle . \tag{8}$$

### 2.1 Random Features

We consider random feature model with first-layer weights $(\boldsymbol{w}_i)_{i \leq N} \sim \mathsf{N}(\mathbf{0}, \boldsymbol{\Gamma})$. We make the following assumptions:

**A1.** The activation function $\sigma$ verifies $\sigma(u)^2 \leq c_0 \exp(c_1 u^2/2)$ for some constants $c_0, c_1$ with $c_1 < 1$. Further it is nonlinear (i.e. there is no $a_0, a_1 \in \mathbb{R}$ such that $\sigma(u) = a_0 + a_1 u$ almost everywhere).

**A2.** We fix the weights' normalization by requiring $\mathbb{E}\{\|\boldsymbol{w}_i\|_2^2\} = \mathrm{Tr}(\boldsymbol{\Gamma}) = 1$. We assume the operator norm $\|d \cdot \boldsymbol{\Gamma}\|_{\mathrm{op}} \leq C$ for some constant $C$, and that the empirical spectral distribution of $d \cdot \boldsymbol{\Gamma}$ converges weakly, as $d \to \infty$ to a probability distribution $\mathcal{D}$ over $\mathbb{R}_{\geq 0}$.

**Theorem 1.** *Let $f_*$ be a quadratic function as per Eq. (8), with $\mathbb{E}(f_*) = 0$. Assume conditions **A1** and **A2** to hold. Denote by $\lambda_k = \mathbb{E}_{G \sim \mathsf{N}(0,1)}[\sigma(G) \mathrm{He}_k(G)]$ the $k$-th Hermite coefficient of $\sigma$ and assume $\lambda_0 = 0$. Define $\tilde{\lambda} = \mathbb{E}_{G \sim \mathsf{N}(0,1)}[\sigma(G)^2] - \lambda_1^2$. Let $\psi > 0$ be the unique solution of*

$$-\tilde{\lambda} = -\frac{\rho}{\psi} + \int \frac{\lambda_1^2 t}{1 + \lambda_1^2 t \psi} \mathcal{D}(\mathrm{d}t) . \tag{9}$$

*Then, the following holds as $N, d \to \infty$ with $N/d \to \rho$:*

$$R_{\mathsf{RF},N}(f_*) = \|f_*\|_{L_2}^2 \left( 1 - \frac{\psi \lambda_2^2 d \langle \boldsymbol{\Gamma}, \boldsymbol{B} \rangle^2}{\|\boldsymbol{B}\|_F^2 \left( 2 + \psi \lambda_2^2 d \|\boldsymbol{\Gamma}\|_F^2 \right)} + o_{d,\mathbb{P}}(1) \right) . \tag{10}$$

*Moreover, assuming $\langle \boldsymbol{\Gamma}, \boldsymbol{B} \rangle^2 / \|\boldsymbol{\Gamma}\|_F^2 \|\boldsymbol{B}\|_F^2$ to have a limit as $d \to \infty$, (10) simplifies as follows for $\rho \to \infty$:*

$$\lim_{\rho \to \infty} \lim_{d \to \infty, N/d \to \rho} \frac{R_{\mathsf{RF},N}(f_*)}{\|f_*\|_{L_2}^2} = \lim_{d \to \infty} \left( 1 - \frac{\langle \boldsymbol{\Gamma}, \boldsymbol{B} \rangle^2}{\|\boldsymbol{\Gamma}\|_F^2 \|\boldsymbol{B}\|_F^2} \right) . \tag{11}$$

Notice that $R_{\mathsf{RF},N}(f_*)/\|f_*\|_{L_2}^2$ is the RF risk normalized by the risk of the trivial predictor $f(\boldsymbol{x}) = 0$. The asymptotic result in (11) is remarkably simple. By Cauchy-Schwartz, the normalized risk is bounded away from zero even as the number of neurons per dimension diverges $\rho = N/d \to \infty$, unless $\boldsymbol{\Gamma} \propto \boldsymbol{B}$, i.e. the random features are perfectly aligned with the function to be learned. For isotropic random features, the right-hand side of Eq. (11) reduces to $1 - \mathrm{Tr}(\boldsymbol{B})^2/(d\|\boldsymbol{B}\|_F^2)$. In particular, RF performs very poorly when $\mathrm{Tr}(\boldsymbol{B}) \ll \sqrt{d}\|\boldsymbol{B}\|_F$, and no better than the trivial predictor $f(\boldsymbol{x}) = 0$ if $\mathrm{Tr}(\boldsymbol{B}) = 0$.

Notice that the above result applies to quite general activation functions. The formulas simplify significantly for quadratic activations.

**Corollary 1.** *Under the assumptions of Theorem 1, further assume $\sigma(x) = x^2 - 1$. Then we have, as $N, d \to \infty$ with $N/d \to \rho$:*

$$R_{\mathsf{RF},N}(f_*) = \|f_*\|_{L_2}^2 \left(1 - \frac{\rho d \langle \boldsymbol{B}, \boldsymbol{\Gamma} \rangle^2}{\|\boldsymbol{B}\|_F^2 \left(1 + \rho d \|\boldsymbol{\Gamma}\|_F^2\right)} + o_{d,\mathbb{P}}(1)\right). \tag{12}$$

The right-hand side of Eq. (12) is plotted in Fig. 1 for isotropic features $\boldsymbol{\Gamma} = \mathbf{I}/d$, and for optimal features $\boldsymbol{\Gamma} = \boldsymbol{\Gamma}^* \propto \boldsymbol{B}$.

## 2.2 Neural Tangent

For the NT regime, we focus on quadratic activations and isotropic weights $\boldsymbol{w}_i \sim \mathsf{N}(\mathbf{0}, \mathbf{I}_d/d)$.

**Theorem 2.** *Let $f_*$ be a quadratic function as per Eq. (8), with $\mathbb{E}(f_*) = 0$, and assume $\sigma(x) = x^2$. Then, we have for $N, d \to \infty$ with $N/d \to \rho$*

$$\mathbb{E}[R_{\mathsf{NT},N}(f_*)] = \|f_*\|_{L^2}^2 \left\{(1-\rho)_+^2 \left(1 - \frac{\mathrm{Tr}(\boldsymbol{B})^2}{d\|\boldsymbol{B}\|_F^2}\right) + (1-\rho)_+ \frac{\mathrm{Tr}(\boldsymbol{B})^2}{d\|\boldsymbol{B}\|_F^2} + o_d(1)\right\}.$$

*where the expectation is taken over $\boldsymbol{w}_i \sim_{i.i.d} \mathsf{N}(\mathbf{0}, \mathbf{I}_d/d)$.*

As for the case of random features, the NT risk depends on the target function $f_*(\boldsymbol{x})$ only through the ratio $\mathrm{Tr}(\boldsymbol{B})^2/(d\|\boldsymbol{B}\|_F^2)$. However, the normalized risk is always smaller than the baseline $R_{\mathsf{NT},N}(f_*) = \|f_*\|_{L^2}^2$. Note that, by Cauchy-Schwartz, $\mathbb{E}[R_{\mathsf{NT},N}(f_*)] \leq (1-\rho)_+\|f_*\|_{L^2}^2 + o_d(1)$, with this worst case achieved when $\boldsymbol{B} \propto \mathbf{I}$. In particular, $\mathbb{E}[R_{\mathsf{NT},N}(f_*)]$ vanishes asymptotically for $\rho \geq 1$. This comes at the price of a larger number of parameters to be fitted, namely $Nd$ instead of $N$.

## 2.3 Neural Network

For the analysis of SGD-trained neural networks, we assume $f_*$ to be a quadratic function as per Eq. (8), but we will now restrict to the positive semidefinite case $\boldsymbol{B} \succeq 0$. We consider quadratic activations $\sigma(x) = x^2$, and we fix the second layers weights to be 1:

$$\hat{f}(\boldsymbol{x}; \boldsymbol{W}, c) = \sum_{i=1}^{N} \langle \boldsymbol{w}_i, \boldsymbol{x} \rangle^2 + c.$$

Notice that we use an explicit offset to account for the mismatch in means between $f_*$ and $\hat{f}$. It is useful to introduce the population risk, as a function of the network parameters $\boldsymbol{W}, c$:

$$L(\boldsymbol{W}, c) = \mathbb{E}[(f_*(\boldsymbol{x}) - \hat{f}(\boldsymbol{x}; \boldsymbol{W}, c))^2] = \mathbb{E}\left[\left(\langle \boldsymbol{x}\boldsymbol{x}^\mathsf{T}, \boldsymbol{B} - \boldsymbol{W}\boldsymbol{W}^\mathsf{T}\rangle + b_0 - c\right)^2\right].$$

Here expectation is with respect to $\boldsymbol{x} \sim \mathsf{N}(\mathbf{0}, \mathbf{I}_d)$. We will study a one-pass version of SGD, whereby at each iteration $k$ we perform a stochastic gradient step with respect to a fresh sample $(\boldsymbol{x}_k, f_*(\boldsymbol{x}_k))$

$$(\boldsymbol{W}_{k+1}, c_{k+1}) = (\boldsymbol{W}_k, c_k) - \varepsilon \nabla_{\boldsymbol{W},c}\left(f_*(\boldsymbol{x}_k) - \hat{f}(\boldsymbol{x}_k; \boldsymbol{W}, c)\right)^2,$$

and define

$$R_{\mathsf{NN},N}(f_*; \ell, \varepsilon) \equiv L(\boldsymbol{W}_\ell, c_\ell) = \mathbb{E}_{\boldsymbol{x} \sim \mathsf{N}(\mathbf{0}, \mathbf{I}_d)}[(f_*(\boldsymbol{x}) - \hat{f}(\boldsymbol{x}; \boldsymbol{W}_\ell, c_\ell))^2].$$

Notice that this is the risk with respect to a new sample, independent from the ones used to train $\boldsymbol{W}_\ell, c_\ell$. It is the test error. Also notice that $\ell$ is the number of SGD steps but also (because of the one-pass assumption) the sample size. Our next theorem characterizes the asymptotic risk achieved by SGD. This prediction is reported in Figure 1.

**Theorem 3.** *Let $f_*$ be a quadratic function as per Eq. (8), with $\boldsymbol{B} \succeq 0$. Consider SGD with initialization $(\boldsymbol{W}_0, c_0)$ whose distribution is absolutely continuous with respect to the Lebesgue measure. Let $R_{\mathsf{NN},N}(f_*; \ell, \varepsilon)$ be the test prediction error after $\ell$ SGD steps with step size $\varepsilon$.*

*Then we have (probability is over the initialization $(\boldsymbol{W}_0, c_0)$ and the samples)*

$$\lim_{t \to \infty} \lim_{\varepsilon \to 0} \mathbb{P}\left(\left|R_{\mathsf{NN},N}(f_*; \ell = t/\varepsilon, \varepsilon) - \inf_{\boldsymbol{W},c} L(\boldsymbol{W}, c)\right| \geq \delta\right) = 0, \quad \inf_{\boldsymbol{W},c} L(\boldsymbol{W}, c) = 2\sum_{i=N+1}^{d} \lambda_i(\boldsymbol{B})^2,$$

*where $\lambda_1(\boldsymbol{B}) \geq \lambda_2(\boldsymbol{B}) \geq \cdots \geq \lambda_d(\boldsymbol{B})$ are the ordered eigenvalues of $\boldsymbol{B}$.*

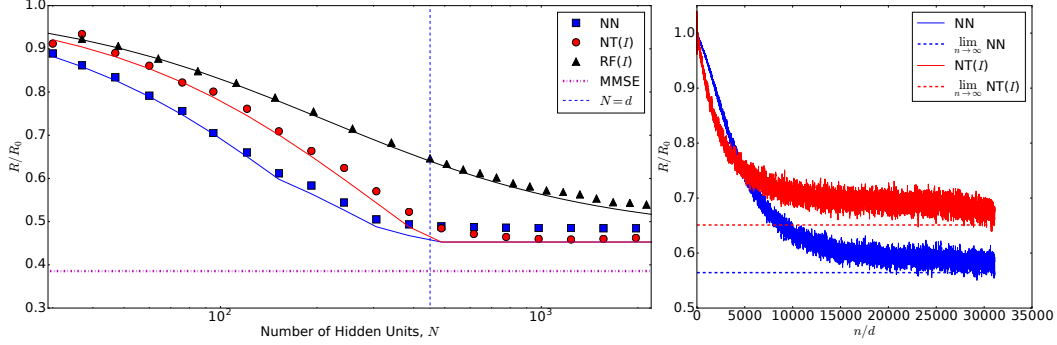

Figure 2: Left frame: Prediction (test) error of a two-layer neural networks in fitting a mixture of Gaussians in $d = 450$ dimensions, as a function of the number of neurons $N$, within the three regimes RF, NT, NN. Lines are analytical predictions obtained in this paper, and dots are empirical results (both in the population limit). Dotted line is the Bayes error. Right frame: Evolution of the risk for NT and NN with the number of samples.

The proof of this theorem depends on the following proposition concerning the landscape of the population risk, which is of independent interest.

**Proposition 1.** *Let $f_*$ be a quadratic function as per Eq. (8), with $\boldsymbol{B} \succeq 0$. For any sub-level set of the risk function $\Omega(B_0) = \{\boldsymbol{x} = (\boldsymbol{W}, c) : L(\boldsymbol{W}, c) \leq B_0\}$, there exists constants $\varepsilon, \delta > 0$ such that $L$ is $(\varepsilon, \delta)$-strict saddle in the region $\Omega(B_0)$. Namely, for any $\boldsymbol{x} \in \Omega(B_0)$ with $\|\nabla L(\boldsymbol{x})\|_2 \leq \varepsilon$, we have $\lambda_{\min}(\nabla^2 L(\boldsymbol{x})) < -\delta$.*

We can now compare the risk achieved within the regimes RF, NT and NN. Gathering the results of Corollary 1, and Theorems 2, 3 (using $\boldsymbol{w}_i \sim \mathsf{N}(0, \mathbf{I}/d)$ for RF and NT), we obtain

$$\frac{R_{\mathsf{M},N}(f_*)}{\|f_*\|_{L_2}^2} \approx \begin{cases} 1 - \dfrac{\rho}{1+\rho}\dfrac{\mathrm{Tr}(\boldsymbol{B})^2}{d\|\boldsymbol{B}\|_F^2} & \text{for } \mathsf{M} = \mathsf{RF}, \\[2ex] (1-\rho)_+^2 + \rho(1-\rho)_+ \dfrac{\mathrm{Tr}(\boldsymbol{B})^2}{d\|\boldsymbol{B}\|_F^2} & \text{for } \mathsf{M} = \mathsf{NT}, \\[2ex] 1 - \dfrac{\sum_{i=1}^{d\wedge N} \lambda_i(\boldsymbol{B})^2}{\|\boldsymbol{B}\|_F^2} & \text{for } \mathsf{M} = \mathsf{NN}. \end{cases} \tag{13}$$

As anticipated, NN learns the most important directions in $f_*$, while RF, NT do not.

## 3 Main Results: Mixture of Gaussians

In this section, we consider the mixture of Gaussian setting (mg): $y_i = \pm 1$ with equal probability $1/2$, and $\boldsymbol{x}_i | y_i = +1 \sim \mathsf{N}(0, \boldsymbol{\Sigma}^{(1)})$, $\boldsymbol{x}_i | y_i = -1 \sim \mathsf{N}(0, \boldsymbol{\Sigma}^{(2)})$. We parametrize the covariances as $\boldsymbol{\Sigma}^{(1)} = \boldsymbol{\Sigma} - \boldsymbol{\Delta}$ and $\boldsymbol{\Sigma}^{(2)} = \boldsymbol{\Sigma} + \boldsymbol{\Delta}$, and will make the following assumptions:

**M1.** There exists constants $0 < c_1 < c_2$ such that $c_1 \mathbf{I}_d \preceq \boldsymbol{\Sigma} \preceq c_2 \mathbf{I}_d$;

**M2.** $\|\boldsymbol{\Delta}\|_{\mathrm{op}} = \Theta_d(1/\sqrt{d})$.

The scaling in assumption **M2** ensures the signal-to-noise ratio to be of order one. If the eigenvalues of $\boldsymbol{\Delta}$ are much larger than $1/\sqrt{d}$, then it is easy to distinguish the two classes with high probability (they are asymptotically mutually singular). If $\|\boldsymbol{\Delta}\|_{\mathrm{op}} = o_d(1/\sqrt{d})$ then no non-trivial classifier exists.

We will denote by $\mathbb{P}_{\boldsymbol{\Sigma}, \boldsymbol{\Delta}}$ the joint distribution of $(y, \boldsymbol{x})$ under the (mg) model, and by $\mathbb{E}_{\boldsymbol{\Sigma}, \boldsymbol{\Delta}}$ or $\mathbb{E}_{(y, \boldsymbol{x})}$ the corresponding expectation. The minimum prediction risk within any of the regimes RF, NT, NN is defined by

$$R_{\mathsf{M},N}(\mathbb{P}) = \inf_{f \in \mathcal{F}_{\mathsf{M},N}} \mathbb{E}_{(y, \boldsymbol{x})}\{(y - f(\boldsymbol{x}))^2\}, \qquad \mathsf{M} \in \{\mathsf{RF}, \mathsf{NT}, \mathsf{NN}\}.$$

As mentioned in the introduction, the picture emerging from our analysis of the mg model is aligned with the results obtained in the previous section. We will limit ourselves to stating the results without repeating comments that were made above. Our results are compared with simulations in Figure 2. Notice that, in this case, the Bayes error (MMSE) is not achieved even for very wide networks $N/d \gg 1$ either by NT or NN.

## 3.1 Random Features

As in the previous section, we generate random first-layer weights $(\boldsymbol{w}_i)_{i \leq N} \sim \mathsf{N}(\boldsymbol{0}, \boldsymbol{\Gamma})$. We consider a general activation function satisfying condition **A1**. We make the following assumption on $\boldsymbol{\Gamma}, \boldsymbol{\Sigma}$:

**B2.** We fix the weights' normalization by requiring $\mathbb{E}\{\langle \boldsymbol{w}_i, \boldsymbol{\Sigma}\boldsymbol{w}_i \rangle\} = \mathrm{Tr}(\boldsymbol{\Gamma}\boldsymbol{\Sigma}) = 1$. We assume that there exists a constant $C$ such that $\|d \cdot \boldsymbol{\Gamma}\|_{\mathrm{op}} \leq C$, and that the empirical spectral distribution of $d \cdot (\boldsymbol{\Gamma}^{1/2}\boldsymbol{\Sigma}\boldsymbol{\Gamma}^{1/2})$ converges weakly, as $d \to \infty$ to a probability distribution $\mathcal{D}$ over $\mathbb{R}_{\geq 0}$.

**Theorem 4.** *Consider the* mg *distribution, with* $\boldsymbol{\Sigma}$ *and* $\boldsymbol{\Delta}$ *satisfying condition* **M1** *and* **M2**. *Assume conditions* **A1** *and* **B2** *to hold. Define* $\lambda_k = \mathbb{E}_{G \sim \mathsf{N}(0,1)}[\sigma(G)\mathrm{He}_k(G)]$ *to be the* $k$-*th Hermite coefficient of* $\sigma$ *and assume without loss of generality* $\lambda_0 = 0$. *Define* $\tilde{\lambda} = \mathbb{E}[\sigma(G)^2] - \lambda_1^2$. *Let* $\psi > 0$ *be the unique solution of*

$$-\tilde{\lambda} = -\frac{\rho}{\psi} + \int \frac{\lambda_1^2 t}{1 + \lambda_1^2 t \psi} \, \mathcal{D}(\mathrm{d}t) \,. \tag{14}$$

*Define* $\zeta_1(d) \equiv d \,\mathrm{Tr}(\boldsymbol{\Sigma}\boldsymbol{\Gamma}\boldsymbol{\Sigma}\boldsymbol{\Gamma})/2$, $\zeta_2(d) \equiv d \,\mathrm{Tr}(\boldsymbol{\Delta}\boldsymbol{\Gamma})^2/4$. *Then, the following holds as* $N, d \to \infty$ *with* $N/d \to \rho$:

$$R_{\mathsf{RF},N}(\mathbb{P}_{\boldsymbol{\Sigma},\boldsymbol{\Delta}}) = \frac{1 + \zeta_1(d)\lambda_2^2\psi}{1 + (\zeta_1(d) + \zeta_2(d))\lambda_2^2\psi} + o_{d,\mathbb{P}}(1), \,. \tag{15}$$

*Moreover, assume* $\zeta_1(d) \, \zeta_2(d)$ *to have limits as* $d \to \infty$, *i.e. we have* $\lim_{d \to \infty} \zeta_j(d) = \zeta_{j,*}$ *for* $j = 1, 2$. *Then the following holds as* $\rho \to \infty$:

$$\lim_{\rho \to \infty} \lim_{d \to \infty, N/d \to \rho} R_{\mathsf{RF},N}(\mathbb{P}_{\boldsymbol{\Sigma},\boldsymbol{\Delta}}) = \frac{\zeta_{1,*}}{\zeta_{1,*} + \zeta_{2,*}}. \tag{16}$$

## 3.2 Neural Tangent

For the NT model, we first state our theorem for general $\boldsymbol{\Sigma}$ and $\boldsymbol{w}_i \sim \mathsf{N}(\boldsymbol{0}, \boldsymbol{\Gamma})$ and then give an explicit concentration result in the case $\boldsymbol{\Sigma} = \mathbf{I}$ and isotropic weights $\boldsymbol{w}_i \sim \mathsf{N}(\boldsymbol{0}, \mathbf{I}/d)$.

**Theorem 5.** *Let* $\mathbb{P}_{\boldsymbol{\Sigma},\boldsymbol{\Delta}}$ *be the mixture of Gaussian distribution, with* $\boldsymbol{\Sigma}$ *and* $\boldsymbol{\Delta}$ *satisfying conditions* **M1** *and* **M2**. *Further assume* $\sigma(x) = x^2$. *Then, the following holds for almost every* $\boldsymbol{W} \in \mathbb{R}^{d \times N}$ *(with respect to the Lebesgue measure):*

$$R_{\mathsf{NT},N}(\mathbb{P}_{\boldsymbol{\Sigma},\boldsymbol{\Delta}}) = \frac{2}{2 + \|\tilde{\boldsymbol{\Delta}}\|_F^2 - \|\boldsymbol{P}_\perp \tilde{\boldsymbol{\Delta}} \boldsymbol{P}_\perp\|_F^2} + o_d(1),$$

*where* $\tilde{\boldsymbol{\Delta}} = \boldsymbol{\Sigma}^{-1/2}\boldsymbol{\Delta}\boldsymbol{\Sigma}^{-1/2}$ *and* $\boldsymbol{P}_\perp = \mathbf{I} - \boldsymbol{\Sigma}^{1/2}\boldsymbol{W}(\boldsymbol{W}^\mathsf{T}\boldsymbol{\Sigma}\boldsymbol{W})^{-1}\boldsymbol{W}^\mathsf{T}\boldsymbol{\Sigma}^{1/2}$ *is the projection perpendicular to* $\mathrm{span}(\boldsymbol{\Sigma}^{1/2}\boldsymbol{W})$.

*Assuming further that* $\boldsymbol{\Sigma} = \mathbf{I}$ *and* $\boldsymbol{w}_i \sim_{i.i.d.} \mathsf{N}(\boldsymbol{0}, \mathbf{I}_d/d)$, *we have as* $N, d \to \infty$ *with* $N/d \to \rho$:

$$R_{\mathsf{NT},N}(\mathbb{P}_{\mathbf{I},\boldsymbol{\Delta}}) = \frac{2}{2 + \kappa(\rho, \boldsymbol{\Delta}) \|\boldsymbol{\Delta}\|_F^2} + o_{d,\mathbb{P}}(1),$$

$$\kappa(\rho, \boldsymbol{\Delta}) = 1 - (1 - \rho)_+^2 \left(1 - \frac{\mathrm{Tr}(\boldsymbol{\Delta})^2}{d\|\boldsymbol{\Delta}\|_F^2}\right) - (1 - \rho)_+ \frac{\mathrm{Tr}(\boldsymbol{\Delta})^2}{d\|\boldsymbol{\Delta}\|_F^2},$$

*In particular, for* $\rho \geq 1$, *we have (for almost every* $\boldsymbol{W}$)

$$R_{\mathsf{NT},N}(\mathbb{P}_{\mathbf{I},\boldsymbol{\Delta}}) = \frac{1}{1 + \|\boldsymbol{\Delta}\|_F^2/2} + o_{d,\mathbb{P}}(1).$$

## 3.3 Neural Network

We consider quadratic activations with general offset and coefficients $\hat{f}(\boldsymbol{x}; \boldsymbol{W}, \boldsymbol{a}, c) = \sum_{i=1}^{N} a_i \langle \boldsymbol{w}_i, \boldsymbol{x} \rangle^2 + c$. This is optimized over $(a_i, \boldsymbol{w}_i)_{i \leq N}$ and $c$.

**Theorem 6.** *Let $\mathbb{P}_{\boldsymbol{\Sigma}, \boldsymbol{\Delta}}$ be the mixture of Gaussian distribution, with $\boldsymbol{\Sigma}$ and $\boldsymbol{\Delta}$ satisfying conditions* **M1** *and* **M2**. *Then, the following holds*

$$R_{\mathsf{NN},N}(\mathbb{P}_{\boldsymbol{\Sigma}, \boldsymbol{\Delta}}) = \frac{2}{2 + \sum_{i=1}^{N \wedge d} \lambda_i(\tilde{\boldsymbol{\Delta}})^2} + o_d(1),$$

*where $\tilde{\boldsymbol{\Delta}} = \boldsymbol{\Sigma}^{-1/2} \boldsymbol{\Delta} \boldsymbol{\Sigma}^{-1/2}$ and $\lambda_1(\tilde{\boldsymbol{\Delta}}) \geq \lambda_1(\tilde{\boldsymbol{\Delta}}) \geq \cdots \geq \lambda_d(\tilde{\boldsymbol{\Delta}})$ are the singular values of $\tilde{\boldsymbol{\Delta}}$. In particular, for $\rho \geq 1$, we have*

$$R_{\mathsf{NN},N}(\mathbb{P}_{\mathbf{I}, \boldsymbol{\Delta}}) = \frac{1}{1 + \|\tilde{\boldsymbol{\Delta}}\|_F^2 / 2} + o_d(1).$$

Let us emphasize that, for this setting, we do not have a convergence result for SGD as for the model qf, cf. Theorem 3. However, because of certain analogies between the two models, we expect a similar result to hold for mixtures of Gaussians.

We can now compare the risks achieved within the regimes RF, NT and NN. Gathering the results of Theorems 4, 5 and 6 for $\boldsymbol{\Sigma} = \mathbf{I}$ and $\sigma(x) = x^2 - 1$ (using $\boldsymbol{w}_i \sim \mathsf{N}(\mathbf{0}, \mathbf{I}/d)$ for RF and NT), we obtain

$$R_{\mathsf{M},N}(\mathbb{P}_{\mathbf{I}, \boldsymbol{\Delta}}) \approx \begin{cases} \dfrac{1}{1 + \frac{\rho}{1+2\rho} \cdot \frac{\mathrm{tr}(\boldsymbol{\Delta})^2}{2d}} & \text{for } \mathsf{M} = \mathsf{RF}, \\[2ex] \dfrac{1}{1 + \kappa(\rho, \boldsymbol{\Delta}) \|\boldsymbol{\Delta}\|_F^2 / 2} & \text{for } \mathsf{M} = \mathsf{NT}, \\[2ex] \dfrac{1}{1 + \sum_{i=1}^{N \wedge d} \lambda_i(\boldsymbol{\Delta})^2 / 2} & \text{for } \mathsf{M} = \mathsf{NN}. \end{cases} \tag{17}$$

We recover a similar behavior as in the case of the (qf) model: NN learns the most important directions of $\boldsymbol{\Delta}$, while RF, NT do not. Note that the Bayes error is not achieved in this model.

## 4 Numerical Experiments

For the experiments illustrated in Figures 1 and 2, we use feature size of $d = 450$, and number of hidden units $N \in \{45, \cdots, 4500\}$. NT and NN models are trained with SGD in TensorFlow [1]. We run a total of $2 \times 10^5$ SGD steps for each (qf) model and $1.4 \times 10^5$ steps for each (mg) model. The SGD batch size is fixed at 100 and the step size is chosen from the grid $\{0.001, \cdots, 0.03\}$ where the hyper-parameter that achieves the best fit is used for the figures. RF models are fitted directly by solving KKT conditions with $5 \times 10^5$ observations. After fitting the model, the test error is evaluated on $10^4$ fresh samples. In our figures, each RF data point corresponds to the test error averaged over 10 models with independent realizations of $\boldsymbol{W}$.

For (qf) experiments, we choose $\boldsymbol{B}$ to be diagonal with diagonal elements chosen i.i.d from standard exponential distribution with parameter 1. For (mg) experiments, $\boldsymbol{\Delta}$ is also diagonal with the diagonal element chosen uniformly from the set $\{\frac{2}{\sqrt{d}}, \frac{1.5}{\sqrt{d}}, \frac{1}{\sqrt{d}}\}$. Experiments with non-diagonal $\boldsymbol{\Delta}$ are presented in the appendix.

While we are only able to provide theory for NN and NT when the activations are quadratic, we have performed extensive experiments examining the behavior of these models with other nonlinearities. These results are reported in the appendix. In general, the phenomena we observe in the case of quadratic activations persist when other activations are used. In particular, the positive gap between NN and NT is still present when $N < d$.

## Acknowledgements

This work was partially supported by grants NSF DMS-1613091, CCF-1714305, IIS-1741162, and ONR N00014-18-1-2729, NSF DMS-1418362, NSF DMS-1407813.

## Footnotes

[1]For simplicity, we focus our introductory discussion on the case in which the response $y_i$ is a noiseless function of the feature vector $\boldsymbol{x}_i$: some of our results go beyond this setting.

[2]Notice that we do not add an offset in the RF model, and will limit ourselves to target functions $f_*$ that are centered: this choice simplifies some calculations without modifying the results.

[3]Note that [15] considers feature vectors $\boldsymbol{x}_i$ uniformly random over the sphere rather than Gaussian. However, the results of [15] can be generalized, with certain modifications, to the Gaussian case. Roughly speaking, for Gaussian features, NT with $N = O(d)$ neurons can represent quadratic functions, and a low-dimensional subspace of higher order polynomials.

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
