[Supplementary Material]

# A  Technical background

## A.1  Hermite polynomials

The Hermite polynomials $\{\mathrm{He}_k\}_{k \geq 0}$ form an orthogonal basis of $L^2(\mathbb{R}, \gamma)$, where $\gamma(\mathrm{d}x) = e^{-x^2/2}\mathrm{d}x/\sqrt{2\pi}$ is the standard Gaussian measure, and $\mathrm{He}_k$ has degree $k$. We will follow the classical normalization (here and below, expectation is with respect to $G \sim \mathsf{N}(0,1)$):

$$\mathbb{E}\big\{\mathrm{He}_j(G)\,\mathrm{He}_k(G)\big\} = k!\,\delta_{jk}\,. \tag{18}$$

As a consequence, for any function $g \in L^2(\mathbb{R}, \gamma)$, we have the decomposition

$$g(x) = \sum_{k=0}^{\infty} \frac{\mu_k(g)}{k!}\,\mathrm{He}_k(x)\,, \qquad \mu_k(g) \equiv \mathbb{E}\big\{g(G)\,\mathrm{He}_k(G)\big\}\,. \tag{19}$$

## A.2  Notations

Throughout the proofs, $O_d(\,\cdot\,)$ (resp. $o_d(\,\cdot\,)$) denotes the standard big-O (resp. little-o) notation, where the subscript $d$ emphasizes the asymptotic variable. We denote $O_{d,\mathbb{P}}(\,\cdot\,)$ (resp. $o_{d,\mathbb{P}}(\,\cdot\,)$) the big-O (resp. little-o) in probability notation: $h_1(d) = O_{d,\mathbb{P}}(h_2(d))$ if for any $\varepsilon > 0$, there exists $C_\varepsilon > 0$ and $d_\varepsilon \in \mathbb{Z}_{>0}$, such that

$$\mathbb{P}(|h_1(d)/h_2(d)| > C_\varepsilon) \leq \varepsilon, \qquad \forall d \geq d_\varepsilon,$$

and respectively: $h_1(d) = o_{d,\mathbb{P}}(h_2(d))$, if $h_1(d)/h_2(d)$ converges to 0 in probability.

We will occasionally hide logarithmic factors using the $\tilde{O}_d(\,\cdot\,)$ notation (resp. $\tilde{o}_d(\,\cdot\,)$): $h_1(d) = \tilde{O}_d(h_2(d))$ if there exists a constant $C$ such that $h_1(d) \leq C(\log d)^C h_2(d)$. Similarly, we will denote $\tilde{O}_{d,\mathbb{P}}(\,\cdot\,)$ (resp. $\tilde{o}_{d,\mathbb{P}}(\,\cdot\,)$) when considering the big-O in probability notation up to a logarithmic factor.

# B  Proofs for quadratic functions

Our results for quadratic functions (qf) assume $\boldsymbol{x}_i \sim \mathsf{N}(0, \mathbf{I}_d)$ and $y_i = f_*(\boldsymbol{x}_i)$ where

$$f_*(\boldsymbol{x}_i) \equiv b_0 + \langle \boldsymbol{x}, \boldsymbol{B}\boldsymbol{x} \rangle\,. \tag{20}$$

Throughout this section, we will denote $\mathbb{E}_{\boldsymbol{x}}$ the expectation operator with respect to $\boldsymbol{x} \sim \mathsf{N}(0, \mathbf{I}_d)$, and $\mathbb{E}_{\boldsymbol{w}}$ the expectation operator with respect to $\boldsymbol{w} \sim \mathsf{N}(0, \boldsymbol{\Gamma})$.

## B.1  Random Features model: proof of Theorem 1

Recall the definition

$$R_{\mathsf{RF},N}(f_*) = \min_{\hat{f} \in \mathcal{F}_{\mathsf{RF},N}(\boldsymbol{W})} \mathbb{E}\big\{(f_*(\boldsymbol{x}) - \hat{f}(\boldsymbol{x}))^2\big\},$$

where

$$\mathcal{F}_{\mathsf{RF},N}(\boldsymbol{W}) = \Big\{f_N(\boldsymbol{x}) = \sum_{i=1}^{N} a_i \sigma(\langle \boldsymbol{w}_i, \boldsymbol{x} \rangle) :\ a_i \in \mathbb{R}, i \in [N]\Big\}.$$

Note that it is easy to see from the proof that the result stays the same if we add an offset $c$.

### B.1.1  Representation of the RF risk

**Lemma 1.** *Consider the* RF *model. We have*

$$R_{\mathsf{RF},N}(f_*) = \mathbb{E}_{\boldsymbol{x}}[f_*(\boldsymbol{x})^2] - \boldsymbol{V}^\top \boldsymbol{U}^{-1} \boldsymbol{V}, \tag{21}$$

*where $\boldsymbol{V} = [V_1, \dots, V_N]^\top$, and $\boldsymbol{U} = (U_{ij})_{i,j \in [N]}$, with*

$$V_i = \mathbb{E}_{\boldsymbol{x}}[f_*(\boldsymbol{x})\sigma(\langle \boldsymbol{w}_i, \boldsymbol{x} \rangle)],$$
$$U_{ij} = \mathbb{E}_{\boldsymbol{x}}[\sigma(\langle \boldsymbol{w}_i, \boldsymbol{x} \rangle)\sigma(\langle \boldsymbol{w}_j, \boldsymbol{x} \rangle)].$$

*Proof of Lemma 1.* Simply write the KKT conditions. The optimum is achieved at $\boldsymbol{a} = \boldsymbol{U}^{-1}\boldsymbol{V}$. $\qquad\square$

### B.1.2 Approximation of kernel matrix $\boldsymbol{U}$

**Lemma 2.** *Let $\sigma \in L^2(\mathbb{R}, \gamma)$ be an activation function. Denote $\lambda_k = \mathbb{E}_{G \sim \mathsf{N}(0,1)}[\sigma(G)\mathrm{He}_k(G)]$ the $k$-th Hermite coefficient of $\sigma$ and assume $\lambda_0 = 0$. Let $\boldsymbol{U} = (U_{ij})_{i,j \in [N]}$ be a random matrix with*

$$U_{ij} = \mathbb{E}_{\boldsymbol{x}}[\sigma(\langle \boldsymbol{w}_i, \boldsymbol{x}\rangle)\sigma(\langle \boldsymbol{w}_j, \boldsymbol{x}\rangle)],$$

*where $(\boldsymbol{w}_i)_{i \in [N]} \sim \mathsf{N}(\boldsymbol{0}, \boldsymbol{\Gamma})$ independently. Assume conditions **A1** and **A2** hold.*
  *Let $\boldsymbol{W} = (\boldsymbol{w}_1, \ldots, \boldsymbol{w}_N) \in \mathbb{R}^{d \times N}$, and denote $\boldsymbol{U}_0 = \{(U_0)_{ij}\}_{i,j \in [N]}$, with*

$$(U_0)_{ij} = \tilde{\lambda}\delta_{ij} + \lambda_1^2\langle \boldsymbol{w}_i, \boldsymbol{w}_j\rangle + \kappa/d + \mu_i\mu_j,$$

*where*

$$
\begin{aligned}
\mu_i &= \lambda_2(\|\boldsymbol{w}_i\|_2^2 - 1)/2, \\
\tilde{\lambda} &= \mathbb{E}[\sigma(G)^2] - \lambda_1^2, \\
\kappa &= d\lambda_2^2\mathrm{Tr}(\boldsymbol{\Gamma}^2)/2.
\end{aligned}
$$

*Then we have as $N/d = \rho$ and $d \to \infty$,*

$$\|\boldsymbol{U} - \boldsymbol{U}_0\|_{\mathrm{op}} = o_{d,\mathbb{P}}(1).$$

*Proof of Lemma 2.*
**Step 1. Hermite expansion of $\sigma$ for $\|\boldsymbol{w}_i\|_2 \neq 1$.** Denote $\sigma_i(x) = \sigma(\|\boldsymbol{w}_i\|_2 \cdot x)$. First notice that by a change of variables, we get

$$\mathbb{E}[\sigma(tG)] = \mathbb{E}[(\sigma(G)/t)\exp(G^2(1 - 1/t^2)/2)]. \tag{22}$$

By Assumption **A1**, there exists $c_1 < 1$ such that

$$\sigma(u)^2\exp(u^2(1 - 1/t^2)) \leq c_0\exp(u^2(c_1/2 + 1 - 1/t^2)).$$

Hence for $|t - 1|$ sufficiently small, we have $\sigma_i \in L^2(\mathbb{R}, \gamma)$ and we can consider its Hermite expansion

$$\sigma_i(x) = \sum_{k=0}^{\infty}\frac{\zeta_k(\sigma_i)}{k!}\mathrm{He}_k(x),$$

where

$$\zeta_k(\sigma_i) = \mathbb{E}_{G \sim \mathsf{N}(0,1)}[\sigma(\|\boldsymbol{w}_i\|_2 G)\mathrm{He}_k(G)].$$

Denote the Hermite expansion of $\sigma$ to be

$$\sigma(x) = \sum_{k=0}^{\infty}\lambda_k(\sigma)\mathrm{He}_k(x)/k!,$$

where

$$\lambda_k(\sigma) = \mathbb{E}_{G \sim \mathsf{N}(0,1)}[\sigma(G)\mathrm{He}_k(G)].$$

By dominated convergence theorem, we have

$$\lim_{t \to 1}\mathbb{E}_{G \sim \mathsf{N}(0,1)}[(\sigma(G) - \sigma(tG))^2] = 0.$$

In addition, by sub-Gaussianity of the norm of a multivariate Gaussian random variable (see [Ver10]), it is easy to show that

$$\sup_{i \in [N]}|\|\boldsymbol{w}_i\|_2 - 1| = o_{d,\mathbb{P}}(1). \tag{23}$$

Hence we have

$$\sup_{i \in [N]} \|\sigma - \sigma_i\|_{L^2} = o_{d,\mathbb{P}}(1),$$

$$\sup_{i \in [N]} |\zeta_k(\sigma_i) - \lambda_k(\sigma)| \le \sup_{i \in [N]} \|\sigma - \sigma_i\|_{L^2} \mathbb{E}[\mathrm{He}_k(G)^2]^{1/2} = o_{d,\mathbb{P}}(1), \tag{24}$$

for any fixed integer $k$.

**Step 2. Expansion of $U$.** Denote $\boldsymbol{u}_i = \boldsymbol{w}_i / \|\boldsymbol{w}_i\|_2$, then we have

$$U_{ij} = \underbrace{\zeta_0(\sigma_i)\zeta_0(\sigma_j)}_{T_{0,ij}} + \underbrace{\zeta_1(\sigma_i)\zeta_1(\sigma_j)\langle \boldsymbol{u}_i, \boldsymbol{u}_j \rangle}_{T_{1,ij}} + \underbrace{\zeta_2(\sigma_i)\zeta_2(\sigma_j)\frac{\langle \boldsymbol{u}_i, \boldsymbol{u}_j \rangle^2}{2}}_{T_{2,ij}} + \underbrace{\sum_{k \ge 3} \zeta_k(\sigma_i)\zeta_k(\sigma_j)\frac{\langle \boldsymbol{u}_i, \boldsymbol{u}_j \rangle^k}{k!}}_{T_{3,ij}}. \tag{25}$$

We define

$$\boldsymbol{T}_k = \left( \zeta_k(\sigma_i)\zeta_k(\sigma_j)\frac{\langle \boldsymbol{w}_i, \boldsymbol{w}_j \rangle^k}{k!} \right)_{i,j \in [N]}.$$

**Step 3. Term $\boldsymbol{T}_0$.** By definition of $\mu_i$, we have

$$\boldsymbol{T}_0 = (\zeta_0(\sigma_i)\zeta_0(\sigma_j))_{i,j \in [N]} = \boldsymbol{D}_0[(\lambda_2/2)^2(\|\boldsymbol{w}_i\|_2^2 - 1)(\|\boldsymbol{w}_j\|_2^2 - 1)]_{i,j \in [N]}\boldsymbol{D}_0,$$

where (by the assumption that $\mathbb{E}_G[\sigma(G)] = 0$)

$$(\boldsymbol{D}_0)_{ii} = \frac{\zeta_0(\sigma_i)}{\lambda_2(\|\boldsymbol{w}_i\|_2^2 - 1)/2} = \mathbb{E}\left[ \frac{\sigma(\|\boldsymbol{w}_i\|G) - \sigma(G)}{\|\boldsymbol{w}_i\|_2 - 1} \right] \cdot \frac{1}{\lambda_2(\|\boldsymbol{w}_i\|_2 + 1)/2}.$$

Let us show:

$$\lim_{t \to 1} \mathbb{E}\left[ \frac{\sigma(tG) - \sigma(G)}{t - 1} \right] = \lambda_2(\sigma), \tag{26}$$

or equivalently:

$$\lim_{t \to 1} \mathbb{E}\left[ \frac{\sigma(tG) - \sigma(G)}{t - 1} - (G^2 - 1)\sigma(G) \right] = 0$$

Recall the change of variable (22) and do a first order Taylor expansion of the exponential: there exists a function $\xi(G) \in [0, G]$ such that

$$\mathbb{E}\left[ \frac{\sigma(tG) - \sigma(G)}{t - 1} - (G^2 - 1)\sigma(G) \right]$$

$$= \mathbb{E}\left[ \sigma(G)\Big( \exp(G^2(1 - 1/t^2)/2) - t - t(t-1)(G^2 - 1) \Big) \right] \cdot \frac{1}{t(t-1)}$$

$$= \mathbb{E}\left[ \sigma(G)(t-1)\Big( 1 - G^2[2t+1]/(2t^2) + G^4(t+1)^2/(8t^4)\exp(\xi(G)^2(1 - 1/t^2)/2) \Big) \right] \cdot \frac{1}{t}.$$

We see that the integrand goes to zero as $t \to 1$. For $|t - 1|$ sufficiently small, we have

$$\frac{\left| \exp(G^2(1 - 1/t^2)/2) - t - t(t-1)(G^2 - 1) \right|}{|t - 1|} \le 2 + 2G^2 + 2G^4\exp(G^2/5),$$

which is squared integrable. Recalling that $\sigma \in L^2(\mathbb{R}, \gamma)$, we obtain (26) by dominated convergence.

Hence, combining (23) and (26) gives

$$\|\boldsymbol{D}_0 - \mathbf{I}_d\|_{\mathrm{op}} = o_{d,\mathbb{P}}(1).$$

Furthermore, for $\boldsymbol{\mu} = (\mu_i)_{i \in [N]}$ with $\mu_i = \lambda_2(\|\boldsymbol{w}_i\|_2^2 - 1)/2$, we have

$$\mathbb{E}[\|\boldsymbol{\mu}\boldsymbol{\mu}^\mathsf{T}\|_{\mathrm{op}}] = \mathbb{E}[\|\boldsymbol{\mu}\|_2^2] = \frac{\lambda_2^2}{4}N\mathbb{E}[(\|\boldsymbol{w}_i\|_2^2 - 1)^2] = \frac{\lambda_2^2}{2}N\|\boldsymbol{\Gamma}\|_F^2 \le \frac{\lambda_2^2}{2}N^2\|\boldsymbol{\Gamma}\|_{\mathrm{op}}^2 = O_{d,\mathbb{P}}(1),$$

where the last equality comes from assumption **A2**. We get

$$\|\boldsymbol{T}_0 - \boldsymbol{\mu}\boldsymbol{\mu}^\mathsf{T}\|_{\mathrm{op}} \le 2\|\boldsymbol{D}_0 - \mathbf{I}_d\|_{\mathrm{op}}\|\boldsymbol{\mu}\boldsymbol{\mu}^\mathsf{T}\|_{\mathrm{op}}(\|\boldsymbol{D}_0\|_{\mathrm{op}} + 1) = o_{d,\mathbb{P}}(1). \tag{27}$$

**Step 4. Term $\boldsymbol{T}_1$.**  For $\boldsymbol{T}_1$, we have

$$\boldsymbol{T}_1 = (\zeta_1(\sigma_i)\zeta_1(\sigma_j)\langle \boldsymbol{u}_i, \boldsymbol{u}_j\rangle)_{i,j\in[N]} = \boldsymbol{D}_1 \boldsymbol{W}^\mathsf{T}\boldsymbol{W}\boldsymbol{D}_1,$$

where

$$\boldsymbol{D}_1 = \mathrm{diag}((\zeta_1(\sigma_i))/\|\boldsymbol{w}_i\|_2).$$

By the uniform convergence of $\zeta_1(\sigma_i)$ to $\lambda_1(\sigma)$, cf Eq. (24), we have

$$\|\boldsymbol{D}_1 - \lambda_1(\sigma)\mathbf{I}_d\|_{\mathrm{op}} = o_{d,\mathbb{P}}(1).$$

Moreover, we have

$$\|\boldsymbol{W}^\mathsf{T}\boldsymbol{W}\|_{\mathrm{op}} = \|\boldsymbol{W}\boldsymbol{W}^\mathsf{T}\|_{\mathrm{op}} \le \|\sqrt{d}\boldsymbol{\Gamma}^{1/2}\|_{\mathrm{op}}^2\|\boldsymbol{G}\boldsymbol{G}^\mathsf{T}\|_{\mathrm{op}} = O_{d,\mathbb{P}}(1),$$

where we denoted by $\boldsymbol{G}$ the matrix with columns $\boldsymbol{g}_i \sim \mathsf{N}(\boldsymbol{0}, \mathbf{I}_d/d)$. Hence, we have

$$\|\boldsymbol{T}_1 - \lambda_1^2\boldsymbol{W}^\mathsf{T}\boldsymbol{W}\|_{\mathrm{op}} \le \|\boldsymbol{D}_1 - \lambda_1\mathbf{I}_d\|_{\mathrm{op}}\|\boldsymbol{W}^\mathsf{T}\boldsymbol{W}\|_{\mathrm{op}}(\|\boldsymbol{D}_1\|_{\mathrm{op}} + 1) = o_{d,\mathbb{P}}(1). \tag{28}$$

**Step 5. Term $\boldsymbol{T}_2$.**  We have

$$\boldsymbol{T}_2 = (\zeta_2(\sigma_i)\zeta_2(\sigma_j)\langle \boldsymbol{u}_i, \boldsymbol{u}_j\rangle^2/2)_{i,j\in[N]} = \boldsymbol{D}_2(\langle \boldsymbol{w}_i, \boldsymbol{w}_j\rangle^2/2)_{i,j\in[N]}\boldsymbol{D}_2,$$

where

$$\boldsymbol{D}_2 = \mathrm{diag}((\zeta_2(\sigma_i))/\|\boldsymbol{w}_i\|_2^2).$$

By the uniform convergence of $\zeta_2(\sigma_i)$ to $\lambda_2(\sigma)$, we have

$$\|\boldsymbol{D}_2 - \lambda_2\mathbf{I}_d\|_{\mathrm{op}} = o_{d,\mathbb{P}}(1).$$

Moreover, we have (see below)

$$\|(\langle \boldsymbol{w}_i, \boldsymbol{w}_j\rangle^2)_{i,j\in[N]}\|_{\mathrm{op}} = O_{d,\mathbb{P}}(1).$$

Hence, we have

$$\|\boldsymbol{T}_2 - \lambda_2^2(\langle \boldsymbol{w}_i, \boldsymbol{w}_j\rangle^2/2)_{i,j\in[N]}\|_{\mathrm{op}} \le \|\boldsymbol{D}_2 - \lambda_2\mathbf{I}_d\|_{\mathrm{op}}\|(\langle \boldsymbol{w}_i, \boldsymbol{w}_j\rangle^2/2)_{i,j\in[N]}\|_{\mathrm{op}}(\|\boldsymbol{D}_2\|_{\mathrm{op}} + 1) = o_{d,\mathbb{P}}(1).$$

Moreover, by the estimates in proof of Theorem 2.1 in [EK$^+$10], we have

$$\|(\langle \boldsymbol{w}_i, \boldsymbol{w}_j\rangle^2/2)_{i,j\in[N]} - [\mathrm{Tr}(\boldsymbol{\Gamma}^2)/2]\mathbf{1}\mathbf{1}^\mathsf{T} - (1/2)\mathbf{I}_N\|_{\mathrm{op}} = o_{d,\mathbb{P}}(1).$$

Hence, we get

$$\|\boldsymbol{T}_2 - \lambda_2^2[\mathrm{Tr}(\boldsymbol{\Gamma}^2)/2]\mathbf{1}\mathbf{1}^\mathsf{T} - [\lambda_2^2/2]\mathbf{I}_N\|_{\mathrm{op}} = o_{d,\mathbb{P}}(1). \tag{29}$$

**Step 6. Term $\sum_{k\ge 3}\mathrm{ddiag}(\boldsymbol{T}_k)$.**  Denote $\mathrm{ddiag}(\boldsymbol{T}_k)$ the diagonal matrix composed of diagonal entries of $\boldsymbol{T}_k$. We have

$$\left|\sum_{k\ge 3}((T_k)_{ii} - \lambda_k(\sigma)^2/k!)\right| = \left|\|\sigma_i\|_{L^2}^2 - \sum_{k=0}^{2}\zeta_k(\sigma_i)^2/k! - \|\sigma\|_{L^2}^2 + \sum_{k=0}^{2}\lambda_k(\sigma)^2/k!\right|$$

$$\le \|\sigma - \sigma_i\|_{L^2}[2\|\sigma\|_{L^2} + \|\sigma - \sigma_i\|_{L^2}] + \sum_{k=0}^{2}|\zeta_k(\sigma_i)^2 - \lambda_k(\sigma)^2|/k!.$$

Note that we have shown (cf Eq. (24))

$$\sup_{i\in[N]} \max\left\{\|\sigma - \sigma_i\|_{L^2}, \max_{k=0,1,2}|\zeta_k(\sigma_i) - \lambda_k(\sigma)|\right\} = o_{d,\mathbb{P}}(1).$$

Therefore, we have

$$\Big\| \sum_{k \geq 3} \mathrm{ddiag}(\boldsymbol{T}_k) - (\tilde{\lambda} - \lambda_2^2/2)\mathbf{I}_N \Big\|_{\mathrm{op}} = o_{d,\mathbb{P}}(1). \tag{30}$$

**Step 7. Term $\sum_{k \geq 3}[\boldsymbol{T}_k - \mathrm{ddiag}(\boldsymbol{T}_k)]$.** We have

$$\Big\| \sum_{k \geq 3}[\boldsymbol{T}_k - \mathrm{ddiag}(\boldsymbol{T}_k)] \Big\|_F \leq \sum_{k \geq 3} \|\boldsymbol{T}_k - \mathrm{ddiag}(\boldsymbol{T}_k)\|_F$$

$$\leq \sum_{k \geq 3} \Big[ \Big( \sum_{i,j=1}^N \zeta_k(\sigma_i)^2 \zeta_k(\sigma_j)^2 \Big) \Big( \sup_{i \neq j} \langle \boldsymbol{u}_i, \boldsymbol{u}_j \rangle^{2k}/(k!)^2 \Big) \Big]^{1/2}$$

$$\leq \Big[ \sum_{k \geq 3} \sum_{i=1}^N \zeta_k(\sigma_i)^2/k! \Big] \max_{i \neq j} \langle \boldsymbol{u}_i, \boldsymbol{u}_j \rangle^3$$

$$\leq \|\sigma_i\|_{L^2}^2 \times N \max_{i \neq j} \langle \boldsymbol{u}_i, \boldsymbol{u}_j \rangle^3.$$

Note we have $\max_{i \in [N]} \|\sigma_i\|_{L^2}^2 = O_{d,\mathbb{P}}(1)$. Moreover, we have (see for example Lemma 10 in [GMMM19])

$$\max_{i \neq j} \langle \boldsymbol{u}_i, \boldsymbol{u}_j \rangle^3 = \tilde{O}_{d,\mathbb{P}}(d^{-3/2}).$$

Therefore, we have

$$\Big\| \sum_{k \geq 3}[\boldsymbol{T}_k - \mathrm{ddiag}(\boldsymbol{T}_k)] \Big\|_F = o_{d,\mathbb{P}}(1). \tag{31}$$

Combining the bounds (27), (28), (29), (30) and (31) into the decomposition (25) proves the lemma. $\square$

### B.1.3 Approximation of the $\boldsymbol{V}$ vector

**Lemma 3.** *Under the assumptions of Theorem 1, define $\boldsymbol{V} = (V_1, \ldots, V_N)^\mathsf{T}$ with*

$$V_i = \mathbb{E}_{\boldsymbol{x}}[f_*(\boldsymbol{x})\sigma(\langle \boldsymbol{w}_i, \boldsymbol{x} \rangle)]$$

*where $(\boldsymbol{w}_i)_{i \in [N]} \sim \mathsf{N}(\boldsymbol{0}, \boldsymbol{\Gamma})$ independently. Then as $N/d = \rho$ with $d \to \infty$, we have*

$$\|\boldsymbol{V} - \tau \mathbf{1}/\sqrt{d}\|_2^2 = \|\boldsymbol{B}\|_F^2 \cdot o_{d,\mathbb{P}}(1),$$

*where*

$$\tau = \sqrt{d} \cdot \lambda_2 \mathrm{Tr}(\boldsymbol{B\Gamma}).$$

*Proof of Lemma 3.* Without loss of generality, we assume $\|\boldsymbol{B}\|_F = 1$ in the proof (it suffices to divide $V_i$ by $\|\boldsymbol{B}\|_F$). Consider $\boldsymbol{w}_i \in \mathbb{R}^d$. Take $\boldsymbol{R}$ to be an orthogonal matrix such that $\boldsymbol{R}\boldsymbol{w}_i = \|\boldsymbol{w}_i\|_2 \boldsymbol{e}_1$, then we have

$$V_i = \mathbb{E}_{\boldsymbol{x}}[f_*(\boldsymbol{R}^\mathsf{T}\boldsymbol{x})\sigma(\|\boldsymbol{w}_i\|_2 x_1)]$$

$$= \mathbb{E}_{\boldsymbol{x}}[(\langle \boldsymbol{x}, \boldsymbol{R}\boldsymbol{B}\boldsymbol{R}^\mathsf{T}\boldsymbol{x} \rangle - \mathrm{Tr}(\boldsymbol{B}))\sigma(\|\boldsymbol{w}_i\|_2 x_1)]$$

$$= \mathbb{E}_{x_1}\Big[ \Big( x_1^2 \frac{\langle \boldsymbol{w}_i, \boldsymbol{B}\boldsymbol{w}_i \rangle}{\|\boldsymbol{w}_i\|_2^2} + \mathrm{Tr}(\boldsymbol{P}_{\perp \boldsymbol{w}_i}\boldsymbol{B}) - \mathrm{Tr}(\boldsymbol{B}) \Big)\sigma(\|\boldsymbol{w}_i\|_2 x_1) \Big]$$

$$= \mathbb{E}_{x_1}\Big[ (x_1^2 - 1) \frac{\langle \boldsymbol{w}_i, \boldsymbol{B}\boldsymbol{w}_i \rangle}{\|\boldsymbol{w}_i\|_2^2}\sigma(\|\boldsymbol{w}_i\|_2 x_1) \Big]$$

$$\equiv \frac{\langle \boldsymbol{w}_i, \boldsymbol{B}\boldsymbol{w}_i \rangle}{\|\boldsymbol{w}_i\|_2^2} \zeta_2(\sigma_i),$$

where $\boldsymbol{P}_{\perp \boldsymbol{w}_i}$ is the projection on the hyperplane orthogonal to $\boldsymbol{w}_i$, and we recall the definition of $\zeta_2(\sigma_i)$ of Lemma 2:

$$\zeta_2(\sigma_i) = \mathbb{E}_G[(G^2 - 1)\sigma(\|\boldsymbol{w}_i\|_2 G)],$$

with $G$ a standard normal random variable.

We define the following interpolating variables:

$$V_i^{(1)} = \frac{\langle \boldsymbol{w}_i, \boldsymbol{B}\boldsymbol{w}_i \rangle}{\|\boldsymbol{w}_i\|_2^2}\lambda_2, \qquad V_i^{(2)} = \langle \boldsymbol{w}_i, \boldsymbol{B}\boldsymbol{w}_i \rangle \lambda_2, \qquad V_i^{(3)} = \mathrm{Tr}(\boldsymbol{\Gamma B})\lambda_2,$$

and the associated vectors $\boldsymbol{V}^{(1)}$, $\boldsymbol{V}^{(2)}$ and $\boldsymbol{V}^{(3)}$. We bound successively the distance between these vectors. We will denote by $\boldsymbol{P}_{\boldsymbol{w}_i}$ the projection onto vector $\boldsymbol{w}_i$. First, we consider:

$$\|\boldsymbol{V} - \boldsymbol{V}^{(1)}\|_2^2 = \sum_{i=1}^N \mathrm{Tr}(\boldsymbol{P}_{\boldsymbol{w}_i}\boldsymbol{B})^2 (\zeta_2(\sigma_i) - \lambda_2)^2.$$

One can check, using a similar argument as for Eq. (26) and dominated convergence, that

$$\lim_{t \to 1} \frac{\mathbb{E}[(G^2 - 1)(\sigma(tG) - \sigma(G))]}{t - 1} = \lambda_4(\sigma) + 2\lambda_2(\sigma). \tag{32}$$

Hence, recalling (23), we have

$$\|\boldsymbol{V} - \boldsymbol{V}^{(1)}\|_2^2 = O_{d,\mathbb{P}}\bigg(\Big(\sup_{i \in [N]} \mathrm{Tr}(\boldsymbol{P}_{\boldsymbol{w}_i}\boldsymbol{B})\Big)^2 \sum_{i=1}^N (\|\boldsymbol{w}_i\|_2 - 1)^2\bigg). \tag{33}$$

Let us first show that the sum is bounded with high probability: denoting $\boldsymbol{g} \sim \mathsf{N}(\boldsymbol{0}, \mathbf{I}_d)$, classical sub-Gaussian concentration inequalities (see for example Theorem 6.3.2 in [Ver10]) shows that

$$\Big\|\|\boldsymbol{\Gamma}^{1/2}\boldsymbol{g}\|_2 - \|\boldsymbol{\Gamma}^{1/2}\|_F\Big\|_{\psi_2} \le C\|\boldsymbol{\Gamma}^{1/2}\|_{\mathrm{op}}, \tag{34}$$

where $\|\cdot\|_{\psi_2}$ denotes the sub-Gaussian Orlicz norm. By assumption, we have $\|\boldsymbol{\Gamma}^{1/2}\|_{\mathrm{op}} = \|\boldsymbol{\Gamma}\|_{\mathrm{op}}^{1/2} = O_d(d^{-1/2})$, and $\|\boldsymbol{\Gamma}^{1/2}\|_F = \sqrt{\mathrm{Tr}\boldsymbol{\Gamma}} = 1$. Hence, for $\boldsymbol{w}_i \sim \mathsf{N}(\boldsymbol{0}, \boldsymbol{\Gamma})$, we have

$$\Big\|\sqrt{d}\|\boldsymbol{w}_i\|_2 - \sqrt{d}\Big\|_{\psi_2} \le C. \tag{35}$$

Therefore, we have

$$\sum_{i=1}^N (\|\boldsymbol{w}_i\|_2 - 1)^2 = O_{d,\mathbb{P}}(1). \tag{36}$$

Furthermore, we readily have (for example from (23))

$$\sup_{i \in [N]} \|\boldsymbol{w}_i\|^{-4} = O_{d,\mathbb{P}}(1). \tag{37}$$

Noticing that $\mathrm{Tr}(\boldsymbol{w}_i\boldsymbol{w}_i^\mathsf{T}\boldsymbol{B}) = \|\boldsymbol{B}^{1/2}\boldsymbol{w}_i\|_2^2$ and by the same argument as for (34), we have:

$$\Big\|\|\boldsymbol{B}^{1/2}\boldsymbol{\Gamma}^{1/2}\boldsymbol{g}\|_2 - \mathbb{E}[\|\boldsymbol{B}^{1/2}\boldsymbol{\Gamma}^{1/2}\boldsymbol{g}\|_2]\Big\|_{\psi_2} \le C\|\boldsymbol{B}^{1/2}\boldsymbol{\Gamma}^{1/2}\|_{\mathrm{op}}. \tag{38}$$

By assumption **A2**, we have $\|\boldsymbol{B}^{1/2}\boldsymbol{\Gamma}^{1/2}\|_{\mathrm{op}} \le \|\boldsymbol{B}^{1/2}\|_{\mathrm{op}}\|\boldsymbol{\Gamma}^{1/2}\|_{\mathrm{op}} = O_d(d^{-1/2})$ and

$$\mathbb{E}[\|\boldsymbol{B}^{1/2}\boldsymbol{\Gamma}^{1/2}\boldsymbol{g}\|_2] \le (\mathbb{E}[\|\boldsymbol{B}^{1/2}\boldsymbol{\Gamma}^{1/2}\boldsymbol{g}\|_2^2])^{1/2} = \mathrm{Tr}(\boldsymbol{\Gamma B})^{1/2} \le \|\boldsymbol{\Gamma}\|_F^{1/2}\|\boldsymbol{B}\|_F^{1/2} \le \|\boldsymbol{\Gamma}\|_{\mathrm{op}}^{1/4}\mathrm{Tr}(\boldsymbol{\Gamma})^{1/4} = O_d(d^{-1/2}),$$

which combined with (38) yields

$$\sup_{i \in [N]} \|\boldsymbol{B}^{1/2}\boldsymbol{w}_i\|_2^2 = o_{d,\mathbb{P}}(1). \tag{39}$$

Combining the bounds (36), (37) and (39) into (33), we get

$$\|\boldsymbol{V} - \boldsymbol{V}^{(1)}\|_2^2 = o_{d,\mathbb{P}}(1). \tag{40}$$

Consider now

$$\|\boldsymbol{V}^{(1)} - \boldsymbol{V}^{(2)}\|_2^2 = \sum_{i=1}^N \lambda_2^2 \langle \boldsymbol{w}_i, \boldsymbol{B}\boldsymbol{w}_i \rangle^2 \Big( \frac{1}{\|\boldsymbol{w}_i\|_2^2} - 1 \Big)^2$$
$$\leq \lambda_2^2 \Big( \sup_{i \in [N]} \|\boldsymbol{B}^{1/2}\boldsymbol{w}_i\|_2^2 / \|\boldsymbol{w}_i\|_2^2 \Big) \sum_{i=1}^N (\|\boldsymbol{w}_i\|_2^2 - 1)^2. \tag{41}$$

We have

$$\mathbb{E}_{\boldsymbol{w}_i \sim \mathsf{N}(\boldsymbol{0},\boldsymbol{\Gamma})}[(\|\boldsymbol{w}_i\|_2^2 - 1)^2] = \mathbb{E}_{\boldsymbol{g} \sim \mathsf{N}(\boldsymbol{0},\mathbf{I})}[(\langle \boldsymbol{g}\boldsymbol{g}^\top, \boldsymbol{\Gamma} \rangle - \mathrm{Tr}(\boldsymbol{\Gamma}))^2] = 2\|\boldsymbol{\Gamma}\|_F^2 = O_{d,\mathbb{P}}(d^{-1}).$$

Hence we must have

$$\sum_{i=1}^N (\|\boldsymbol{w}_i\|_2^2 - 1)^2 = O_{d,\mathbb{P}}(1),$$

which, combined with (39) and (41), yields

$$\|\boldsymbol{V}^{(1)} - \boldsymbol{V}^{(2)}\|_2^2 = o_{d,\mathbb{P}}(1). \tag{42}$$

Consider the last comparison:

$$\|\boldsymbol{V}^{(2)} - \boldsymbol{V}^{(3)}\|_2^2 = \sum_{i=1}^N \lambda_2^2 \Big( \langle \boldsymbol{w}_i, \boldsymbol{B}\boldsymbol{w}_i \rangle - \mathrm{Tr}(\boldsymbol{\Gamma}\boldsymbol{B}) \Big)^2.$$

Taking the expectation:

$$\mathbb{E}_{\boldsymbol{w}_i \sim \mathsf{N}(\boldsymbol{0},\boldsymbol{\Gamma})}[(\langle \boldsymbol{w}_i, \boldsymbol{B}\boldsymbol{w}_i \rangle - \mathrm{Tr}(\boldsymbol{\Gamma}\boldsymbol{B}))^2] = \mathbb{E}_{\boldsymbol{g} \sim \mathsf{N}(\boldsymbol{0},\mathbf{I})}[(\langle \boldsymbol{g}\boldsymbol{g}^\top, \boldsymbol{\Gamma}^{1/2}\boldsymbol{B}\boldsymbol{\Gamma}^{1/2} \rangle - \mathrm{Tr}(\boldsymbol{\Gamma}\boldsymbol{B}))^2]$$
$$= 2\|\boldsymbol{\Gamma}^{1/2}\boldsymbol{B}\boldsymbol{\Gamma}^{1/2}\|_F^2$$
$$\leq 2\|\boldsymbol{\Gamma}\|_{\mathrm{op}}^2 \|\boldsymbol{B}\|_F^2 = O_d(d^{-2}).$$

We conclude that

$$\sum_{i=1}^N \Big( \langle \boldsymbol{w}_i, \boldsymbol{B}\boldsymbol{w}_i \rangle - \mathrm{Tr}(\boldsymbol{\Gamma}\boldsymbol{B}) \Big)^2 = o_{d,\mathbb{P}}(1),$$

and therefore

$$\|\boldsymbol{V}^{(2)} - \boldsymbol{V}^{(3)}\|_2^2 = o_{d,\mathbb{P}}(1), \tag{43}$$

where $\boldsymbol{V}^{(3)} = \lambda_2 \mathrm{Tr}(\boldsymbol{\Gamma}\boldsymbol{B})\mathbf{1}$. Combining the above three bounds (33), (42) and (43) yields the desired result. □

### B.1.4 Calculating $\mathbf{1}^\top \boldsymbol{U}_0^{-1}\mathbf{1}/d$

The following proposition is stated in slightly more general terms, in order to be used in both the proofs of Theorem 1 and Theorem 4.

**Proposition 2.** *Let $(\boldsymbol{w}_i)_{i \in [N]} \sim \mathsf{N}(\boldsymbol{0},\boldsymbol{\Gamma})$ independently, where $\boldsymbol{\Gamma}$ satisfies assumption **A2** (resp. **B2**). Denote by $\lambda_k = \mathbb{E}_{G \sim \mathsf{N}(0,1)}[\sigma(G)\mathrm{He}_k(G)]$ the $k$-th Hermite coefficient of $\sigma$. Define $\tilde{\lambda} = \mathbb{E}_{G \sim \mathsf{N}(0,1)}[\sigma(G)^2] - \lambda_1^2$. Consider $\kappa \equiv \kappa(d)$ positive constants that are uniformly upper bounded. Define*

$$\boldsymbol{U}_0 = \boldsymbol{A}_0 + \kappa \mathbf{1}\mathbf{1}^\top/d + \boldsymbol{\mu}\boldsymbol{\mu}^\top,$$

*where*

$$\boldsymbol{A}_0 = \tilde{\lambda}\mathbf{I}_N + \lambda_1^2 \boldsymbol{W}^\mathsf{T}\boldsymbol{W},$$

$$\mu_i = \lambda_2(\|\boldsymbol{w}_i\|_2^2 - 1)/2.$$

*Then we have*

$$\langle \mathbf{1}, \boldsymbol{U}_0^{-1}\mathbf{1}\rangle/d = \psi/(1 + \kappa\psi) + o_{d,\mathbb{P}}(1),$$

*where $\psi > 0$ is the unique solution of*

$$-\tilde{\lambda} = -\frac{\rho}{\psi} + \int \frac{\lambda_1^2 t}{1 + \lambda_1^2 t\psi}\mathcal{D}(\mathrm{d}t),\tag{44}$$

*where $\mathcal{D}$ is the empirical distribution of eigenvalues of $d\cdot\boldsymbol{\Gamma}$.*

The proof of Proposition 2 is a direct combination of Lemma 4, 5, and 6 below.

**Lemma 4.** *Let $(\boldsymbol{w}_i)_{i\in[N]} \sim \mathsf{N}(\mathbf{0},\boldsymbol{\Gamma})$ independently. Assume condition **A2** holds (resp. **B2**). Let $\boldsymbol{\mu} = (\|\boldsymbol{w}_i\|_2^2 - 1)_{i\in[N]}$, and $\boldsymbol{A}_0 = c_1\mathbf{I}_N + c_2\boldsymbol{W}^\mathsf{T}\boldsymbol{W}$, where $c_1 \equiv c_1(d)$ and $c_2 \equiv c_2(d)$ are constants that are asymptotically upper and lower bounded by strictly positive constants. Then as $d \to \infty$ and $N/d \to \rho$, we have*

$$\langle \mathbf{1}, \boldsymbol{A}_0^{-1}\boldsymbol{\mu}\rangle/\sqrt{d} = o_{d,\mathbb{P}}(1).\tag{45}$$

*Proof.* We first prove the lemma under the following extra assumption on the covariance matrix: there exists a (fixed) integer $K$ such that

$$\boldsymbol{\Gamma} = \boldsymbol{Q}\mathrm{diag}(\gamma_1\mathbf{I}_{d_1},\dots,\gamma_K\mathbf{I}_{d_K})\boldsymbol{Q}^\mathsf{T},\tag{46}$$

for some orthogonal matrix $\boldsymbol{Q}$ and $d\cdot\gamma_i \le C$. Furthermore, there exists an $\varepsilon > 0$ such that $d_k/d \ge \varepsilon$ for $d$ sufficiently large.

Without loss of generality, we assume $\boldsymbol{\Gamma} = \mathrm{diag}(\gamma_1\mathbf{I}_{d_1},\dots,\gamma_K\mathbf{I}_{d_K})$, and we divide $\boldsymbol{w}_i$ into vectors corresponding to each block

$$\boldsymbol{w}_i = (\boldsymbol{w}_{i,1};\dots;\boldsymbol{w}_{i,K}) \in \mathbb{R}^d,$$

where $\boldsymbol{w}_{i,k} \in \mathbb{R}^{d_k}$, and we denote $\boldsymbol{W}_k = [\boldsymbol{w}_{1,k},\boldsymbol{w}_{2,k},\dots,\boldsymbol{w}_{N,k}] \in \mathbb{R}^{d_k\times N}$ for $k \in [K]$.

**Step 1. Decouple the randomness.**

Let $(\tilde{\boldsymbol{w}}_i)_{i\in[N]} \sim \mathsf{N}(\mathbf{0},\boldsymbol{\Gamma})$ independently and independent of $(\boldsymbol{w}_i)_{i\in[N]}$. We divide $\tilde{\boldsymbol{w}}_i$ into segments corresponding to each blocks

$$\tilde{\boldsymbol{w}}_i = (\tilde{\boldsymbol{w}}_{i,1};\dots;\tilde{\boldsymbol{w}}_{i,K}),$$

where $\tilde{\boldsymbol{w}}_{i,k} \in \mathbb{R}^{d_k}$, and we denote $\tilde{\boldsymbol{W}}_k = [\tilde{\boldsymbol{w}}_{1,k},\tilde{\boldsymbol{w}}_{2,k},\dots,\tilde{\boldsymbol{w}}_{N,k}] \in \mathbb{R}^{d_k\times N}$ for $k \in [K]$.

Define

$$\boldsymbol{D}_{k,\boldsymbol{w}} = \mathrm{diag}(\|\boldsymbol{w}_{1,k}\|_2,\dots,\|\boldsymbol{w}_{N,k}\|_2) \in \mathbb{R}^{N\times N},$$

$$\boldsymbol{D}_{k,\tilde{\boldsymbol{w}}} = \mathrm{diag}(\|\tilde{\boldsymbol{w}}_{1,k}\|_2,\dots,\|\tilde{\boldsymbol{w}}_{N,k}\|_2) \in \mathbb{R}^{N\times N}.$$

Using the fact that $\|\boldsymbol{g}\|_2$ is independent of $\boldsymbol{g}/\|\boldsymbol{g}\|_2$ for $\boldsymbol{g} \sim \mathsf{N}(\mathbf{0},\mathbf{I})$, the following two sets of random variables have the same distribution:

$$\left\{(\boldsymbol{W}_k^\mathsf{T}\boldsymbol{W}_k)_{k\in[K]}, (\|\boldsymbol{w}_{ik}\|_2)_{i\in[N],k\in[K]}\right\} \overset{\mathrm{d}}{=} \left\{(\boldsymbol{D}_{k,\boldsymbol{w}}\boldsymbol{D}_{k,\tilde{\boldsymbol{w}}}^{-1}\tilde{\boldsymbol{W}}_k^\mathsf{T}\tilde{\boldsymbol{W}}_k\boldsymbol{D}_{k,\tilde{\boldsymbol{w}}}^{-1}\boldsymbol{D}_{k,\boldsymbol{w}})_{k\in[K]}, (\|\boldsymbol{w}_{ik}\|_2)_{i\in[N],k\in[K]}\right\}.$$

Define

$$\bar{\boldsymbol{A}}_0 = c_1\mathbf{I}_d + c_2\sum_{k\in[K]}\boldsymbol{D}_{k,\boldsymbol{w}}\boldsymbol{D}_{k,\tilde{\boldsymbol{w}}}^{-1}\tilde{\boldsymbol{W}}_k^\mathsf{T}\tilde{\boldsymbol{W}}_k\boldsymbol{D}_{k,\tilde{\boldsymbol{w}}}^{-1}\boldsymbol{D}_{k,\boldsymbol{w}}.$$

Then we have

$$\langle \mathbf{1}, \boldsymbol{A}_0^{-1}\boldsymbol{\mu}\rangle/\sqrt{d} \overset{\mathrm{d}}{=} \langle \mathbf{1}, \bar{\boldsymbol{A}}_0^{-1}\boldsymbol{\mu}\rangle/\sqrt{d}.\tag{47}$$

**Step 2. Bound the difference between $\bar{\boldsymbol{A}}_0$ and $\tilde{\boldsymbol{A}}_0$.**

Define

$$\tilde{\boldsymbol{A}}_0 = c_1\mathbf{I}_d + c_2\sum_{k\in[K]}\tilde{\boldsymbol{W}}_k^\mathsf{T}\tilde{\boldsymbol{W}}_k.$$

Since $d_k \to \infty$ as $d \to \infty$, we have

$$\|\boldsymbol{D}_{k,\tilde{\boldsymbol{w}}}^{-1} \boldsymbol{D}_{k,\boldsymbol{w}} - \mathbf{I}_N\|_{\mathrm{op}} = o_{d,\mathbb{P}}(1),$$

and hence

$$\|\tilde{\boldsymbol{A}}_0 - \bar{\boldsymbol{A}}_0\|_{\mathrm{op}} \leq 2c_2 \sum_{k \in [K]} \|\boldsymbol{D}_{k,\boldsymbol{w}} \boldsymbol{D}_{k,\tilde{\boldsymbol{w}}} - \mathbf{I}_d\|_{\mathrm{op}} \|\tilde{\boldsymbol{W}}_k^\mathsf{T} \tilde{\boldsymbol{W}}_k\|_{\mathrm{op}} \|\boldsymbol{D}_{k,\boldsymbol{w}} \boldsymbol{D}_{k,\tilde{\boldsymbol{w}}}\|_{\mathrm{op}} = o_{d,\mathbb{P}}(1).$$

By definition, $\tilde{\boldsymbol{A}}_0, \bar{\boldsymbol{A}}_0 \succeq c_1 \mathbf{I}$ and therefore $\|\tilde{\boldsymbol{A}}_0^{-1}\|_{\mathrm{op}}, \|\bar{\boldsymbol{A}}_0^{-1}\|_{\mathrm{op}} = O_{d,\mathbb{P}}(1)$. We deduce

$$\|\tilde{\boldsymbol{A}}_0^{-1} - \bar{\boldsymbol{A}}_0^{-1}\|_{\mathrm{op}} = \|\tilde{\boldsymbol{A}}_0^{-1}(\bar{\boldsymbol{A}}_0 - \tilde{\boldsymbol{A}}_0)\bar{\boldsymbol{A}}_0^{-1}\|_{\mathrm{op}} = o_{d,\mathbb{P}}(1).$$

This gives (recalling that $\|\boldsymbol{\mu}\|_2^2 = O_{d,\mathbb{P}}(1)$)

$$\langle \mathbf{1}, \bar{\boldsymbol{A}}_0^{-1} \boldsymbol{\mu} \rangle / \sqrt{d} - \langle \mathbf{1}, \tilde{\boldsymbol{A}}_0^{-1} \boldsymbol{\mu} \rangle / \sqrt{d} = o_{d,\mathbb{P}}(1). \tag{48}$$

**Step 3. Calculating the second moment of $\langle \mathbf{1}, \tilde{\boldsymbol{A}}_0^{-1} \boldsymbol{\mu} \rangle / \sqrt{d}$.**

Since we have
$$\mathbb{E}_{\boldsymbol{W}}[(\langle \mathbf{1}, \tilde{\boldsymbol{A}}_0^{-1} \boldsymbol{\mu} \rangle / \sqrt{d})^2] = \langle \mathbf{1}, \tilde{\boldsymbol{A}}_0^{-2} \mathbf{1} \rangle / d \cdot \mathbb{E}_{\boldsymbol{w} \sim \mathsf{N}(\mathbf{0},\boldsymbol{\Gamma})}[(\|\boldsymbol{w}\|_2^2 - 1)^2].$$

Note that
$$\mathbb{E}_{\boldsymbol{w} \sim \mathsf{N}(\mathbf{0},\boldsymbol{\Gamma})}[(\|\boldsymbol{w}\|_2^2 - 1)^2] = O_{d,\mathbb{P}}(1/d),$$

and using that $\|\tilde{\boldsymbol{A}}_0^{-1}\|_{\mathrm{op}} = O_{d,\mathbb{P}}(1)$,
$$\langle \mathbf{1}, \tilde{\boldsymbol{A}}_0^{-2} \mathbf{1} \rangle / d = O_{d,\mathbb{P}}(1).$$

Therefore
$$\mathbb{E}_{\boldsymbol{W}}[(\langle \mathbf{1}, \tilde{\boldsymbol{A}}_0^{-1} \boldsymbol{\mu} \rangle / \sqrt{d})^2] = o_{d,\mathbb{P}}(1).$$

By Chebyshev inequality we have
$$\langle \mathbf{1}, \tilde{\boldsymbol{A}}_0^{-1} \boldsymbol{\mu} \rangle / \sqrt{d} = o_{d,\mathbb{P}}(1). \tag{49}$$

Combining (47), (48) and (49) proves the lemma in the case of a covariance of the form (46):

$$\langle \mathbf{1}, \boldsymbol{A}_0^{-1} \boldsymbol{\mu} \rangle / \sqrt{d} = o_{d,\mathbb{P}}(1). \tag{50}$$

**Step 4. From discrete to continuous spectrum.**

We consider $\boldsymbol{\Gamma}$ a covariance matrix verifying assumption **A2**. For a given $\varepsilon > 0$ and $K$ sufficiently large, we consider $\boldsymbol{\Gamma}_\varepsilon$ a matrix obtained from $\boldsymbol{\Gamma}$ by binning its eigenvalues to at most $K$ points of $[0, C/d]$, such that we have $\mathrm{Tr}(\boldsymbol{\Gamma}_\varepsilon) = 1$ and $\lim_{d \to \infty} d \cdot \|\boldsymbol{\Gamma} - \boldsymbol{\Gamma}_\varepsilon\|_{\mathrm{op}} \leq \varepsilon$ (recall that $\|\boldsymbol{\Gamma}\|_{\mathrm{op}} \leq C/d$ by assumption). Such a matrix always exists from the condition $\mathrm{Tr}(\boldsymbol{\Gamma}) = 1$ and the weak convergence of the spectrum of $d \cdot \boldsymbol{\Gamma}$.

By construction $\boldsymbol{\Gamma}_\varepsilon$ is of the form (46). Consider $\boldsymbol{G} = (\boldsymbol{g}_1, \ldots, \boldsymbol{g}_N) \in \mathbb{R}^{d \times N}$ where $\boldsymbol{g}_i \sim_{i.i.d.} \mathsf{N}(\mathbf{0}, \mathbf{I}_d)$. We define:
$$\boldsymbol{\mu} = (\|\boldsymbol{\Gamma}^{1/2} \boldsymbol{g}_i\|_2^2 - 1)_{i \in [N]}, \qquad \boldsymbol{\mu}_\varepsilon = (\|\boldsymbol{\Gamma}_\varepsilon^{1/2} \boldsymbol{g}_i\|_2^2 - 1)_{i \in [N]},$$
$$\boldsymbol{A}_0 = c_1 \mathbf{I}_d + c_2 \boldsymbol{G}^\mathsf{T} \boldsymbol{\Gamma} \boldsymbol{G}, \qquad \boldsymbol{A}_{0,\varepsilon} = c_1 \mathbf{I}_d + c_2 \boldsymbol{G}^\mathsf{T} \boldsymbol{\Gamma}_\varepsilon \boldsymbol{G}.$$

We have for $d$ sufficiently large,

$$\|\boldsymbol{A}_0 - \boldsymbol{A}_{0,\varepsilon}\|_{\mathrm{op}} = \|\boldsymbol{G}^\mathsf{T}(\boldsymbol{\Gamma} - \boldsymbol{\Gamma}_\varepsilon)\boldsymbol{G}\|_{\mathrm{op}} \leq \|\boldsymbol{G}\|_{\mathrm{op}}^2 \|\boldsymbol{\Gamma} - \boldsymbol{\Gamma}_\varepsilon\|_{\mathrm{op}} \leq 2\varepsilon \|\boldsymbol{G}\|_{\mathrm{op}}^2 / d.$$

Furthermore, using $\mathrm{Tr}(\boldsymbol{\Gamma} - \boldsymbol{\Gamma}_\varepsilon) = 0$, we have

$$\mathbb{E}[\|\boldsymbol{\mu} - \boldsymbol{\mu}_\varepsilon\|_2^2] = N \mathbb{E}[(\langle \boldsymbol{g}_i \boldsymbol{g}_i^\mathsf{T}, \boldsymbol{\Gamma} - \boldsymbol{\Gamma}_\varepsilon \rangle)^2] = 2N \|\boldsymbol{\Gamma} - \boldsymbol{\Gamma}_\varepsilon\|_F^2 \leq 2\rho \varepsilon^2.$$

Therefore

$$\left|\langle \mathbf{1}, \boldsymbol{A}_0^{-1}\boldsymbol{\mu} - \boldsymbol{A}_{0,\varepsilon}^{-1}\boldsymbol{\mu}_\varepsilon\rangle/\sqrt{d}\right| \leq \left|\langle \mathbf{1}, \boldsymbol{A}_0^{-1}(\boldsymbol{A}_{0,\varepsilon} - \boldsymbol{A}_0)\boldsymbol{A}_{0,\varepsilon}^{-1}\boldsymbol{\mu}\rangle/\sqrt{d}\right| + \left|\langle \mathbf{1}, \boldsymbol{A}_{0,\varepsilon}^{-1}(\boldsymbol{\mu}_\varepsilon - \boldsymbol{\mu})\rangle/\sqrt{d}\right|$$
$$\leq \|\boldsymbol{A}_0^{-1}\|_{\mathrm{op}}\|\boldsymbol{A}_0 - \boldsymbol{A}_{0,\varepsilon}\|_{\mathrm{op}}\|\boldsymbol{A}_{0,\varepsilon}^{-1}\|_{\mathrm{op}}\|\boldsymbol{\mu}\|_2 + \|\boldsymbol{A}_{0,\varepsilon}^{-1}\|_{\mathrm{op}}\|\boldsymbol{\mu} - \boldsymbol{\mu}_\varepsilon\|_2.$$

Noticing that $\|\boldsymbol{A}_0^{-1}\|_{\mathrm{op}}, \|\boldsymbol{A}_{0,\varepsilon}^{-1}\|_{\mathrm{op}} \leq c_1^{-1}$, and using (50) applied to $\boldsymbol{\Gamma}_\varepsilon$, we get for $d$ sufficiently large:

$$\left|\langle \mathbf{1}, \boldsymbol{A}_0^{-1}\boldsymbol{\mu}\rangle/\sqrt{d}\right| \leq o_{d,\mathbb{P}}(1) + 2\varepsilon c_1^{-2}\|\boldsymbol{\mu}\|_2\|\boldsymbol{G}\|_{\mathrm{op}}^2/d + c_1^{-1}\|\boldsymbol{\mu} - \boldsymbol{\mu}_\varepsilon\|_2. \tag{51}$$

We have $\|\boldsymbol{\mu}\|_2\|\boldsymbol{G}\|_{\mathrm{op}}^2/d = O_{d,\mathbb{P}}(1)$ hence for any $\delta > 0$ there exists a constant $C_\delta$ (which do not depend on $\varepsilon$) such that:
$$\mathbb{P}(\varepsilon\|\boldsymbol{\mu}\|_2\|\boldsymbol{G}\|_{\mathrm{op}}^2/d > \varepsilon C_\delta) \leq \delta.$$

Taking a sequence $\delta \to 0$ and $\varepsilon$ such that $\varepsilon \propto C_\delta^{-1}$ shows that this is equivalent to

$$\varepsilon\|\boldsymbol{\mu}\|_2\|\boldsymbol{G}\|_{\mathrm{op}}^2/d = o_{d,\mathbb{P}}(1). \tag{52}$$

By Markov inequality,
$$\lim_{d\to\infty} \mathbb{P}(\|\boldsymbol{\mu} - \boldsymbol{\mu}_\varepsilon\|_2 \geq \varepsilon\sqrt{2\rho/\delta}) \leq \delta.$$

Taking $\varepsilon \propto \sqrt{\delta}$, we deduce that this is equivalent to

$$\|\boldsymbol{\mu} - \boldsymbol{\mu}_\varepsilon\|_2 = o_{d,\mathbb{P}}(1). \tag{53}$$

Substituting (52) and (53) in (51) concludes the proof. $\qquad\square$

**Lemma 5.** *Under the same setting as Proposition 2, we have*

$$\langle \mathbf{1}, \boldsymbol{U}_0^{-1}\mathbf{1}\rangle/d = \frac{\mathbf{1}^\top \boldsymbol{A}_0^{-1}\mathbf{1}/d}{1 + \kappa\mathbf{1}^\top \boldsymbol{A}_0^{-1}\mathbf{1}/d} + o_{d,\mathbb{P}}(1). \tag{54}$$

*Proof of Lemma 5.* Define $\boldsymbol{z} = \sqrt{\kappa}\mathbf{1}/\sqrt{d}$. Then we have

$$\boldsymbol{U}_0 = \boldsymbol{A}_0 + \boldsymbol{z}\boldsymbol{z}^\top + \boldsymbol{\mu}\boldsymbol{\mu}^\top.$$

By assumption, we have $\kappa = O_{d,\mathbb{P}}(1)$ and therefore $\|\boldsymbol{z}\|_2 = O_{d,\mathbb{P}}(1)$. We have already seen that $\|\boldsymbol{A}_0^{-1}\|_{\mathrm{op}}, \|\boldsymbol{A}_0^{-1}\|_{\mathrm{op}} = O_{d,\mathbb{P}}(1)$. Furthermore

$$\|\boldsymbol{A}_0\|_{\mathrm{op}} \leq \tilde{\lambda} + \lambda_1^2\lambda_{\max}(\boldsymbol{W}^\top\boldsymbol{W}) = O_{d,\mathbb{P}}(1).$$

By Sherman Morrison Woodbury formula, we have

$$\mathbf{1}^\top\boldsymbol{U}_0^{-1}\mathbf{1}/d = \mathbf{1}^\top\boldsymbol{A}_0^{-1}\mathbf{1}/d - \mathbf{1}^\top\boldsymbol{A}_0^{-1}[\boldsymbol{z},\boldsymbol{\mu}](\mathbf{I}_2 + [\boldsymbol{z},\boldsymbol{\mu}]^\top\boldsymbol{A}_0^{-1}[\boldsymbol{z},\boldsymbol{\mu}])^{-1}[\boldsymbol{z},\boldsymbol{\mu}]^\top\boldsymbol{A}_0^{-1}\mathbf{1}/d.$$

Note that by
$$\|(\mathbf{I}_2 + [\boldsymbol{z},\boldsymbol{\mu}]^\top\boldsymbol{A}_0^{-1}[\boldsymbol{z},\boldsymbol{\mu}])^{-1}\|_F = O_{d,\mathbb{P}}(1),$$
and by Lemma 4, we have (since $\boldsymbol{z}^\top\boldsymbol{A}_0^{-1}\boldsymbol{\mu}, \mathbf{1}^\top\boldsymbol{A}_0^{-1}\boldsymbol{\mu}/\sqrt{d} = o_{d,\mathbb{P}}(1)$)

$$\mathbf{1}^\top\boldsymbol{A}_0^{-1}[\boldsymbol{z},\boldsymbol{\mu}](\mathbf{I}_2 + [\boldsymbol{z},\boldsymbol{\mu}]^\top\boldsymbol{A}_0^{-1}[\boldsymbol{z},\boldsymbol{\mu}])^{-1}[\boldsymbol{z},\boldsymbol{\mu}]^\top\boldsymbol{A}_0^{-1}\mathbf{1}/d$$
$$= (\mathbf{1}^\top\boldsymbol{A}_0^{-1}\boldsymbol{z})^2(1 + \boldsymbol{z}^\top\boldsymbol{A}_0^{-1}\boldsymbol{z})^{-1}/d + o_{d,\mathbb{P}}(1) = \kappa(\mathbf{1}^\top\boldsymbol{A}_0^{-1}\mathbf{1}/d)^2(1 + \kappa\mathbf{1}^\top\boldsymbol{A}_0^{-1}\mathbf{1}/d)^{-1} + o_{d,\mathbb{P}}(1).$$

This proves the lemma. $\qquad\square$

In the following, we give an asymptotic expression for $\langle \mathbf{1}, \boldsymbol{A}_0^{-1}\mathbf{1}\rangle/d$.

**Lemma 6.** *Let* $(\boldsymbol{w}_i)_{i\in[N]} \sim \mathsf{N}(\mathbf{0},\boldsymbol{\Gamma})$ *independently, while* $\boldsymbol{\Gamma}$ *satisfies assumption* **A2** *(resp.* **B2***). Denote* $\boldsymbol{W} = (\boldsymbol{w}_1,\ldots,\boldsymbol{w}_N) \in \mathbb{R}^{d\times N}$. *Let* $\tilde{\lambda}$ *and* $\lambda_1$ *be two positive constants. Define*

$$\boldsymbol{A}_0 = \tilde{\lambda}\mathbf{I}_N + \lambda_1^2 \boldsymbol{W}^\top\boldsymbol{W}.$$

*Let* $\rho \in (0,\infty)$. *We have almost surely*

$$\lim_{N/d=\rho,d\to\infty} |\mathbf{1}^\top \boldsymbol{A}_0^{-1}\mathbf{1}/d - \mathrm{Tr}(\boldsymbol{A}_0^{-1})/d| = 0. \tag{55}$$

*In addition, assume* $\mathcal{D}$ *is the limiting spectral distribution of* $d \cdot \boldsymbol{\Gamma}$. *Then, we have almost surely*

$$\lim_{N/d=\rho,d\to\infty} \frac{1}{d}\mathrm{Tr}(\boldsymbol{A}_0^{-1}) = m_{\mathcal{D}}(-\tilde{\lambda}), \tag{56}$$

*where* $m_{\mathcal{D}}(\cdot) : \mathbb{C}^+ \to \mathbb{C}^+$ *is the companion Stieltjes transform associated with* $\mathcal{D}$. *For any* $x \in \mathbb{C}^+$, $m_{\mathcal{D}}(x)$ *satisfies the so called Silverstein's equation:*

$$x = -\frac{\rho}{m_{\mathcal{D}}(x)} + \int \frac{\lambda_1^2 t}{1 + \lambda_1^2 t m_{\mathcal{D}}(x)}\mathcal{D}(\mathrm{d}t). \tag{57}$$

*Proof of Lemma 6.*   Consider the event

$$A_N(t) := \{|\mathbf{1}^\top \boldsymbol{A}_0^{-1}\mathbf{1}/d - \mathrm{Tr}(\boldsymbol{A}_0^{-1})/d| > t\}.$$

Let $\boldsymbol{Q} \in \mathbb{R}^{N\times N}$ be an orthogonal matrix. By rotation invariance of Gaussian random variables, $\boldsymbol{Q}\boldsymbol{W}^\top$ has the same distribution as $\boldsymbol{W}$. In fact, by Fubini's theorem, we can draw $\boldsymbol{Q}$ uniformly (independent of $\boldsymbol{A}_0$) from orthogonal matrices and the distribution would still be unchanged. Let

$$\tilde{A}_N(t) := \{|\mathbf{1}^\top (\boldsymbol{Q}\boldsymbol{A}_0^{-1}\boldsymbol{Q}^\top)^{-1}\mathbf{1}/d - \mathrm{Tr}(\boldsymbol{Q}\boldsymbol{A}_0^{-1}\boldsymbol{Q}^\top)/d| > t\}.$$

By the argument above,

$$\mathbb{P}[A_N(t)] = \mathbb{P}[\tilde{A}_N(t)].$$

Since $\boldsymbol{Q}$ is orthogonal, $\tilde{A}_N(t)$ can be written as

$$\{|\mathbf{1}^\top \boldsymbol{Q}\boldsymbol{A}_0^{-1}\boldsymbol{Q}^\top\mathbf{1}/d - \mathrm{Tr}(\boldsymbol{A}_0^{-1})/d| > t\}. \tag{58}$$

Since $\boldsymbol{Q}$ is a uniformly chosen orthogonal matrix, $\boldsymbol{Q}^\top\mathbf{1}/\sqrt{d}$ is uniformly distributed on $\mathbb{S}^{N-1}(\sqrt{\rho})$, independently of $\boldsymbol{A}_0$. Hence $\boldsymbol{Q}^\top\mathbf{1}/\sqrt{d}$ has the same distribution as $\sqrt{\rho}\boldsymbol{z}/\|\boldsymbol{z}\|_2$ where $\boldsymbol{z} \sim \mathsf{N}(0,\mathbf{I}_N)$. In particular,

$$\mathbb{P}[\tilde{A}_N(t)] = \mathbb{P}\Big\{\Big|\frac{1}{\|\boldsymbol{z}\|_2^2}\boldsymbol{z}^\top \boldsymbol{A}_0^{-1}\boldsymbol{z} - \mathrm{Tr}(\boldsymbol{A}_0^{-1})/N\Big| > \frac{t}{\rho}\Big\} \tag{59}$$

$$\le \mathbb{P}\Big\{\Big|\frac{N}{\|\boldsymbol{z}\|_2^2} - 1\Big|\boldsymbol{z}^\top \boldsymbol{A}_0^{-1}\boldsymbol{z}/N + |\boldsymbol{z}^\top \boldsymbol{A}_0^{-1}\boldsymbol{z}/N - \mathrm{Tr}(\boldsymbol{A}_0^{-1})/N| > \frac{t}{\rho}\Big\} \tag{60}$$

$$\le P_1 + P_2, \tag{61}$$

where

$$P_1 = \mathbb{P}\Big\{\Big|\frac{N}{\|\boldsymbol{z}\|^2} - 1\Big|\boldsymbol{z}^\top \boldsymbol{A}_0^{-1}\boldsymbol{z}/N > \frac{t}{2\rho}\Big\}, \qquad P_2 = \mathbb{P}\Big\{|\boldsymbol{z}^\top \boldsymbol{A}_0^{-1}\boldsymbol{z}/N - \mathrm{Tr}(\boldsymbol{A}_0^{-1})/N| > \frac{t}{2\rho}\Big\}.$$

Let's consider $P_1$ first. Since $\boldsymbol{A}_0^{-1} \preceq \mathbf{I}/\tilde{\lambda}$, we have

$$\frac{\boldsymbol{z}^\top \boldsymbol{A}_0^{-1}\boldsymbol{z}}{N} \le \frac{1}{\tilde{\lambda}}\frac{\|\boldsymbol{z}\|^2}{N},$$

which yields

$$P_1 \leq \mathbb{P}\left\{\left|\frac{N}{\|\boldsymbol{z}\|^2} - 1\right|\frac{\|\boldsymbol{z}\|^2}{N} > \frac{\tilde{\lambda}t}{2\rho}\right\} = \mathbb{P}\left\{\left|\frac{\|\boldsymbol{z}\|^2}{N} - 1\right| > \frac{\tilde{\lambda}t}{2\rho}\right\}. \tag{62}$$

We know due to fast concentration of $\|z\|^2/N$ around one (see e.g. [BLM13]), $P_1$ vanish exponentially fast in $N$ (equivalently in $d$ since $N/d$ is fixed to be $\rho$).

Now, let's consider $P_2$. $|\boldsymbol{z}^{\mathsf{T}}\boldsymbol{A}_0^{-1}\boldsymbol{z}/N - \text{Tr}(\boldsymbol{A}_0^{-1})/N|$. By Hanson-Wright inequality (see e.g. [BLM13]), we have

$$\mathbb{P}\left(|\boldsymbol{z}^{\mathsf{T}}\boldsymbol{A}_0^{-1}\boldsymbol{z}/N - \text{Tr}(\boldsymbol{A}_0^{-1})/N| > \frac{t}{2\rho}\Big|\boldsymbol{A}_0\right) \leq 2\exp\left\{-c\min\left(\frac{t^2}{\|\boldsymbol{A}_0^{-1}/N\|_F^2}, \frac{t}{\|\boldsymbol{A}_0^{-1}/N\|_{\text{op}}}\right)\right\} \tag{63}$$

$$\leq 2\exp\left\{-c'\min\left(N\tilde{\lambda}^2t^2, \tilde{\lambda}tN\right)\right\}. \tag{64}$$

Since the bound in (64) is independent of $\boldsymbol{A}_0$, it holds unconditionally. Therefore, we conclude $P_2$ vanishes exponentially fast in $N$ and $d$. We conclude that $\Pr[\tilde{A}_N(t)]$ vanishes exponentially fast as $d, N \to \infty$. Therefore, by Borel-Cantelli lemma we recover (55).

Convergence of $\text{Tr}(\boldsymbol{A}_0^{-1})/d$ to $m_D(-\tilde{\lambda})$ is a standard result in random matrix theory. We refer the reader to [BS10] Chapters 3 and 6. □

### B.1.5 Proof of Theorem 1

By Lemma 1, the risk has a representation

$$R_{\text{RF},N}(f_*) = \mathbb{E}_{\boldsymbol{x}\sim\text{N}(\boldsymbol{0},\mathbf{I}_d)}[f_*(\boldsymbol{x})^2] - \boldsymbol{V}^{\mathsf{T}}\boldsymbol{U}^{-1}\boldsymbol{V}.$$

By Lemma 2, we have

$$\|\boldsymbol{U} - \boldsymbol{U}_0\|_{\text{op}} = o_{d,\mathbb{P}}(1).$$

By Lemma 3, we have

$$\|\boldsymbol{V} - \tau\boldsymbol{1}/\sqrt{d}\|_2 = \|\boldsymbol{B}\|_F \cdot o_{d,\mathbb{P}}(1),$$

where

$$\tau = \sqrt{d} \cdot \lambda_2 \text{Tr}(\boldsymbol{B}\boldsymbol{\Gamma}).$$

Hence, we have

$$|\boldsymbol{V}^{\mathsf{T}}\boldsymbol{U}^{-1}\boldsymbol{V} - \tau^2\boldsymbol{1}^{\mathsf{T}}\boldsymbol{U}_0^{-1}\boldsymbol{1}/d| = \|\boldsymbol{B}\|_F^2 \cdot o_{d,\mathbb{P}}(1).$$

Proposition 2 gives the expression for

$$\boldsymbol{1}^{\mathsf{T}}\boldsymbol{U}_0^{-1}\boldsymbol{1}/d = \psi/(1 + \kappa\psi) + o_{d,\mathbb{P}}(1),$$

where

$$\kappa = d \cdot \lambda_2^2 \text{Tr}(\boldsymbol{\Gamma}^2)/2.$$

Hence we have

$$\boldsymbol{V}^{\mathsf{T}}\boldsymbol{U}\boldsymbol{V} = \tau^2\psi/(1 + \kappa\psi) + \|\boldsymbol{B}\|_F^2 \cdot o_{d,\mathbb{P}}(1).$$

Recalling the assumption $\mathbb{E}(f_*) = 0$, we have $\|f_*\|_{L^2}^2 = 2\|\boldsymbol{B}\|_F^2$, which concludes the proof.

## B.2 Neural Tangent model: proof of Theorem 2

Recall the definition

$$R_{\text{NT},N}(f_*) = \min_{\hat{f}\in\mathcal{F}_{\text{NT},N}(\boldsymbol{W})} \mathbb{E}\left\{(f_*(\boldsymbol{x}) - \hat{f}(\boldsymbol{x}))^2\right\},$$

where

$$\mathcal{F}_{\text{NT},N}(\boldsymbol{W}) = \left\{f_N(\boldsymbol{x}) = c + \sum_{i=1}^{N}\sigma'(\langle\boldsymbol{w}_i, \boldsymbol{x}\rangle)\langle\boldsymbol{a}_i, \boldsymbol{x}\rangle : c \in \mathbb{R}, \boldsymbol{a}_i \in \mathbb{R}^d, i \in [N]\right\}.$$

*Proof of Theorem 2.* We can rewrite the neural tangent model with a squared non-linearity $\sigma(x) = x^2$ as

$$\hat{f}(\boldsymbol{W}, \boldsymbol{A}, c) = 2\sum_{i=1}^{N}\langle\boldsymbol{w}_i, \boldsymbol{x}\rangle\langle\boldsymbol{a}_i, \boldsymbol{x}\rangle + c = 2\langle\boldsymbol{W}\boldsymbol{A}^\mathsf{T}, \boldsymbol{x}\boldsymbol{x}^\mathsf{T}\rangle + c.$$

with $\boldsymbol{W} = [\boldsymbol{w}_1, \ldots, \boldsymbol{w}_N] \in \mathbb{R}^{d \times N}$ and $\boldsymbol{A} = [\boldsymbol{a}_1, \ldots, \boldsymbol{a}_N] \in \mathbb{R}^{d \times N}$. Note that we have

$$\mathbb{E}_{\boldsymbol{x}}[\langle\boldsymbol{B} - 2\boldsymbol{W}\boldsymbol{A}^\mathsf{T}, \boldsymbol{x}\boldsymbol{x}^\mathsf{T}\rangle + b_0 - c)^2]$$
$$= 2\|\boldsymbol{B} - \boldsymbol{W}\boldsymbol{A}^\mathsf{T} - \boldsymbol{A}\boldsymbol{W}^\mathsf{T}\|_F^2 + \text{Tr}(\boldsymbol{B} - 2\boldsymbol{W}\boldsymbol{A}^\mathsf{T})^2 - 2\text{Tr}(\boldsymbol{B} - 2\boldsymbol{W}\boldsymbol{A}^\mathsf{T})(c - b_0) + (c - b_0)^2,$$

which, after minimizing over $c \in \mathbb{R}$, simplifies to:

$$\min_{c \in \mathbb{R}}\|f_* - \hat{f}(\boldsymbol{W}, \boldsymbol{A}, c)\|_{L^2}^2 = 2\|\boldsymbol{B} - \boldsymbol{W}\boldsymbol{A}^\mathsf{T} - \boldsymbol{A}\boldsymbol{W}^\mathsf{T}\|_F^2.$$

For $\boldsymbol{w}_i \sim \mathsf{N}(\boldsymbol{0}, \mathbf{I}_d)$, we have $\text{rank}(\boldsymbol{W}) = \min(d, N) \equiv r$ with probability one. Let $\boldsymbol{W} = \boldsymbol{P}_1\boldsymbol{S}\boldsymbol{V}^\mathsf{T}$ be the singular value decomposition of $\boldsymbol{W}$, with $\boldsymbol{P}_1 \in \mathbb{R}^{d \times r}$, $\boldsymbol{S} \in \mathbb{R}^{r \times r}$ and $\boldsymbol{V} \in \mathbb{R}^{N \times r}$. Defining $\boldsymbol{G} = \boldsymbol{S}\boldsymbol{V}^\mathsf{T}\boldsymbol{A} \in \mathbb{R}^{r \times d}$, we get almost surely

$$\min_{\boldsymbol{A} \in \mathbb{R}^{d \times N}, c \in \mathbb{R}}\|f_* - \hat{f}(\boldsymbol{W}, \boldsymbol{A}, c)\|_{L^2}^2 = \min_{\boldsymbol{G} \in \mathbb{R}^{r \times d}} 2\|\boldsymbol{B} - \boldsymbol{P}_1\boldsymbol{G} - \boldsymbol{G}^\mathsf{T}\boldsymbol{P}_1^\mathsf{T}\|_F^2.$$

In the case $N \geq d$, we can take $\boldsymbol{G} = \boldsymbol{P}_1^\mathsf{T}\boldsymbol{B}/2$ and we get almost surely over $\boldsymbol{W} \in \mathbb{R}^{d \times N}$

$$R_{\mathsf{NT}, N}(f_*) = 0.$$

Consider the case when $N < d$, we define $\boldsymbol{P}_2 \in \mathbb{R}^{d \times (d-N)}$ the completion of $\boldsymbol{P}_1$ to a full basis $\boldsymbol{P} = [\boldsymbol{P}_1, \boldsymbol{P}_2] \in \mathbb{R}^{d \times d}$. We define $\boldsymbol{G}_1 = \boldsymbol{G}\boldsymbol{P}_1 \in \mathbb{R}^{N \times N}$ and $\boldsymbol{G}_2 = \boldsymbol{G}\boldsymbol{P}_2 \in \mathbb{R}^{N \times (d-N)}$ and we perform our computation in the $\boldsymbol{P}$ basis. We have

$$\boldsymbol{B} - \boldsymbol{P}_1\boldsymbol{G} - \boldsymbol{G}^\mathsf{T}\boldsymbol{P}_1^\mathsf{T} = \begin{pmatrix} \boldsymbol{B}_{11} - \boldsymbol{G}_1 - \boldsymbol{G}_1^\mathsf{T} & \boldsymbol{B}_{12} - \boldsymbol{G}_2 \\ \boldsymbol{B}_{21} - \boldsymbol{G}_2^\mathsf{T} & \boldsymbol{B}_{22} \end{pmatrix},$$

where $\boldsymbol{B}_{ij} = \boldsymbol{P}_i^\mathsf{T}\boldsymbol{B}\boldsymbol{P}_j$ for $i, j = 1, 2$. We readily deduce that

$$\min_{\boldsymbol{G} \in \mathbb{R}^{r \times d}} 2\|\boldsymbol{B} - \boldsymbol{P}_1\boldsymbol{G} - \boldsymbol{G}^\mathsf{T}\boldsymbol{P}_1^\mathsf{T}\|_F^2 = 2\|\boldsymbol{P}_2^\mathsf{T}\boldsymbol{B}\boldsymbol{P}_2\|_F^2.$$

Let us compute its expectation over $\boldsymbol{w}_i \sim \mathsf{N}(\boldsymbol{0}, \mathbf{I}_d)$, i.e over $\boldsymbol{P}_2 = [\boldsymbol{v}_1, \ldots, \boldsymbol{v}_{d-N}]$ where the $\boldsymbol{v}_i \in \mathbb{R}^d$ are $(d - N)$ orthogonal vectors uniformly distributed on the unit sphere in $\mathbb{R}^d$. Let $\boldsymbol{B} = \sum_{i=1}^{s}\lambda_i\boldsymbol{e}_i\boldsymbol{e}_i^\mathsf{T}$ with $\boldsymbol{e}_i$ the orthonormal eigenvectors of $\boldsymbol{B}$. We get:

$$\begin{aligned}\mathbb{E}[\|\boldsymbol{P}_2^\mathsf{T}\boldsymbol{B}\boldsymbol{P}_2\|_F^2] &= \sum_{i,j=1}^{s}\sum_{k,l=1}^{d-N}\lambda_i\lambda_j\mathbb{E}[\langle\boldsymbol{v}_k, \boldsymbol{e}_i\rangle\langle\boldsymbol{v}_k, \boldsymbol{e}_j\rangle\langle\boldsymbol{v}_l, \boldsymbol{e}_i\rangle\langle\boldsymbol{v}_l, \boldsymbol{e}_j\rangle] \\ &= \|\boldsymbol{B}\|_F^2(d-N)\mathbb{E}[\langle\boldsymbol{v}_1, \boldsymbol{e}_1\rangle^4] + \|\boldsymbol{B}\|_F^2(d-N)(d-N-1)\mathbb{E}[\langle\boldsymbol{v}_1, \boldsymbol{e}_1\rangle^2\langle\boldsymbol{v}_2, \boldsymbol{e}_1\rangle^2] \\ &\quad + 2\Big(\sum_{i<j}\lambda_i\lambda_j\Big)(d-N)\mathbb{E}[\langle\boldsymbol{v}_1, \boldsymbol{e}_1\rangle^2\langle\boldsymbol{v}_1, \boldsymbol{e}_2\rangle^2] \\ &\quad + 2\Big(\sum_{i<j}\lambda_i\lambda_j\Big)(d-N)(d-N-1)\mathbb{E}[\langle\boldsymbol{v}_1, \boldsymbol{e}_1\rangle\langle\boldsymbol{v}_1, \boldsymbol{e}_2\rangle\langle\boldsymbol{v}_2, \boldsymbol{e}_1\rangle\langle\boldsymbol{v}_2, \boldsymbol{e}_2\rangle]\Big].\end{aligned} \tag{65}$$

We bound each term separately. For $\boldsymbol{u} \sim \text{Unif}(\mathbb{S}^{d-1})$, we have the convergence in distribution of the first two coordinates $\sqrt{d}(u_1, u_2) \Rightarrow \mathsf{N}(\boldsymbol{0}, \mathbf{I}_2)$, hence:

$$\lim_{d\to\infty} d^2\mathbb{E}[\langle\boldsymbol{v}_1, \boldsymbol{e}_1\rangle^4] = 3, \qquad \lim_{d\to\infty} d^2\mathbb{E}[\langle\boldsymbol{v}_1, \boldsymbol{e}_1\rangle^2\langle\boldsymbol{v}_1, \boldsymbol{e}_2\rangle^2] = 1. \tag{66}$$

Furthermore, conditioned on $\boldsymbol{v}_1$, $\boldsymbol{v}_2$ is uniformly distributed over the sphere $\mathbb{S}^{d-2}$ in the hyperplane orthogonal to $\boldsymbol{v}_1$. We get the uniform convergence

$$\lim_{d\to\infty}\sup_{\boldsymbol{v}_1\in\mathbb{S}^{d-1}}|d\mathbb{E}[\langle\boldsymbol{v}_2,\boldsymbol{e}_1\rangle^2|\boldsymbol{v}_1]-(1-\langle\boldsymbol{v}_1,\boldsymbol{e}_1\rangle^2)|=0.$$

By dominated convergence theorem, we get

$$\lim_{d\to\infty}d^2\mathbb{E}[\langle\boldsymbol{v}_1,\boldsymbol{e}_1\rangle^2\langle\boldsymbol{v}_2,\boldsymbol{e}_1\rangle^2]=1. \tag{67}$$

The last term of the sum (65) is also derived by first conditioning on $\boldsymbol{v}_1$. Let us denote $\boldsymbol{z}_1=\boldsymbol{P}_{\perp\boldsymbol{v}_1}\boldsymbol{e}_1$ and $\boldsymbol{z}_2=\boldsymbol{P}_{\perp\boldsymbol{v}_1}\boldsymbol{e}_2$ the projections of $(\boldsymbol{e}_1,\boldsymbol{e}_2)$ on the hyperplane perpendicular to $\boldsymbol{v}_1$, on which $\boldsymbol{v}_2$ is uniformly distributed over the unit sphere. We decompose $\boldsymbol{z}_2$ into two components: one along $\boldsymbol{z}_1$ that we denote $\boldsymbol{z}_2^{(1)}=\boldsymbol{P}_{\|\boldsymbol{z}_1}\boldsymbol{z}_2$ and one perpendicular to $\boldsymbol{z}_1$, denoted $\boldsymbol{z}_2^{(2)}=\boldsymbol{P}_{\perp\boldsymbol{z}_1}\boldsymbol{z}_2$. Then we have:

$$\begin{aligned}\mathbb{E}[\langle\boldsymbol{v}_2,\boldsymbol{e}_1\rangle\langle\boldsymbol{v}_2,\boldsymbol{e}_2\rangle|\boldsymbol{v}_1]&=\mathbb{E}[\langle\boldsymbol{v}_2,\boldsymbol{z}_1\rangle\langle\boldsymbol{v}_2,\boldsymbol{z}_2\rangle|\boldsymbol{v}_1]\\&=\mathbb{E}\Big[\langle\boldsymbol{v}_2,\boldsymbol{z}_1\rangle\Big(\langle\boldsymbol{v}_2,\boldsymbol{z}_2^{(1)}\rangle+\langle\boldsymbol{v}_2,\boldsymbol{z}_2^{(2)}\rangle\Big)\Big|\boldsymbol{v}_i\Big]\\&=\langle\boldsymbol{z}_1,\boldsymbol{z}_2\rangle\mathbb{E}[u_1^2]+\|\boldsymbol{z}_1\|_2\|\boldsymbol{z}_2^{(2)}\|_2\mathbb{E}[u_1u_2]\\&=\frac{\langle\boldsymbol{z}_1,\boldsymbol{z}_2\rangle}{d-1},\end{aligned}$$

where $(u_1,u_2)$ are the first two coordinates of a uniform random variable on the sphere $\mathbb{S}^{d-2}$. Using that:

$$\langle\boldsymbol{z}_1,\boldsymbol{z}_2\rangle=\langle\boldsymbol{e}_1-\langle\boldsymbol{e}_1,\boldsymbol{v}_1\rangle\boldsymbol{v}_1,\boldsymbol{e}_2-\langle\boldsymbol{e}_2,\boldsymbol{v}_1\rangle\boldsymbol{v}_1\rangle=-\langle\boldsymbol{e}_1,\boldsymbol{v}_1\rangle\langle\boldsymbol{e}_2,\boldsymbol{v}_1\rangle,$$

we get

$$\mathbb{E}[\langle\boldsymbol{v}_1,\boldsymbol{e}_1\rangle\langle\boldsymbol{v}_1,\boldsymbol{e}_2\rangle\langle\boldsymbol{v}_2,\boldsymbol{e}_1\rangle\langle\boldsymbol{v}_2,\boldsymbol{e}_2\rangle]=-\frac{1}{d-1}\mathbb{E}[\langle\boldsymbol{v}_1,\boldsymbol{e}_1\rangle^2\langle\boldsymbol{v}_1,\boldsymbol{e}_2\rangle^2]=-\frac{1}{d^3}+o_d(d^{-3}), \tag{68}$$

where we used the same argument as for (66). Plugging the above limits (66), (67) and (68) in the expansion (65), we get

$$\mathbb{E}[R_{\mathsf{NT},N}(f_*)]=2\|\boldsymbol{B}\|_F^2\Big[(1-\rho)_+^2+(1-\rho)_+\frac{\mathrm{Tr}(\boldsymbol{B})^2}{d\|\boldsymbol{B}\|_F^2}-(1-\rho)_+^2\frac{\mathrm{Tr}(\boldsymbol{B})^2}{d\|\boldsymbol{B}\|_F^2}+o_d(1)\Big]. \tag{69}$$

Recalling the assumption $\mathbb{E}(f_*)=0$, we have $\|f_*\|_{L^2}^2=2\|\boldsymbol{B}\|_F^2$, which concludes the proof. $\qquad\square$

**Remark 1.** *The above formula for the* RF *risk Eq.* (69) *has two terms that corresponds to the two limits* $\mathrm{Tr}(\boldsymbol{B})/\|\boldsymbol{B}\|_F=o_d(\sqrt{d})$ *(e.g. spiked matrix)*

$$\mathbb{E}[R_{\mathsf{NT},N}(f_*)]=2(1-\rho)_+^2\|\boldsymbol{B}\|_F^2+o_d(\|\boldsymbol{B}\|_F^2),$$

*and* $\mathrm{Tr}(\boldsymbol{B})^2=d\|\boldsymbol{B}\|_F^2$ *(i.e.* $\boldsymbol{B}\propto\mathbf{I}$)

$$\mathbb{E}[R_{\mathsf{NT},N}(f_*)]=2(1-\rho)_+\|\boldsymbol{B}\|_F^2.$$

It is possible to show concentration of $\|\boldsymbol{P}_2^\mathsf{T}\boldsymbol{B}\boldsymbol{P}_2\|_F^2$ on its mean $\mathbb{E}[\|\boldsymbol{P}_2^\mathsf{T}\boldsymbol{B}\boldsymbol{P}_2\|_F^2]$ for $\boldsymbol{B}$ that satisfies $\|\boldsymbol{B}\|_{\mathrm{op}}\|\boldsymbol{B}\|_F\leq C$ (see Theorem 5).

## B.3   Neural Network model: proof of Theorem 3

We consider two-layers neural networks with quadratic activation function $\sigma(x)=x^2$ and we fix the second layer weights to 1,

$$\hat{f}(\boldsymbol{x};\boldsymbol{W},c)=\sum_{i=1}^N\langle\boldsymbol{w}_i,\boldsymbol{x}\rangle^2+c.$$

We consider the ground truth function $f_*$ to be a quadratic function as per Eq. (20), and the risk function defined by

$$L(\boldsymbol{W}, c) = \mathbb{E}_{\boldsymbol{x}}[(f_*(\boldsymbol{x}) - \hat{f}(\boldsymbol{x}; \boldsymbol{W}, c))^2] = \mathbb{E}_{\boldsymbol{x}}\left[\left(\langle \boldsymbol{x}\boldsymbol{x}^\mathsf{T}, \boldsymbol{B} - \boldsymbol{W}\boldsymbol{W}^\mathsf{T}\rangle + b_0 - c\right)^2\right].$$

We consider running SGD dynamics upon the risk function for a fresh sample $(\boldsymbol{x}_k, f_*(\boldsymbol{x}_k))$ for each iteration

$$(\boldsymbol{W}_{k+1}, c_{k+1}) = (\boldsymbol{W}_k, c_k) - \varepsilon \nabla_{\boldsymbol{W}, c}\left(f_*(\boldsymbol{x}_k) - \hat{f}(\boldsymbol{x}_k; \boldsymbol{W}, c)\right)^2,$$

and denote

$$R_{\mathsf{NN}, N}(f_*; \ell, \varepsilon) = \mathbb{E}_{\boldsymbol{x}}[(f_*(\boldsymbol{x}) - \hat{f}(\boldsymbol{x}; \boldsymbol{W}_\ell, c_\ell))^2].$$

### B.3.1 Global minimum

**Lemma 7.** *Let $f_* = \langle \boldsymbol{x}, \boldsymbol{B}\boldsymbol{x}\rangle + b_0$ for some $\boldsymbol{B} \succeq 0$ and $b_0 \in \mathbb{R}$. Denote by $(\lambda_i(\boldsymbol{B}))_{i \in [r]}$ the positive eigenvalues of $\boldsymbol{B}$ in descending order. Then we have*

$$\inf_{\boldsymbol{W}, c} L(\boldsymbol{W}, c) = 2\sum_{i=N+1}^{r} \lambda_i(\boldsymbol{B})^2.$$

*Proof of Lemma 7.* Note we have

$$\begin{aligned}
L(\boldsymbol{W}, c) &= \mathbb{E}_{\boldsymbol{x}}[(\langle \boldsymbol{B} - \boldsymbol{W}\boldsymbol{W}^\mathsf{T}, \boldsymbol{x}\boldsymbol{x}^\mathsf{T}\rangle + b_0 - c)^2] \\
&= 2\|\boldsymbol{B} - \boldsymbol{W}\boldsymbol{W}^\mathsf{T}\|_F^2 + \mathrm{Tr}(\boldsymbol{B} - \boldsymbol{W}\boldsymbol{W}^\mathsf{T})^2 - 2\mathrm{Tr}(\boldsymbol{B} - \boldsymbol{W}\boldsymbol{W}^\mathsf{T})(c - b_0) + (c - b_0)^2,
\end{aligned}$$

minimizing over $c$ gives

$$\inf_c L(\boldsymbol{W}, c) = 2\|\boldsymbol{B} - \boldsymbol{W}\boldsymbol{W}^\mathsf{T}\|_F^2.$$

The infimum of $L$ over $\boldsymbol{W}$ is equivalent to the low-rank approximation problem of matrix $\boldsymbol{B}$ in Frobenius norm, with rank less or equal to $\max(d, N)$, and is given by the Eckart-Young-Mirsky theorem (see [EY36]). $\square$

### B.3.2 Landscape: proof of Proposition 1

Without loss of generality, throughout the proof, we assume that $\boldsymbol{B}$ is diagonal and $b_0 = 0$. Our first proposition characterizes the critical points of $L(\boldsymbol{W}, c)$.

**Proposition 3.** *Let $\boldsymbol{W} \in \mathbb{R}^{d \times N}$, and $\boldsymbol{B} \in \mathbb{R}^{d \times d}$ to be a positive semi-definite diagonal matrix. Define the risk function to be*

$$L(\boldsymbol{W}, c) = \mathbb{E}_{\boldsymbol{x}}[(\langle \boldsymbol{B} - \boldsymbol{W}\boldsymbol{W}^\mathsf{T}, \boldsymbol{x}\boldsymbol{x}^\mathsf{T}\rangle - c)^2].$$

*Then for any critical point $(\boldsymbol{W}_0, c_0)$ of $L(\boldsymbol{W}, c)$, there exists a projection matrix $\boldsymbol{P} = \sum_{i=1}^{k} \boldsymbol{e}_{\tau(i)}\boldsymbol{e}_{\tau(i)}^\mathsf{T}$ for some injection $\tau : [k] \to [d]$, such that $\boldsymbol{\Gamma}_0 = \boldsymbol{W}_0\boldsymbol{W}_0^\mathsf{T}$ is diagonal and satisfy*

$$\begin{aligned}
\boldsymbol{\Gamma}_0 &= \boldsymbol{P}\boldsymbol{B}\boldsymbol{P}, \\
c_0 &= \mathrm{Tr}(\boldsymbol{B} - \boldsymbol{\Gamma}_0).
\end{aligned}$$

*Proof.* Calculating the risk function, we get

$$L(\boldsymbol{W}, c) = c^2 + 2c \cdot \mathrm{Tr}(\boldsymbol{W}\boldsymbol{W}^\mathsf{T} - \boldsymbol{B}) + \mathrm{Tr}(\boldsymbol{W}\boldsymbol{W}^\mathsf{T} - \boldsymbol{B})^2 + 2\|\boldsymbol{W}\boldsymbol{W}^\mathsf{T} - \boldsymbol{B}\|_F^2.$$

We consider the gradient of this function. We get:

$$\begin{aligned}
\frac{\partial}{\partial c} L(\boldsymbol{W}, c) &= 2c + 2\mathrm{Tr}(\boldsymbol{W}\boldsymbol{W}^\mathsf{T} - \boldsymbol{B}), \\
\nabla_{\boldsymbol{W}} L(\boldsymbol{W}, c) &= 2c\boldsymbol{W} + 2\mathrm{Tr}(\boldsymbol{W}\boldsymbol{W}^\mathsf{T} - \boldsymbol{B})\boldsymbol{W} + 8(\boldsymbol{W}\boldsymbol{W}^\mathsf{T} - \boldsymbol{B})\boldsymbol{W}.
\end{aligned}$$

By the stationary condition, at a critical point $(\boldsymbol{W}_0, c_0)$, we must have:

$$c_0 = -\mathrm{Tr}(\boldsymbol{W}_0 \boldsymbol{W}_0^\mathsf{T} - \boldsymbol{B}), \tag{70}$$

$$\boldsymbol{B}\boldsymbol{W}_0 = \boldsymbol{W}_0 \boldsymbol{W}_0^\mathsf{T} \boldsymbol{W}_0. \tag{71}$$

Let us denote $\boldsymbol{W}_0 = \boldsymbol{U}\boldsymbol{S}\boldsymbol{V}^\mathsf{T}$ the (extended) singular value decomposition of $\boldsymbol{W}_0 \in \mathbb{R}^{d \times N}$ with $\boldsymbol{U} \in \mathbb{R}^{d \times d}$, $\boldsymbol{S} \in \mathbb{R}^{d \times N}$ and $\boldsymbol{V} \in \mathbb{R}^{N \times N}$. Then the stationary condition (71) gives

$$\boldsymbol{B}\boldsymbol{U}\boldsymbol{S}\boldsymbol{V}^\mathsf{T} = \boldsymbol{U}\boldsymbol{S}^3\boldsymbol{V}^\mathsf{T}. \tag{72}$$

Let $r$ be the rank of $\boldsymbol{W}_0$ and $\boldsymbol{S} = \mathrm{diag}(\boldsymbol{S}_1, \boldsymbol{0})$, $\boldsymbol{U} = (\boldsymbol{U}_1, \boldsymbol{U}_2)$ with $\boldsymbol{S}_1 \in \mathbb{R}^{r \times r}$, $\boldsymbol{U}_1 \in \mathbb{R}^{d \times r}$ and $\boldsymbol{U}_2 \in \mathbb{R}^{d \times (d-r)}$. Then we get:

$$\boldsymbol{B}\boldsymbol{U}_1 = \boldsymbol{U}_1 \boldsymbol{S}_1^2.$$

This is of the form of the eigenvalue equation of matrix $\boldsymbol{B}$. Hence we must have the columns of $\boldsymbol{U}_1$ to be a set of eigenvectors and $\boldsymbol{S}_1^2$ to be positive eigenvalues of $\boldsymbol{B}$. This proves the proposition. □

Note the global minimizers are attained for $\boldsymbol{\Gamma}_0 = \boldsymbol{W}_0 \boldsymbol{W}_0^\mathsf{T}$ corresponding to the $\min(N, d)$ directions of $\boldsymbol{B}$ with the largest eigenvalues. We prove in the following proposition that stationary points that are not global minimizers are strict saddle points.

Define the spectral separation of $\boldsymbol{B}$ as

$$\delta^{\mathrm{sep}} = \min\{|\lambda_i(\boldsymbol{B}) - \lambda_j(\boldsymbol{B})| \ : \ i, j \in [d], \lambda_i(\boldsymbol{B}) \neq \lambda_j(\boldsymbol{B})\},$$

and $\delta^{\mathrm{eig}}$ the minimum strictly positive eigenvalue of $\boldsymbol{B}$.

**Proposition 4.** *Consider $(\boldsymbol{W}_0, c_0)$ a stationary point of $L(\boldsymbol{W}, c)$ but not a global minimizer. Then, we have*

$$\lambda_{\min}(\nabla^2_{\boldsymbol{W}} L(\boldsymbol{W}_0, c_0)) \leq -4 \min\{\delta^{\mathrm{eig}}, \delta^{\mathrm{sep}}\} < 0.$$

*Proof.* Let us first compute the Hessian of the risk with respect to the $\boldsymbol{W}$ variable. We have

$$\langle \boldsymbol{Z}, \nabla^2_{\boldsymbol{W}} L(\boldsymbol{W}, c)\boldsymbol{Z} \rangle = 2c \cdot \mathrm{Tr}(\boldsymbol{Z}\boldsymbol{Z}^\mathsf{T}) + 2\mathrm{Tr}(\boldsymbol{W}\boldsymbol{W}^\mathsf{T} - \boldsymbol{B})\mathrm{Tr}(\boldsymbol{Z}\boldsymbol{Z}^\mathsf{T}) + 4\mathrm{Tr}(\boldsymbol{W}\boldsymbol{Z}^\mathsf{T})^2$$
$$+ 4\|\boldsymbol{W}\boldsymbol{Z}^\mathsf{T}\|_F^2 + 4\mathrm{Tr}(\boldsymbol{W}\boldsymbol{Z}^\mathsf{T}\boldsymbol{W}\boldsymbol{Z}^\mathsf{T}) + 4\langle \boldsymbol{W}\boldsymbol{W}^\mathsf{T} - \boldsymbol{B}, \boldsymbol{Z}\boldsymbol{Z}^\mathsf{T} \rangle.$$

Plugging the value of $c_0$ at a critical point (cf Eq. (70)), we get

$$\langle \boldsymbol{Z}, \nabla^2_{\boldsymbol{W}} L(\boldsymbol{W}_0, c_0)\boldsymbol{Z} \rangle = 4\mathrm{Tr}(\boldsymbol{W}_0\boldsymbol{Z}^\mathsf{T})^2 + 4\|\boldsymbol{W}_0\boldsymbol{Z}^\mathsf{T}\|_F^2 + 4\mathrm{Tr}(\boldsymbol{W}_0\boldsymbol{Z}^\mathsf{T}\boldsymbol{W}_0\boldsymbol{Z}^\mathsf{T}) + 4\langle \boldsymbol{W}_0\boldsymbol{W}_0^\mathsf{T} - \boldsymbol{B}, \boldsymbol{Z}\boldsymbol{Z}^\mathsf{T} \rangle. \tag{73}$$

**Case 1:** Consider the case $\mathrm{rank}(\boldsymbol{W}_0) < \min\{\mathrm{rank}(\boldsymbol{B}), N\}$. Then there exists an $i \in [d]$ such that $\boldsymbol{B}_{ii} > 0$ (recall that we assumed $\boldsymbol{B}$ diagonal , with diagonal elements given by the positive eigenvalues of $\boldsymbol{B}$) and $(\boldsymbol{W}_0 \boldsymbol{W}_0^\mathsf{T})_{ii} = 0$. For simplicity, let us permute the coordinates so that $i = 1$. The singular value decomposition of $\boldsymbol{W}_0$ verifies

$$\boldsymbol{W}_0 = \boldsymbol{U}_0 \boldsymbol{S}_0 \boldsymbol{V}_0^\mathsf{T} = \begin{pmatrix} 0 & 0 & \ldots & 0 \\ 0 & & & \\ \vdots & & \tilde{\boldsymbol{U}}_0 \tilde{\boldsymbol{S}}_0 & \\ 0 & & & \end{pmatrix} \boldsymbol{V}_0^\mathsf{T},$$

where $\tilde{\boldsymbol{U}}_0$ and $\tilde{\boldsymbol{S}}_0$ are the sub-matrices corresponding respectively to the $(d-1) \times (d-1)$ last coordinates of $\boldsymbol{U}_0$ and $(d-1) \times (N-1)$ last coordinates of $\boldsymbol{S}_0$. Let us consider

$$\boldsymbol{Z} = \begin{pmatrix} 1 & 0 & \ldots & 0 \\ 0 & & & \\ \vdots & & \boldsymbol{0} & \\ 0 & & & \end{pmatrix} \boldsymbol{V}_0^\mathsf{T}.$$

We have $\|\boldsymbol{Z}\|_F = 1$ and $\boldsymbol{W}_0 \boldsymbol{Z}^\top = 0$. Plugging these matrices in the above expression of the Hessian, see Eq. (73), we get

$$\langle \boldsymbol{Z}, \nabla^2_{\boldsymbol{W}} L(\boldsymbol{W}_0, c_0) \boldsymbol{Z} \rangle = -4\boldsymbol{B}_{11} \leq -4\delta_{\text{eig}}.$$

**Case 2:** Consider the case when $\text{rank}(\boldsymbol{W}_0 \boldsymbol{W}_0^\top) = N < \text{rank}(\boldsymbol{B})$ and $\boldsymbol{W}_0 \boldsymbol{W}_0^\top$ does not correspond to the $N$ largest eigenvalues of $\boldsymbol{B}$. Then there exists $i \neq j \in [n]$, such that $\boldsymbol{B}_{ii} > \boldsymbol{B}_{jj}$, $(\boldsymbol{W}_0 \boldsymbol{W}_0^\top)_{ii} = 0$ and $(\boldsymbol{W}_0 \boldsymbol{W}_0^\top)_{jj} = \boldsymbol{B}_{jj}$. For simplicity, let us permute the coordinates such that $i = 1$ and $j = 2$. The SVD decomposition of $\boldsymbol{W}_0$ now verifies:

$$\boldsymbol{W}_0 = \boldsymbol{U}_0 \boldsymbol{S}_0 \boldsymbol{V}_0^\top = \begin{pmatrix} 0 & 0 & \dots & 0 \\ \sqrt{\boldsymbol{B}_{22}} & 0 & \dots & 0 \\ 0 & & & \\ \vdots & & \tilde{\boldsymbol{U}}_0 \tilde{\boldsymbol{S}}_0 & \\ 0 & & & \end{pmatrix} \boldsymbol{V}_0^\top,$$

where $\tilde{\boldsymbol{U}}_0 \tilde{\boldsymbol{S}}_0$ is the sub-matrix of the last $(d-2) \times (N-1)$ coordinate of $\boldsymbol{U}_0 \boldsymbol{S}_0$. Let us consider again

$$\boldsymbol{Z} = \begin{pmatrix} 1 & 0 & \dots & 0 \\ 0 & & & \\ \vdots & & \boldsymbol{0} & \\ 0 & & & \end{pmatrix} \boldsymbol{V}_0^\top.$$

We have $\|\boldsymbol{Z}\|_F = 1$. Plugging these matrices in the above expression of the Hessian (73), note

$$\text{Tr}(\boldsymbol{W}_0 \boldsymbol{Z}^\top) = \text{Tr}(\boldsymbol{W}_0 \boldsymbol{Z}^\top \boldsymbol{W}_0 \boldsymbol{Z}^\top) = 0, \quad \|\boldsymbol{W}_0 \boldsymbol{Z}^\top\|_F^2 = \boldsymbol{B}_{22}, \quad \langle \boldsymbol{W}_0 \boldsymbol{W}_0^\top - \boldsymbol{B}, \boldsymbol{Z} \boldsymbol{Z}^\top \rangle = \boldsymbol{B}_{11},$$

we get

$$\langle \boldsymbol{Z}, \nabla^2_{\boldsymbol{W}} L(\boldsymbol{W}_0, c_0) \boldsymbol{Z} \rangle = -4(\boldsymbol{B}_{11} - \boldsymbol{B}_{22}) \leq -4\delta^{\text{sep}}.$$

This proves the proposition. $\square$

We can now prove Proposition 1.

*Proof of Proposition 1.* First, remark that $L(\boldsymbol{W}, c)$ has compact sub-level sets. The proposition then follows from Proposition 4 and the continuity of the gradient $\nabla L(\boldsymbol{x})$ and of the minimum eigenvalue of the Hessian $\lambda_{\min}(\nabla^2 L(\boldsymbol{x}))$. $\square$

### B.3.3 Dynamics

The following lemma is a standard combination of Lojasiewicz inequality and center and stable manifold theorem. We prove it for completeness.

**Lemma 8.** *Let $f : \mathbb{R}^d \to \mathbb{R}$ be an analytic function that has compact level sets. Consider the gradient flow*

$$\dot{\boldsymbol{x}}_t = -\nabla f(\boldsymbol{x}_t).$$

*Then for (Lebesgue) almost all initialization $\boldsymbol{x}_0$, there exists a second order local minimizer $\boldsymbol{x}_*$, such that*

$$\lim_{t \to +\infty} \boldsymbol{x}_t = \boldsymbol{x}_*.$$

*Proof of Lemma 8.*
**Step 1. Show convergence to a critical point.** Since $f$ is an analytic function, by Lojasiewicz inequality [Loj82], and the fact that the level set of $f$ is compact, we have

$$\lim_{t \to +\infty} \boldsymbol{x}_t = \boldsymbol{x}_*$$

for $\boldsymbol{x}_*$ some critical point of $f$.

**Step 2. Show convergence to a local minimizer.** In this step, we proceed similarly to the proof of Theorem 3 in [PP16]. First, consider a sublevel set

$$\Omega(K) = \{\boldsymbol{x} : f(\boldsymbol{x}) \leq K\}.$$

Then we have $\Omega(K)$ compact. Since $f$ is an analytic function, $\nabla f$ is Lipschitz in the compact set $\Omega(K)$. We define the map $\phi_t : \Omega(K) \to \phi_t(\Omega(K))$, $\boldsymbol{x} \mapsto \boldsymbol{x}_t$ where $\boldsymbol{x}_t$ is defined as the solution of

$$\dot{\boldsymbol{x}}_t = -\nabla f(\boldsymbol{x}_t),$$
$$\boldsymbol{x}_0 = \boldsymbol{x}.$$

By Picard's existence and uniqueness theorem, we have $\phi_t$ is a diffeomorphism from $\Omega(K)$ to $\phi(\Omega(K))$ for any $t > 0$. Fix an $\varepsilon_0 > 0$, and we define $g = \phi_{\varepsilon_0} : \Omega(K) \to \Omega(K)$.

Let $\boldsymbol{r}$ be a strict saddle point of $f$, then $\boldsymbol{r}$ must be an unstable fixed point of the diffeomorphism $g = \phi_{\varepsilon_0}$. By center and stable manifold theorem (such as Theorem 9 in [PP16]), there exists a manifold $W_{\text{loc}}^{\text{sc}}(\boldsymbol{r})$ of dimension at most $d - 1$, and a ball $\mathsf{B}(\boldsymbol{r}, \varepsilon(\boldsymbol{r}))$ centered at $\boldsymbol{r}$ with radius $\varepsilon(\boldsymbol{r})$, such that we have the following facts:

(1) $g\left(W_{\text{loc}}^{\text{sc}}(\boldsymbol{r}) \cap \mathsf{B}(\boldsymbol{r}, \varepsilon(\boldsymbol{r}))\right) \subseteq W_{\text{loc}}^{\text{sc}}(\boldsymbol{r})$;

(2) If $g^n(\boldsymbol{x}) \in \mathsf{B}(\boldsymbol{r}, \varepsilon(\boldsymbol{r}))$ for all $n \geq 0$, we have $\boldsymbol{x} \in W_{\text{loc}}^{\text{sc}}(\boldsymbol{r})$ (here $g^n$ means composition of $g$ for $n$ times).

We consider the union of the balls associated to all the strict saddle points of $f$ in $\Omega(K)$

$$A = \cup_{\boldsymbol{r} \in \Omega(K): \boldsymbol{r} \text{ strict saddle}} \mathsf{B}(\boldsymbol{r}, \varepsilon(\boldsymbol{r})).$$

Due to Lindelof's lemma, we can find a countable subcover for $A$, i.e., there exists fixed-points $\boldsymbol{r}_1, \boldsymbol{r}_2, \ldots$ such that $A = \cup_{m=1}^{\infty} \mathsf{B}(\boldsymbol{r}_m, \varepsilon(\boldsymbol{r}_m))$. If gradient descent converges to a strict saddle point, starting from a point $\boldsymbol{v} \in \Omega(K)$, there must exist a $t_0$ and $m$ such that $\phi_t(\boldsymbol{v}) \in \mathsf{B}(\boldsymbol{r}_m, \varepsilon(\boldsymbol{r}_m))$ for all $t \geq t_0$. By center and stable manifold theorem, we get that $\phi_t(\boldsymbol{v}) \in W_{\text{loc}}^{\text{sc}}(\boldsymbol{r}_m) \cap \Omega(K)$. By setting $D_1(\boldsymbol{r}_m) = g^{-1}(W_{\text{loc}}^{\text{sc}}(\boldsymbol{r}_m) \cap \Omega(K))$ and $D_{i+1}(\boldsymbol{r}_m) = g^{-1}(D_i(\boldsymbol{r}_m) \cap \Omega(K))$ we get that $\boldsymbol{v} \in D_k(\boldsymbol{r}_m)$ for all $k\varepsilon_0 \geq t_0$. Hence the set of initial points in $\Omega(K)$ such that gradient descent converges to a strict saddle point is a subset of

$$P = \cup_{m=1}^{\infty} \cup_{k \in \mathbb{N}} D_k(\boldsymbol{r}_m).$$

Note that the set $W_{\text{loc}}^{\text{sc}}(\boldsymbol{r}_m) \cap \Omega(K)$ has Lebesgue measure zero in $\mathbb{R}^d$. Since $g$ is a diffeomorphism, $g^{-1}$ is continuously differentiable and thus it is locally Lipschitz. Therefore, $g^{-1}$ preserves the null-sets and hence (by induction) $D_i(\boldsymbol{r}_m)$ has measure zero for all $i$. Thereby we get that $P$ is a countable union of measure zero sets. Hence $P$ has measure 0.

Finally, note we have

$$\{\boldsymbol{x} \in \Omega(K) : \exists \boldsymbol{r}, \boldsymbol{r} \text{ is strict saddle}, \boldsymbol{r} = \lim_{t \to +\infty} \phi_t(\boldsymbol{x})\} \subseteq P.$$

Since $P$ has measure 0, we have

$$\{\boldsymbol{x} \in \mathbb{R}^d : \exists \boldsymbol{r}, \boldsymbol{r} \text{ is strict saddle}, \boldsymbol{r} = \lim_{t \to +\infty} \phi_t(\boldsymbol{x})\}$$
$$= \cup_{K \in \mathbb{N}} \{\boldsymbol{x} \in \Omega(K) : \exists \boldsymbol{r}, \boldsymbol{r} \text{ is strict saddle}, \boldsymbol{r} = \lim_{t \to +\infty} \phi_t(\boldsymbol{x})\}$$

has measure 0. This proves the lemma. □

The following lemma is standard, and a corollary of Theorem 2.11 in [Kur70].

**Lemma 9.** *Let*

$$F(\boldsymbol{x}) = \mathbb{E}_{\boldsymbol{z}}[f(\boldsymbol{x}; \boldsymbol{z})]$$

*be a $C^2$ function on $\Omega \subseteq \mathbb{R}^d$. Assume*

$$\sup_{\boldsymbol{x} \in \Omega} \mathbb{E}_{\boldsymbol{z}}[\|\nabla_{\boldsymbol{x}} f(\boldsymbol{x}; \boldsymbol{z})\|_2] < \infty,$$

$$\sup_{\boldsymbol{x} \in \Omega} \|\nabla^2 F(\boldsymbol{x})\|_{\mathrm{op}} < \infty.$$

*Let $\boldsymbol{x}_t$ be the trajectory of*

$$\dot{\boldsymbol{x}}_t = -\nabla F(\boldsymbol{x}_t),$$

*with initialization $\boldsymbol{x}_0 \in \Omega$. Further assume that there exists $\eta > 0$, such that $\cup_{t \geq 0} \mathsf{B}(\boldsymbol{x}_t, \eta) \subseteq \Omega$.*

*Consider the following Markov jump process $\boldsymbol{x}_{t,\varepsilon}$ starting from $\boldsymbol{x}_0$, with jump time to be an exponential random variable with fixed mean $\varepsilon$, and jump direction $-\varepsilon \nabla f(\boldsymbol{x}; \boldsymbol{z})$ where $\boldsymbol{x}$ is the current state, and $\boldsymbol{z}$ an independent sample. Then we have for any fixed $T > 0$ and $\delta > 0$,*

$$\lim_{\varepsilon \to 0+} \mathbb{P}\Big( \sup_{0 \leq t \leq T} \|\boldsymbol{x}_t - \boldsymbol{x}_{t,\varepsilon}\|_2 \geq \delta \Big) = 0.$$

### B.3.4   Proof of Theorem 3

By Proposition 4, we know that for $L(\boldsymbol{W}, c)$, any critical point that is not a global minimizer is a strict saddle point. Consider the gradient flow

$$\frac{\mathrm{d}}{\mathrm{d}t}(\boldsymbol{W}_t, c_t) = -\nabla L(\boldsymbol{W}_t, c_t)$$

with random initialization $(\boldsymbol{W}_0, c_0) \sim \nu_0$ where $\nu_0$ is a distribution that is absolutely continuous with respect to Lebesgue measure. Since $L(\boldsymbol{W}, c)$ is an analytic function, by Lemma 8, we have $(\boldsymbol{W}_t, c_t)$ converges to a global minimizer of $L(\boldsymbol{W}, c)$. That is, we have almost surely (over $\nu_0$)

$$\lim_{t \to \infty} L(\boldsymbol{W}_t, c_t) = \inf_{\boldsymbol{W}, c} L(\boldsymbol{W}, c),$$

where $\inf_{\boldsymbol{W}, c} L(\boldsymbol{W}, c)$ is calculated in Lemma 7.

Consider the following Markov jump process $(\boldsymbol{W}_{t,\varepsilon}, c_{t,\varepsilon})$ starting from $(\boldsymbol{W}_0, c_0) \sim \nu_0$, with jump time to be an exponential random variable with fixed mean $\varepsilon$, and jump direction to be $-\varepsilon \nabla L(\boldsymbol{W}, c; \boldsymbol{z})$ where

$$\nabla L(\boldsymbol{W}, c; \boldsymbol{z}) = \begin{pmatrix} \nabla_{\boldsymbol{W}} L(\boldsymbol{W}, c; \boldsymbol{z}) \\ \partial_c L(\boldsymbol{W}, c; \boldsymbol{z}) \end{pmatrix} = \begin{pmatrix} 2(c - b_0 + \langle \boldsymbol{z}\boldsymbol{z}^{\mathsf{T}}, \boldsymbol{W}\boldsymbol{W}^{\mathsf{T}} - \boldsymbol{B} \rangle) \boldsymbol{z}\boldsymbol{z}^{\mathsf{T}} \boldsymbol{W} \\ 2(c - b_0 + \langle \boldsymbol{z}\boldsymbol{z}^{\mathsf{T}}, \boldsymbol{W}\boldsymbol{W}^{\mathsf{T}} - \boldsymbol{B} \rangle) \end{pmatrix}$$

with $(\boldsymbol{W}, c)$ the current state, and $\boldsymbol{z}$ an independent sample. By Lemma 9, we have for any fixed $T > 0$ and $\delta > 0$,

$$\lim_{\varepsilon \to 0+} \mathbb{P}\Big( \sup_{0 \leq t \leq T} \|(\boldsymbol{W}_{t,\varepsilon}, c_{t,\varepsilon}) - (\boldsymbol{W}_t, c_t)\|_2 \geq \delta \Big) = 0.$$

Note the sequence of Markov jump process at jump time is exactly the SGD iterates. Hence the SGD iterates with properly scaled number of iterations is uniformly close to $(\boldsymbol{W}_t, c_t)$ over finite horizon as $\varepsilon \to 0$. This proves the Theorem.

## C   Proofs for Mixture of Gaussians

In this section, we consider the mixture of Gaussian setting (mg): $y_i = \pm 1$ with equal probability $1/2$, and $\boldsymbol{x}_i | y_i = +1 \sim \mathsf{N}(0, \boldsymbol{\Sigma}^{(1)})$, $\boldsymbol{x}_i | y_i = -1 \sim \mathsf{N}(0, \boldsymbol{\Sigma}^{(2)})$ where $\boldsymbol{\Sigma}^{(1)} = \boldsymbol{\Sigma} - \boldsymbol{\Delta}$ and $\boldsymbol{\Sigma}^{(2)} = \boldsymbol{\Sigma} + \boldsymbol{\Delta}$. With these notations,

$$\boldsymbol{\Sigma} = \frac{1}{2}(\boldsymbol{\Sigma}^{(1)} + \boldsymbol{\Sigma}^{(2)}),$$

$$\boldsymbol{\Delta} = \frac{1}{2}(\boldsymbol{\Sigma}^{(2)} - \boldsymbol{\Sigma}^{(1)}).$$

Throughout this section, we will make the following assumptions:

**M1.** There exists constants $0 < c_1 < c_2$ such that $c_1 \mathbf{I}_d \preceq \mathbf{\Sigma} \preceq c_2 \mathbf{I}_d$;

**M2.** $\|\mathbf{\Delta}\|_{\mathrm{op}} = \Theta_d(1/\sqrt{d})$.

Throughout this section, we will denote $\mathbb{P}_{\mathbf{\Sigma},\mathbf{\Delta}}$ the joint distribution of $(y, \boldsymbol{x})$ under the $\mathsf{mg}$ model, $\mathbb{E}_{\boldsymbol{x},y}$ the expectation operator with respect to $(y, \boldsymbol{x}) \sim \mathbb{P}_{\mathbf{\Sigma},\mathbf{\Delta}}$ and $\mathbb{E}_{\boldsymbol{x}}$ the expectation operator with respect to the marginal distribution $\boldsymbol{x} \sim (1/2) \cdot \mathsf{N}(0, \mathbf{\Sigma}^{(1)}) + (1/2) \cdot \mathsf{N}(0, \mathbf{\Sigma}^{(2)})$.

## C.1    Random Features model: proof of Theorem 4

Recall the definition

$$R_{\mathsf{RF},N}(\mathbb{P}) = \min_{\hat{f} \in \mathcal{F}_{\mathsf{RF},N}(\boldsymbol{W})} \mathbb{E}\big\{(y - \hat{f}(\boldsymbol{x}))^2\big\},$$

where

$$\mathcal{F}_{\mathsf{RF},N}(\boldsymbol{W}) = \Big\{ f_N(\boldsymbol{x}) = \sum_{i=1}^{N} a_i \sigma(\langle \boldsymbol{w}_i, \boldsymbol{x} \rangle) : \; a_i \in \mathbb{R}, i \in [N] \Big\}.$$

Note that it is easy to see from the proof that the result stays the same if we add an offset $c$.

**Remark 2.** *We will state the lemmas for the case $\mathbf{\Sigma} = \mathbf{I}_d$, which amounts to re-scaling $\tilde{\mathbf{\Gamma}} = \mathbf{\Sigma}^{1/2} \mathbf{\Gamma} \mathbf{\Sigma}^{1/2}$ and $\tilde{\mathbf{\Delta}} = \mathbf{\Sigma}^{-1/2} \mathbf{\Delta} \mathbf{\Sigma}^{-1/2}$.*

### C.1.1    Representation of the $\mathsf{RF}$ risk

**Lemma 10.** *Consider the RF model introduced above. We have*

$$R_{\mathsf{RF},N}(\mathbb{P}_{\mathbf{I},\mathbf{\Delta}}) = \mathbb{E}_{\boldsymbol{x},y}[y^2] - \boldsymbol{V}^{\mathsf{T}} \boldsymbol{U}^{-1} \boldsymbol{V}, \tag{74}$$

*where $\boldsymbol{V} = [V_1, \ldots, V_N]^{\mathsf{T}}$, and $\boldsymbol{U} = (U_{ij})_{i,j \in [N]}$, with*

$$\begin{aligned} V_i &= \mathbb{E}_{\boldsymbol{x},y}[y\sigma(\langle \boldsymbol{w}_i, \boldsymbol{x} \rangle)], \\ U_{ij} &= \mathbb{E}_{\boldsymbol{x},y}[\sigma(\langle \boldsymbol{w}_i, \boldsymbol{x} \rangle)\sigma(\langle \boldsymbol{w}_j, \boldsymbol{x} \rangle)]. \end{aligned}$$

*Proof.* Simply write the KKT conditions. The optimum is achieved at $\boldsymbol{a} = \boldsymbol{U}^{-1} \boldsymbol{V}$. $\qquad\qquad \square$

### C.1.2    Approximation of kernel matrix $U$

**Lemma 11.** *Let $\sigma \in L^2(\mathsf{N}(0,1))$ be an activation function. Denote $\lambda_k = \mathbb{E}_{G \sim \mathsf{N}(0,1)}[\sigma(G)\mathrm{He}_k(G)]$ the $k$-th Hermite coefficient of $\sigma$ and assume $\lambda_0 = 0$. Let $\boldsymbol{U} = (U_{ij})_{i,j \in [N]}$ be a random matrix with*

$$U_{ij} = \mathbb{E}_{\boldsymbol{x}}[\sigma(\langle \boldsymbol{w}_i, \boldsymbol{x} \rangle)\sigma(\langle \boldsymbol{w}_j, \boldsymbol{x} \rangle)],$$

*where $(\boldsymbol{w}_i)_{i \in [N]} \sim \mathsf{N}(\mathbf{0}, \mathbf{\Gamma})$ independently. Assume conditions **A1** and **B2** hold.*
    *Define $\boldsymbol{W} = (\boldsymbol{w}_1, \ldots, \boldsymbol{w}_N) \in \mathbb{R}^{d \times N}$, and $\boldsymbol{U}_0 = \{(U_0)_{ij}\}_{i,j \in [N]}$, with*

$$(U_0)_{ij} = \tilde{\lambda}\delta_{ij} + \lambda_1^2 \langle \boldsymbol{w}_i, \boldsymbol{w}_j \rangle + \kappa/d + \mu_i \mu_j,$$

*where*

$$\begin{aligned} \mu_i &= \lambda_2(\|\boldsymbol{w}_i\|_2^2 - 1)/2, \\ \tilde{\lambda} &= \mathbb{E}[\sigma(G)^2] - \lambda_1^2, \\ \kappa &= d \cdot \lambda_2^2 [\mathrm{Tr}(\mathbf{\Gamma}^2)/2 + \mathrm{Tr}(\mathbf{\Delta}\mathbf{\Gamma})^2/4]. \end{aligned}$$

*Then we have as $N/d = \rho$ and $d \to \infty$, we have*

$$\|\boldsymbol{U} - \boldsymbol{U}_0\|_{\mathrm{op}} = o_{d,\mathbb{P}}(1).$$

*Proof of Lemma 11.* Recalling that in the (mg) model, we have $\boldsymbol{x} \sim (1/2) \cdot \mathsf{N}(\boldsymbol{0}, \mathbf{I} - \boldsymbol{\Delta}) + (1/2) \cdot \mathsf{N}(\boldsymbol{0}, \mathbf{I} + \boldsymbol{\Delta})$, we have

$$
\begin{aligned}
U_{ij} =& \mathbb{E}_{\boldsymbol{x}}[\sigma(\langle \boldsymbol{w}_i, \boldsymbol{x} \rangle)\sigma(\langle \boldsymbol{w}_j, \boldsymbol{x} \rangle)] \\
=& \Big\{ \mathbb{E}_{\boldsymbol{x} \sim \mathsf{N}(\boldsymbol{0}, \mathbf{I})}[\sigma(\langle (\mathbf{I} - \boldsymbol{\Delta})^{1/2}\boldsymbol{w}_i, \boldsymbol{x} \rangle)\sigma(\langle (\mathbf{I} - \boldsymbol{\Delta})^{1/2}\boldsymbol{w}_j, \boldsymbol{x} \rangle)] \\
& + \mathbb{E}_{\boldsymbol{x} \sim \mathsf{N}(\boldsymbol{0}, \mathbf{I})}[\sigma(\langle (\mathbf{I} + \boldsymbol{\Delta})^{1/2}\boldsymbol{w}_i, \boldsymbol{x} \rangle)\sigma(\langle (\mathbf{I} + \boldsymbol{\Delta})^{1/2}\boldsymbol{w}_j, \boldsymbol{x} \rangle)] \Big\}/2.
\end{aligned}
$$

We can therefore readily use the result of Lemma 2 for $\tilde{\boldsymbol{w}}_i \sim \mathsf{N}(\boldsymbol{0}, (\mathbf{I} - \boldsymbol{\Delta})^{1/2}\boldsymbol{\Gamma}(\mathbf{I} - \boldsymbol{\Delta})^{1/2})$ and $\tilde{\boldsymbol{w}}_i \sim \mathsf{N}(\boldsymbol{0}, (\mathbf{I} + \boldsymbol{\Delta})^{1/2}\boldsymbol{\Gamma}(\mathbf{I} + \boldsymbol{\Delta})^{1/2})$, to get

$$
\|\boldsymbol{U} - \tilde{\boldsymbol{U}}_0\|_{\mathrm{op}} = o_{d,\mathbb{P}}(1), \tag{75}
$$

where $\tilde{\boldsymbol{U}}_0 = (\tilde{U}_0)_{i,j \in [N]}$ with

$$
(\tilde{U}_0)_{ij} = \tilde{\lambda}\delta_{ij} + \lambda_1^2\langle \boldsymbol{w}_i, \boldsymbol{w}_j \rangle + \kappa/d + (\mu_i^+\mu_j^+ + \mu_i^-\mu_j^-)/2,
$$

and

$$
\begin{aligned}
\tilde{\lambda} =& \mathbb{E}[\sigma(G)^2] - \lambda_1^2, \\
\tilde{\kappa} =& d\lambda_2^2[\mathrm{Tr}((\mathbf{I} - \boldsymbol{\Delta})\boldsymbol{\Gamma}(\mathbf{I} - \boldsymbol{\Delta})\boldsymbol{\Gamma}) + \mathrm{Tr}((\mathbf{I} + \boldsymbol{\Delta})\boldsymbol{\Gamma}(\mathbf{I} + \boldsymbol{\Delta})\boldsymbol{\Gamma})]/4 \\
=& d\lambda_2^2[\mathrm{Tr}(\boldsymbol{\Gamma}^2) + \mathrm{Tr}(\boldsymbol{\Delta}\boldsymbol{\Gamma}\boldsymbol{\Delta}\boldsymbol{\Gamma})]/2, \\
\mu_i^+ =& \lambda_2(\|(\mathbf{I} + \boldsymbol{\Delta})^{1/2}\boldsymbol{w}_i\|_2^2 - 1)/2, \\
\mu_i^- =& \lambda_2(\|(\mathbf{I} - \boldsymbol{\Delta})^{1/2}\boldsymbol{w}_i\|_2^2 - 1)/2.
\end{aligned}
$$

Note that we have

$$
(\mu_i^+\mu_j^+ + \mu_i^-\mu_j^-)/2 = \mu_i\mu_j + \lambda_2^2\langle \boldsymbol{w}_i, \boldsymbol{\Delta}\boldsymbol{w}_i \rangle\langle \boldsymbol{w}_j, \boldsymbol{\Delta}\boldsymbol{w}_j \rangle/4,
$$

where

$$
\mu_i = \lambda_2(\|\boldsymbol{w}_i\|_2^2 - 1)/2.
$$

The matrix $(\langle \boldsymbol{w}_i, \boldsymbol{\Delta}\boldsymbol{w}_i \rangle\langle \boldsymbol{w}_j, \boldsymbol{\Delta}\boldsymbol{w}_j \rangle)_{i,j \in [N]}$ is simply $\boldsymbol{s}\boldsymbol{s}^{\mathsf{T}}$ with $\boldsymbol{s} = (\langle \boldsymbol{w}_i, \boldsymbol{\Delta}\boldsymbol{w}_i \rangle)_{i \in [N]}$. Defining $\nu = \mathbb{E}[\langle \boldsymbol{w}_i, \boldsymbol{\Delta}\boldsymbol{w}_i \rangle] = \mathrm{Tr}(\boldsymbol{\Gamma}\boldsymbol{\Delta})$, we have

$$
\boldsymbol{s}\boldsymbol{s}^{\mathsf{T}} = (\boldsymbol{s} - \nu\boldsymbol{1})\nu\boldsymbol{1}^{\mathsf{T}} + \nu\boldsymbol{1}(\boldsymbol{s} - \nu\boldsymbol{1})^{\mathsf{T}} + \nu^2\boldsymbol{1}\boldsymbol{1}^{\mathsf{T}} + (\boldsymbol{s} - \nu\boldsymbol{1})(\boldsymbol{s} - \nu\boldsymbol{1})^{\mathsf{T}}.
$$

Furthermore:

$$
\|\boldsymbol{s} - \nu\boldsymbol{1}\|_2^2 = \sum_{i=1}^d \mathrm{Tr}((\boldsymbol{w}_i\boldsymbol{w}_i^{\mathsf{T}} - \boldsymbol{\Gamma})\boldsymbol{\Delta})^2.
$$

Note that by assumptions **M2** and **B2**, we have $\mathbb{E}[\mathrm{Tr}((\boldsymbol{w}_i\boldsymbol{w}_i^{\mathsf{T}} - \boldsymbol{\Gamma})\boldsymbol{\Delta})^2] = 2\|\boldsymbol{\Delta}\boldsymbol{\Gamma}\|_F^2 = o_{d,\mathbb{P}}(d^{-1})$. We deduce that $\|\boldsymbol{s} - \nu\boldsymbol{1}\|_2 = o_{d,\mathbb{P}}(1)$, and therefore

$$
\|(\boldsymbol{s} - \nu\boldsymbol{1})\nu\boldsymbol{1}^{\mathsf{T}}\|_{\mathrm{op}} = o_{d,\mathbb{P}}(1),
$$
$$
\|(\boldsymbol{s} - \nu\boldsymbol{1})(\boldsymbol{s} - \nu\boldsymbol{1})^{\mathsf{T}}\|_{\mathrm{op}} = o_{d,\mathbb{P}}(1).
$$

Hence, we get

$$
\|(\boldsymbol{\mu}^+\boldsymbol{\mu}^{+\mathsf{T}} + \boldsymbol{\mu}^-\boldsymbol{\mu}^{-\mathsf{T}})/2 - \boldsymbol{\mu}\boldsymbol{\mu}^{\mathsf{T}} - \mathrm{Tr}(\boldsymbol{\Gamma}\boldsymbol{\Delta})^2\boldsymbol{1}\boldsymbol{1}^{\mathsf{T}}\|_{\mathrm{op}} = o_{d,\mathbb{P}}(1). \tag{76}
$$

We also have $\mathrm{Tr}(\boldsymbol{\Delta}\boldsymbol{\Gamma}\boldsymbol{\Delta}\boldsymbol{\Gamma})^2 = o_d(d^{-1})$ by assumptions **M2** and **B2**, hence

$$
\|\mathrm{Tr}(\boldsymbol{\Delta}\boldsymbol{\Gamma}\boldsymbol{\Delta}\boldsymbol{\Gamma})\boldsymbol{1}\boldsymbol{1}^{\mathsf{T}}\|_{\mathrm{op}} = o_{d,\mathbb{P}}(1). \tag{77}
$$

Therefore, combining (76) and (77), we get:

$$
\|\tilde{\boldsymbol{U}}_0 - \boldsymbol{U}_0\|_{\mathrm{op}} = o_{d,\mathbb{P}}(1). \tag{78}
$$

Combining (75) and (78) concludes the proof. $\qquad\square$

### C.1.3 Approximation of the $V$ vector

**Lemma 12.** *Under the assumption of Theorem 4, define $\boldsymbol{V} = (V_1, \ldots, V_N)^\mathsf{T}$ with*

$$V_i = \mathbb{E}_{\boldsymbol{x}, y}[y\sigma(\langle \boldsymbol{w}_i, \boldsymbol{x} \rangle)]$$

*where $(\boldsymbol{w}_i)_{i \in [N]} \sim \mathsf{N}(\mathbf{0}, \boldsymbol{\Gamma})$ independently. Then as $N/d = \rho$ with $d \to \infty$, we have*

$$\|\boldsymbol{V} - \tau \mathbf{1}/\sqrt{d}\|_2 = o_{d,\mathbb{P}}(1),$$

*where*

$$\tau = -\sqrt{d} \cdot \lambda_2 \mathrm{Tr}(\boldsymbol{\Delta}\boldsymbol{\Gamma})/2.$$

*Proof of Lemma 12.* We have

$$
\begin{aligned}
V_i =& \{\mathbb{E}_{\boldsymbol{x} \sim \mathsf{N}(\mathbf{0}, \mathbf{I} - \boldsymbol{\Delta})}[\sigma(\langle \boldsymbol{w}_i, \boldsymbol{x} \rangle)] - \mathbb{E}_{\boldsymbol{x} \sim \mathsf{N}(\mathbf{0}, \mathbf{I} + \boldsymbol{\Delta})}[\sigma\langle \boldsymbol{w}_i, \boldsymbol{x} \rangle]\}/2 \\
=& \{\mathbb{E}_{\boldsymbol{x} \sim \mathsf{N}(\mathbf{0}, \mathbf{I})}[\sigma(\langle (\mathbf{I} - \boldsymbol{\Delta})^{1/2} \boldsymbol{w}_i, \boldsymbol{x} \rangle)] - \mathbb{E}_{\boldsymbol{x} \sim \mathsf{N}(\mathbf{0}, \mathbf{I})}[\sigma\langle (\mathbf{I} + \boldsymbol{\Delta})^{1/2} \boldsymbol{w}_i, \boldsymbol{x} \rangle]\}/2 \\
=& \mathbb{E}_{G \sim \mathsf{N}(0,1)}[\sigma(\|(\mathbf{I} - \boldsymbol{\Delta})^{1/2} \boldsymbol{w}_i\|_2 G) - \sigma(\|(\mathbf{I} + \boldsymbol{\Delta})^{1/2} \boldsymbol{w}_i\|_2 G)]/2.
\end{aligned}
$$

We define three interpolating variables:

$$
\begin{aligned}
V_i^{(1)} &= \lambda_2 \{\|(\mathbf{I} - \boldsymbol{\Delta})^{1/2} \boldsymbol{w}_i\|_2 - \|(1 + \boldsymbol{\Delta})^{1/2} \boldsymbol{w}_i\|_2\}/2, \\
V_i^{(2)} &= -\lambda_2 \{\mathrm{Tr}(\boldsymbol{\Delta} \boldsymbol{w}_i \boldsymbol{w}_i^\mathsf{T})\}/2, \\
V_i^{(3)} &= -\lambda_2 \mathrm{Tr}(\boldsymbol{\Delta}\boldsymbol{\Gamma})/2.
\end{aligned}
$$

We begin by bounding the difference between $\boldsymbol{V}$ and $\boldsymbol{V}^{(1)}$. For convenience, we will define $\tilde{\boldsymbol{w}}_i = (\mathbf{I} - \boldsymbol{\Delta})^{1/2} \boldsymbol{w}_i$. We have:

$$
\begin{aligned}
&\mathbb{E}[\sigma(\|\tilde{\boldsymbol{w}}_i\|_2 G) - \sigma(G)] - \lambda_2(\|\tilde{\boldsymbol{w}}_i\|_2 - 1) \\
=& \mathbb{E}\Big[\frac{\sigma(\|\tilde{\boldsymbol{w}}_i\|_2 G) - \sigma(G) - (\|\tilde{\boldsymbol{w}}_i\|_2 - 1)G\sigma'(G)}{(\|\tilde{\boldsymbol{w}}_i\|_2 - 1)^2}\Big](\|\tilde{\boldsymbol{w}}_i\|_2 - 1)^2.
\end{aligned}
\tag{79}
$$

Using dominated convergence theorem and arguments similar to those used to prove (26), one can check that

$$
\lim_{t \to 1} \mathbb{E}\Big[\frac{\sigma(tG) - \sigma(G) - (t-1)G\sigma'(G)}{(t-1)^2}\Big] = (\lambda_4(\sigma) + \lambda_2(\sigma))/2.
\tag{80}
$$

The same arguments as in the proofs of Lemma 2 and Lemma 3 show

$$
\begin{aligned}
&\sup_{i \in [N]} |\|(\mathbf{I} - \boldsymbol{\Delta})^{1/2} \boldsymbol{w}_i\|_2 - 1| = o_{d,\mathbb{P}}(1), \\
&\sum_{i=1}^{N} (\|(\mathbf{I} - \boldsymbol{\Delta})^{1/2} \boldsymbol{w}_i\|_2 - 1)^2 = O_{d,\mathbb{P}}(1).
\end{aligned}
\tag{81}
$$

Combining (80) with (81) in (79), we get:

$$
\begin{aligned}
&\sum_{i=1}^{N} \Big(\mathbb{E}[\sigma(\|\tilde{\boldsymbol{w}}_i\|_2 G) - \sigma(G)] - \lambda_2(\|\tilde{\boldsymbol{w}}_i\|_2 - 1)\Big)^2 \\
=& \sum_{i=1}^{N} \Big(\frac{\mathbb{E}[\sigma(\|\tilde{\boldsymbol{w}}_i\|_2 G) - \sigma(G)] - \lambda_2(\|\tilde{\boldsymbol{w}}_i\|_2 - 1)}{(\|\tilde{\boldsymbol{w}}_i\|_2 - 1)^2}\Big)(\|\tilde{\boldsymbol{w}}_i\|_2 - 1)^4 \\
=& O_{d,\mathbb{P}}(1) \cdot \Big(\sup_{i \in [N]} |\|(\mathbf{I} - \boldsymbol{\Delta})^{1/2} \boldsymbol{w}_i\|_2 - 1|^2\Big) \sum_{i=1}^{N} (\|(\mathbf{I} - \boldsymbol{\Delta})^{1/2} \boldsymbol{w}_i\|_2 - 1)^2 = o_{d,\mathbb{P}}(1).
\end{aligned}
$$

Bounding similarly the term depending on $(\mathbf{I} + \boldsymbol{\Delta})^{1/2} \boldsymbol{w}_i$ in $V_i^{(1)}$, we get

$$\|\boldsymbol{V} - \boldsymbol{V}^{(1)}\|_2 = o_{d,\mathbb{P}}(1). \tag{82}$$

Now, consider the difference between $\boldsymbol{V}^{(1)}$ and $\boldsymbol{V}^{(2)}$. We use the fact for $x$ on a neighborhood of 0, there exists $c$ such that

$$|\sqrt{1-x} - \sqrt{1+x} + x| \le c|x|^3.$$

Hence, with high probability

$$\big|\|(\mathbf{I} - \boldsymbol{\Delta})^{1/2} \boldsymbol{w}_i\|_2 - \|(\mathbf{I} + \boldsymbol{\Delta})^{1/2} \boldsymbol{w}_i\|_2 + \langle \boldsymbol{w}_i, \boldsymbol{\Delta} \boldsymbol{w}_i \rangle\big| \le c \frac{|\langle \boldsymbol{w}_i, \boldsymbol{\Delta} \boldsymbol{w}_i \rangle|^3}{\|\boldsymbol{w}_i\|_2^2}.$$

Furthermore, we have:

$$\mathbb{E}_{\boldsymbol{w}_i \sim \mathsf{N}(\boldsymbol{0}, \boldsymbol{\Gamma})}\Big[(\langle \boldsymbol{w}_i, \boldsymbol{\Delta} \boldsymbol{w}_i \rangle)^6/\|\boldsymbol{w}_i\|_2^4\Big] \le \|\boldsymbol{\Delta}\|_{\mathrm{op}}^2 \mathbb{E}[(\langle \boldsymbol{w}_i, \boldsymbol{\Delta} \boldsymbol{w}_i \rangle)^4]$$

$$\le C\|\boldsymbol{\Delta}\|_{\mathrm{op}}^2 (\mathrm{Tr}[\boldsymbol{\Gamma}^{1/2} \boldsymbol{\Delta} \boldsymbol{\Gamma}^{1/2}]^4 + \|\boldsymbol{\Gamma}^{1/2} \boldsymbol{\Delta} \boldsymbol{\Gamma}^{1/2}\|_F^4) = o_d(d^{-1}),$$

where the last equality is due to assumptions **M2** and **B2**. We conclude that

$$\|\boldsymbol{V}^{(1)} - \boldsymbol{V}^{(2)}\|_2 = o_{d,\mathbb{P}}(1). \tag{83}$$

For the last comparison between $\boldsymbol{V}^{(2)}$ and $\boldsymbol{V}^{(3)}$, we take the expectation:

$$\mathbb{E}_{\boldsymbol{w}_i \sim \mathsf{N}(\boldsymbol{0}, \boldsymbol{\Gamma})}[(\langle \boldsymbol{w}_i, \boldsymbol{\Delta} \boldsymbol{w}_i \rangle - \mathrm{Tr}(\boldsymbol{\Gamma} \boldsymbol{\Delta}))^2] = \mathbb{E}_{\boldsymbol{g} \sim \mathsf{N}(\boldsymbol{0}, \mathbf{I})}[(\langle \boldsymbol{g} \boldsymbol{g}^\mathsf{T}, \boldsymbol{\Gamma}^{1/2} \boldsymbol{\Delta} \boldsymbol{\Gamma}^{1/2} \rangle - \mathrm{Tr}(\boldsymbol{\Gamma} \boldsymbol{\Delta}))^2]$$

$$= 2\|\boldsymbol{\Gamma}^{1/2} \boldsymbol{\Delta} \boldsymbol{\Gamma}^{1/2}\|_F^2$$

$$\le 2\|\boldsymbol{\Gamma}\|_{\mathrm{op}}^2 \|\boldsymbol{\Delta}\|_F^2 = O_d(d^{-2}).$$

We get

$$\|\boldsymbol{V}^{(3)} - \boldsymbol{V}^{(2)}\|_2 = o_{d,\mathbb{P}}(1). \tag{84}$$

Combining the above three bounds (82), (83) and (84) yields the desired result. $\qquad\square$

### C.1.4  Proof of Theorem 4

By Lemma 10, the risk has a representation

$$R_{\mathsf{RF},N}(f_*) = 1 - \boldsymbol{V}^\mathsf{T} \boldsymbol{U}^{-1} \boldsymbol{V}.$$

By Lemma 11, we have

$$\|\boldsymbol{U} - \boldsymbol{U}_0\|_{\mathrm{op}} = o_{d,\mathbb{P}}(1).$$

By Lemma 12, we have

$$\|\boldsymbol{V} - \tau \mathbf{1}/\sqrt{d}\|_2 = o_{d,\mathbb{P}}(1),$$

where

$$\tau = -\sqrt{d} \cdot \lambda_2 \mathrm{Tr}(\boldsymbol{\Delta} \boldsymbol{\Gamma})/2.$$

Hence, we have

$$|\boldsymbol{V}^\mathsf{T} \boldsymbol{U}^{-1} \boldsymbol{V} - \tau^2 \mathbf{1}^\mathsf{T} \boldsymbol{U}_0^{-1} \mathbf{1}/d| = o_{d,\mathbb{P}}(1).$$

Proposition 2 gives the expression

$$\mathbf{1}^\mathsf{T} \boldsymbol{U}_0^{-1} \mathbf{1}/d = \psi/(1 + \kappa\psi) + o_{d,\mathbb{P}}(1),$$

where

$$\kappa = d \cdot \lambda_2^2 [\mathrm{Tr}(\boldsymbol{\Gamma}^2)/2 + \mathrm{Tr}(\boldsymbol{\Delta} \boldsymbol{\Gamma})^2/4].$$

Hence we have

$$\boldsymbol{V}^\mathsf{T} \boldsymbol{U} \boldsymbol{V} = \tau^2 \psi/(1 + \kappa\psi) + o_{d,\mathbb{P}}(1).$$

This proves the theorem.

## C.2 Neural Tangent model: proof of Theorem 5

Recall the definition (note $R_{\mathsf{NT},N}(\mathbb{P})$ is a function of $\boldsymbol{W}$)

$$R_{\mathsf{NT},N}(\mathbb{P}) = \min_{\hat{f} \in \mathcal{F}_{\mathsf{NT},N}(\boldsymbol{W})} \mathbb{E}\big\{(y - \hat{f}(\boldsymbol{x}))^2\big\},$$

where

$$\mathcal{F}_{\mathsf{NT},N}(\boldsymbol{W}) = \Big\{f_N(\boldsymbol{x}) = c + \sum_{i=1}^{N} \sigma'(\langle \boldsymbol{w}_i, \boldsymbol{x}\rangle)\langle \boldsymbol{a}_i, \boldsymbol{x}\rangle : c \in \mathbb{R}, \boldsymbol{a}_i \in \mathbb{R}^d, i \in [N]\Big\}.$$

### C.2.1 A representation lemma

**Lemma 13.** *Assume conditions* **M1** *and* **M2** *hold. Consider the function*

$$\hat{f}(\boldsymbol{x}; \boldsymbol{\Gamma}, a, c) = a\langle \boldsymbol{\Gamma}, \boldsymbol{x}\boldsymbol{x}^{\mathsf{T}}\rangle + c. \tag{85}$$

*Define the risk function optimized over $a, c$ while $\boldsymbol{\Gamma}$ is fixed*

$$L(\boldsymbol{\Gamma}) = \inf_{a,c} \mathbb{E}_{\boldsymbol{x},y}[(y - \hat{f}(\boldsymbol{x}; \boldsymbol{\Gamma}, a, c))^2]. \tag{86}$$

*Then we have*

$$\sup_{\boldsymbol{\Gamma} \succeq 0} \Big| L(\boldsymbol{\Gamma}) - \frac{2}{2 + \langle \boldsymbol{\Gamma}, \boldsymbol{\Delta}\rangle^2 / \|\boldsymbol{\Sigma}^{1/2}\boldsymbol{\Gamma}\boldsymbol{\Sigma}^{1/2}\|_F^2}\Big| = o_d(1). \tag{87}$$

*Proof of Lemma 13.* Note we have

$$\begin{aligned}
L(\boldsymbol{\Gamma}, a, c) &\equiv \mathbb{E}_{\boldsymbol{x},y}[(y - \hat{f}(\boldsymbol{x}; \boldsymbol{\Gamma}, a, c))^2] \\
&= 1 + c^2 + 2ac\langle \boldsymbol{\Gamma}, \boldsymbol{\Sigma}\rangle + 2a\langle \boldsymbol{\Gamma}, \boldsymbol{\Delta}\rangle \\
&\quad + a^2[\langle \boldsymbol{\Gamma}, \boldsymbol{\Sigma}\rangle^2 + 2\mathrm{Tr}(\boldsymbol{\Sigma}\boldsymbol{\Gamma}\boldsymbol{\Sigma}\boldsymbol{\Gamma}) + \langle \boldsymbol{\Gamma}, \boldsymbol{\Delta}\rangle^2 + 2\mathrm{Tr}(\boldsymbol{\Delta}\boldsymbol{\Gamma}\boldsymbol{\Delta}\boldsymbol{\Gamma})].
\end{aligned}$$

Minimizing successively over $c$ and $a$, we get the following formula:

$$L(\boldsymbol{\Gamma}) \equiv \min_{c,a \in \mathbb{R}} L(\boldsymbol{\Gamma}, a, c) = \frac{2}{2 + \langle \boldsymbol{\Gamma}, \boldsymbol{\Delta}\rangle^2 / [\mathrm{Tr}(\boldsymbol{\Gamma}\boldsymbol{\Sigma}\boldsymbol{\Gamma}\boldsymbol{\Sigma}) + \mathrm{Tr}(\boldsymbol{\Gamma}\boldsymbol{\Delta}\boldsymbol{\Gamma}\boldsymbol{\Delta})]}.$$

By Assumptions **M1** and **M2**, we have $\boldsymbol{\Sigma} \succeq c\mathbf{I}_d$ and $\|\boldsymbol{\Delta}\|_{\mathrm{op}} \leq C/\sqrt{d}$ for some constants $c$ and $C$. We get

$$\frac{\mathrm{Tr}(\boldsymbol{\Gamma}\boldsymbol{\Delta}\boldsymbol{\Gamma}\boldsymbol{\Delta})}{\mathrm{Tr}(\boldsymbol{\Gamma}\boldsymbol{\Sigma}\boldsymbol{\Gamma}\boldsymbol{\Sigma})} \leq \frac{C^2}{dc^2}.$$

We deduce that

$$\sup_{\boldsymbol{\Gamma} \succeq 0} \Big| L(\boldsymbol{\Gamma}) - \frac{2}{2 + \langle \boldsymbol{\Gamma}, \boldsymbol{\Delta}\rangle^2 / \|\boldsymbol{\Sigma}^{1/2}\boldsymbol{\Gamma}\boldsymbol{\Sigma}^{1/2}\|_F^2}\Big| \leq \Big| \frac{1}{1 + C^2/(dc^2)} - 1\Big| = o_d(1).$$

$\square$

### C.2.2 Proof of Theorem 5

We consider the re-scaled matrices $\tilde{\boldsymbol{\Gamma}} = \boldsymbol{\Sigma}^{1/2}\boldsymbol{\Gamma}\boldsymbol{\Sigma}^{1/2}$ and $\tilde{\boldsymbol{\Delta}} = \boldsymbol{\Sigma}^{-1/2}\boldsymbol{\Delta}\boldsymbol{\Sigma}^{-1/2}$. We consider the NT model with a squared non-linearity:

$$\hat{f}(\boldsymbol{W}, \boldsymbol{A}) = 2\sum_{i=1}^{N} \langle \boldsymbol{w}_i, \boldsymbol{x}\rangle\langle \boldsymbol{a}_i, \boldsymbol{x}\rangle + c = 2\langle \boldsymbol{W}\boldsymbol{A}^{\mathsf{T}}, \boldsymbol{x}\boldsymbol{x}^{\mathsf{T}}\rangle + c.$$

with $\boldsymbol{W} = [\boldsymbol{w}_1, \ldots, \boldsymbol{w}_N] \in \mathbb{R}^{d \times N}$ and $\boldsymbol{A} = [\boldsymbol{a}_1, \ldots, \boldsymbol{a}_N] \in \mathbb{R}^{d \times N}$. For $\boldsymbol{w}_i \sim \mathsf{N}(\boldsymbol{0}, \boldsymbol{\Sigma})$, we have with probability one $\mathrm{rank}(\boldsymbol{W}) = \min(d, N) \equiv r$. We consider $\boldsymbol{W} = \boldsymbol{P}_1 \boldsymbol{S} \boldsymbol{V}^\mathsf{T}$ the SVD decomposition of $\boldsymbol{W}$, with $\boldsymbol{P}_1 \in \mathbb{R}^{d \times r}$, $\boldsymbol{S} \in \mathbb{R}^{r \times r}$ and $\boldsymbol{V} \in \mathbb{R}^{N \times r}$. Define $\boldsymbol{G} = \boldsymbol{S} \boldsymbol{V}^\mathsf{T} \boldsymbol{A} \in \mathbb{R}^{r \times d}$, we obtain almost surely that the minimum over $\boldsymbol{A}$ is the same as the minimum over $\boldsymbol{G}$. From Lemma 13, we deduce that almost surely

$$R_{\mathsf{NT}, N}(\mathbb{P}_{\boldsymbol{\Sigma}, \boldsymbol{\Delta}}) = \min_{\boldsymbol{G} \in \mathbb{R}^{d \times d}} \left\{ \frac{2}{2 + \mathrm{Tr}[(\boldsymbol{P}_1 \boldsymbol{G} + \boldsymbol{G}^\mathsf{T} \boldsymbol{P}_1^\mathsf{T}) \boldsymbol{\Delta}]^2 / \|\boldsymbol{P}_1 \boldsymbol{G} + \boldsymbol{G}^\mathsf{T} \boldsymbol{P}_1^\mathsf{T}\|_F^2} \right\} + o_d(1) \tag{88}$$

**Case $N/d \to \rho \geq 1$.** In the case $N \geq d$, we can take $\boldsymbol{G} = \boldsymbol{P}_1^\mathsf{T} \tilde{\boldsymbol{G}}/2$ and we get almost surely over $\boldsymbol{W} \in \mathbb{R}^{d \times N}$

$$R_{\mathsf{NT}, N}(\mathbb{P}_{\boldsymbol{\Sigma}, \boldsymbol{\Delta}}) = \min_{\boldsymbol{G} \in \mathbb{R}^{d \times d}} \left\{ \frac{2}{2 + \langle \boldsymbol{G}, \boldsymbol{\Delta} \rangle^2 / \|\boldsymbol{G}\|_F^2} \right\} + o_d(1) = \frac{2}{2 + \|\boldsymbol{\Delta}\|_F^2} + o_d(1),$$

where the minimizer $\boldsymbol{G} = \boldsymbol{\Delta}$ is obtained by Cauchy-Schwarz inequality.

**Case $N/d \to \rho < 1$.** Consider now the case when $N < d$. From (88), the optimal $\boldsymbol{G}$ is the one maximizing

$$\max_{\boldsymbol{G} \in \mathbb{R}^{N \times d}} \frac{\mathrm{Tr}[(\boldsymbol{P}_1 \boldsymbol{G} + \boldsymbol{G}^\mathsf{T} \boldsymbol{P}_1^\mathsf{T}) \boldsymbol{\Delta}]^2}{\|\boldsymbol{P}_1 \boldsymbol{G} + \boldsymbol{G}^\mathsf{T} \boldsymbol{P}_1^\mathsf{T}\|_F^2},$$

which we rewrite as the following convex problem

$$\max_{\boldsymbol{G} \in \mathbb{R}^{N \times d}} \mathrm{Tr}[\boldsymbol{P}_1 \boldsymbol{G} \boldsymbol{\Delta}], \qquad \text{s.t.} \quad \|\boldsymbol{P}_1 \boldsymbol{G} + \boldsymbol{G}^\mathsf{T} \boldsymbol{P}_1^\mathsf{T}\|_F^2 \leq 1. \tag{89}$$

We define $\boldsymbol{P}_2 \in \mathbb{R}^{d \times (d-N)}$ the completion of $\boldsymbol{P}_1$ to a full basis $\boldsymbol{P} = [\boldsymbol{P}_1, \boldsymbol{P}_2] \in \mathbb{R}^{d \times d}$, and denote $\boldsymbol{G}_1 = \boldsymbol{G} \boldsymbol{P}_1 \in \mathbb{R}^{N \times N}$ and $\boldsymbol{G}_2 = \boldsymbol{G} \boldsymbol{P}_2 \in \mathbb{R}^{N \times (d-N)}$. We can form the Lagrangian of problem (89):

$$\mathcal{L}(\boldsymbol{G}, \lambda) = \mathrm{Tr}(\boldsymbol{P}_1 \boldsymbol{G} \boldsymbol{\Delta}) + \lambda(1 - \|\boldsymbol{P}_1 \boldsymbol{G} + \boldsymbol{G}^\mathsf{T} \boldsymbol{P}_1^\mathsf{T}\|_F^2).$$

The stationary condition implies:

$$\nabla_{\boldsymbol{G}} \mathcal{L}(\boldsymbol{G}, \lambda) = \boldsymbol{P}_1^\mathsf{T} \boldsymbol{\Delta} - 4\lambda(\boldsymbol{P}_1^\mathsf{T} \boldsymbol{G}^\mathsf{T} \boldsymbol{P}_1^\mathsf{T} + \boldsymbol{P}_1^\mathsf{T} \boldsymbol{P}_1 \boldsymbol{G}) = 0,$$

which yields, using $\boldsymbol{P}_1^\mathsf{T} \boldsymbol{P}_1 = \mathbf{I}_N$,

$$\boldsymbol{\Delta}_{12} = 4\lambda \boldsymbol{G}_2, \qquad \boldsymbol{\Delta}_{11} = 4\lambda(\boldsymbol{G}_1 + \boldsymbol{G}_1^\mathsf{T}), \tag{90}$$

where $\boldsymbol{\Delta}_{ij} = \boldsymbol{P}_i^\mathsf{T} \boldsymbol{\Delta} \boldsymbol{P}_j$ for $i, j = 1, 2$. The constraint reads in the $\boldsymbol{P}$ basis

$$\|\boldsymbol{P}_1 \boldsymbol{G} + \boldsymbol{G}^\mathsf{T} \boldsymbol{P}_1^\mathsf{T}\|_F^2 = \|\boldsymbol{G}_1 + \boldsymbol{G}_1^\mathsf{T}\|_F^2 + 2\|\boldsymbol{G}_2\|_F^2 = 1. \tag{91}$$

Substituting (90) in (91) yields:

$$4\lambda = \sqrt{\|\boldsymbol{\Delta}_{11}\|_F^2 + 2\|\boldsymbol{\Delta}_{12}\|_F^2}. \tag{92}$$

Considering the (unique) symmetric optimizer $\boldsymbol{G}_1$ and substituting (92) in (90), we get the minimizer

$$\begin{aligned}
\boldsymbol{G}_1^* &= \frac{1}{8\lambda} \boldsymbol{\Delta}_{11} = \frac{1}{2\sqrt{\|\boldsymbol{\Delta}_{11}\|_F^2 + 2\|\boldsymbol{\Delta}_{12}\|_F^2}} \boldsymbol{\Delta}_{11}, \\
\boldsymbol{G}_2^* &= \frac{1}{4\lambda} \boldsymbol{\Delta}_{12} = \frac{1}{\sqrt{\|\boldsymbol{\Delta}_{11}\|_F^2 + 2\|\boldsymbol{\Delta}_{12}\|_F^2}} \boldsymbol{\Delta}_{12}.
\end{aligned} \tag{93}$$

Let's consider the objective function:

$$\begin{aligned}
\mathrm{Tr}(\boldsymbol{P}_1 \boldsymbol{G}^* \boldsymbol{\Delta}) &= \mathrm{Tr}(\boldsymbol{G}_1^* \boldsymbol{\Delta}_{11} + \boldsymbol{G}_2^* \boldsymbol{\Delta}_{21}) \\
&= \frac{1}{2\sqrt{\|\boldsymbol{\Delta}_{11}\|_F^2 + 2\|\boldsymbol{\Delta}_{12}\|_F^2}} \mathrm{Tr}(\boldsymbol{\Delta}_{11}^2 + 2\boldsymbol{\Delta}_{12} \boldsymbol{\Delta}_{21}) \\
&= \frac{1}{2} \sqrt{\|\boldsymbol{\Delta}_{11}\|_F^2 + 2\|\boldsymbol{\Delta}_{12}\|_F^2} \\
&= \frac{1}{2} \sqrt{\|\boldsymbol{\Delta}\|_F^2 - \|\boldsymbol{\Delta}_{22}\|_F^2}. 
\end{aligned} \tag{94}$$

Substituting (94) in (88), we then obtain

$$R_{\mathsf{NT},N}(\mathbb{P}_{\boldsymbol{\Sigma},\boldsymbol{\Delta}}) = \frac{2}{2 + \|\boldsymbol{\Delta}\|_F^2 - \|\boldsymbol{\Delta}_{22}\|_F^2} + o_d(1), \tag{95}$$

where $\boldsymbol{\Delta}_{22} = \boldsymbol{P}_{\boldsymbol{W}^\perp} \boldsymbol{\Delta} \boldsymbol{P}_{\boldsymbol{W}^\perp}$ with $\boldsymbol{P}_{\boldsymbol{W}^\perp} = \mathbf{I}_d - \boldsymbol{W}(\boldsymbol{W}^\mathsf{T}\boldsymbol{W})^{-1}\boldsymbol{W}^\mathsf{T}$ is the random projection along the orthogonal subspace to the columns of $\boldsymbol{W}$. From Theorem 2, we know that

$$\mathbb{E}[\|\boldsymbol{\Delta}_{22}\|_F^2] = \|\boldsymbol{\Delta}\|_F^2 \Big[ (1-\rho)_+^2 \Big( 1 - \frac{\mathrm{Tr}(\boldsymbol{\Delta})^2}{d\|\boldsymbol{\Delta}\|_F^2} \Big) + (1-\rho)_+ \frac{\mathrm{Tr}(\boldsymbol{\Delta})^2}{d\|\boldsymbol{\Delta}\|_F^2} + o_d(1) \Big]. \tag{96}$$

Let $\mathbb{W}_d^N$ be the Stiefel manifold, i.e. the collection of all the sets of $N$ orthonormal vectors in $\mathbb{R}^d$ endowed with the Frobenius distance. In matrix representation, we have

$$\mathbb{W}_d^N = \{ \boldsymbol{P} \in \mathbb{R}^{d \times N} : \boldsymbol{P}^\mathsf{T}\boldsymbol{P} = \mathbf{I}_N \}.$$

By Theorem 2.4 in [Led01], the volume measure on $\mathbb{W}_d^N$ has normal concentration. In particular, denote by $F : \mathbb{W}_d^N \mapsto \mathbb{R}$, the function $F(\boldsymbol{P}) = \|\boldsymbol{P}^\mathsf{T}\boldsymbol{\Delta}\boldsymbol{P}\|_F^2$. We upper bound the gradient of $F$:

$$\|\nabla F(\boldsymbol{P})\|_F = 4\|\boldsymbol{\Delta}\boldsymbol{P}\boldsymbol{P}^\mathsf{T}\boldsymbol{\Delta}\boldsymbol{P}\|_F \le 4\|\boldsymbol{\Delta}\boldsymbol{P}\boldsymbol{P}^\mathsf{T}\|_{\mathrm{op}}\|\boldsymbol{\Delta}\boldsymbol{P}\|_F \le \|\boldsymbol{\Delta}\|_{\mathrm{op}}\|\boldsymbol{\Delta}\|_F \le C,$$

by assumption **M2** on $\boldsymbol{\Delta}$. We deduce that there exists a constant $c$ (that depends on $\rho$ and $C$) such that:

$$\mathbb{P}(|F(\boldsymbol{P}) - \mathbb{E}[F(\boldsymbol{P})]| > t) \le e^{-cdt^2}.$$

Therefore, we have

$$\mathbb{P}(|\|\boldsymbol{\Delta}_{22}\|_F^2 - E[\|\boldsymbol{\Delta}_{22}\|_F^2]| > t) \le e^{-cdt^2}. \tag{97}$$

Using (97) and (95), we deduce the final high probability formula for the risk of the $\mathsf{NT}$ model:

$$R_{\mathsf{NT},N}(\mathbb{P}_{\boldsymbol{\Sigma},\boldsymbol{\Delta}}) = \frac{2}{2 + \|\boldsymbol{\Delta}\|_F^2 - \mathbb{E}[\|\boldsymbol{\Delta}_{22}\|_F^2]} + o_{d,\mathbb{P}}(1).$$

Substituting $\mathbb{E}[\|\boldsymbol{\Delta}_{22}\|_F^2]$ by its expression (96) concludes the proof.

## C.3  Neural Network model: proof of Theorem 6

Recall the definition

$$R_{\mathsf{NN},N}(\mathbb{P}) = \min_{\hat{f} \in \mathcal{F}_{\mathsf{NN},N}(\boldsymbol{W})} \mathbb{E}\big\{(y - \hat{f}(\boldsymbol{x}))^2\big\},$$

where we consider the function class of two-layers neural networks (with $N$ neurons) with quadratic activation function and general offset and coefficients

$$\mathcal{F}_{\mathsf{NN},N}(\boldsymbol{W}) = \Big\{ f_N(\boldsymbol{x}) = c + \sum_{i=1}^{N} a_i(\langle \boldsymbol{w}_i, \boldsymbol{x} \rangle)^2 : c, a_i \in \mathbb{R}, i \in [N] \Big\}.$$

We define the risk function for a given set of parameters as

$$L(\boldsymbol{W}, \boldsymbol{a}, c) = \mathbb{E}_{\boldsymbol{x},y}[(y - \hat{f}(\boldsymbol{x}; \boldsymbol{W}, \boldsymbol{a}, c))^2].$$

The risk is optimized over $(a_i, \boldsymbol{w}_i)_{i \le N}$ and $c$.

*Proof of Theorem 6.* Without loss of generality, we assume $\boldsymbol{\Sigma} = \mathbf{I}_d$ (it suffices to consider the re-scaled matrices $\tilde{\boldsymbol{\Gamma}} = \boldsymbol{\Sigma}^{1/2}\boldsymbol{\Gamma}\boldsymbol{\Sigma}^{1/2}$ and $\tilde{\boldsymbol{\Delta}} = \boldsymbol{\Sigma}^{-1/2}\boldsymbol{\Delta}\boldsymbol{\Sigma}^{-1/2}$). We rewrite the neural network function in a compact form:

$$\hat{f}(\boldsymbol{x}; \boldsymbol{W}, \boldsymbol{a}, c) = \sum_{i=1}^{N} a_i \langle \boldsymbol{w}_i, \boldsymbol{x} \rangle^2 + c = \langle \boldsymbol{W}\boldsymbol{A}\boldsymbol{W}^\mathsf{T}, \boldsymbol{x}\boldsymbol{x}^\mathsf{T} \rangle + c,$$

where $\boldsymbol{A} = \mathrm{diag}(\boldsymbol{a})$. Define $\boldsymbol{\Gamma} = \boldsymbol{W}\boldsymbol{A}\boldsymbol{W}^\top$ and using Eq. (87) in Lemma 13, the minimizer $\boldsymbol{\Gamma}^*$ is the solution of

$$\max_{\boldsymbol{\Gamma} \in \mathcal{S}(\mathbb{R}^{d \times d})} \frac{\langle \boldsymbol{\Gamma}, \boldsymbol{\Delta} \rangle^2}{\|\boldsymbol{\Gamma}\|_F^2}, \qquad \text{s.t.} \quad \mathrm{rank}(\boldsymbol{\Gamma}) \le \min(N, d) \equiv r.$$

where $\mathcal{S}(\mathbb{R}^{d \times d})$ is the set of symmetric matrices in $\mathbb{R}^{d \times d}$.

Let us denote the eigendecomposition of $\boldsymbol{\Gamma}$ by $\boldsymbol{\Gamma} = \boldsymbol{U}\boldsymbol{S}\boldsymbol{U}^\top$ with $\boldsymbol{U} \in \mathbb{R}^{d \times r}$ and $\boldsymbol{S} = \mathrm{diag}(\boldsymbol{s}) \in \mathbb{R}^{r \times r}$. We have by Cauchy-Schwartz inequality

$$\frac{\langle \boldsymbol{\Gamma}, \boldsymbol{\Delta} \rangle^2}{\|\boldsymbol{\Gamma}\|_F^2} = \frac{\mathrm{Tr}(\boldsymbol{S}\boldsymbol{U}^\top\boldsymbol{\Delta}\boldsymbol{U})^2}{\|\boldsymbol{S}\|_F^2} \le \|\mathrm{diag}(\boldsymbol{U}^\top\boldsymbol{\Delta}\boldsymbol{U})\|_2^2,$$

with equality if and only if $\boldsymbol{S}_* = \mathrm{ddiag}(\boldsymbol{U}^\top\boldsymbol{\Delta}\boldsymbol{U})$ where $\mathrm{ddiag}(\boldsymbol{U}^\top\boldsymbol{\Delta}\boldsymbol{U})$ is the vector of the diagonal elements of $\boldsymbol{U}^\top\boldsymbol{\Delta}\boldsymbol{U}$. Denoting $\mathcal{D}(\mathbb{R}^{d \times d})$ the set of diagonal matrices in $\mathbb{R}^{d \times d}$, we get

$$\max_{\boldsymbol{S} \in \mathcal{D}(\mathbb{R}^{d \times d})} \frac{\langle \boldsymbol{U}\boldsymbol{S}\boldsymbol{U}^\top, \boldsymbol{\Delta} \rangle^2}{\|\boldsymbol{U}\boldsymbol{S}\boldsymbol{U}^\top\|_F^2} = \frac{\langle \boldsymbol{S}_*, \boldsymbol{U}^\top\boldsymbol{\Delta}\boldsymbol{U} \rangle^2}{\|\boldsymbol{S}_*\|_F^2} = \frac{\|\boldsymbol{S}_*\|_F^4}{\|\boldsymbol{S}_*\|_F^2} = \|\boldsymbol{S}_*\|_F^2.$$

Hence, the problem reduces to finding $\boldsymbol{U} \in \mathbb{R}^{d \times r}$ with orthonormal columns which maximizes $\|\mathrm{ddiag}(\boldsymbol{U}^\top\boldsymbol{\Delta}\boldsymbol{U})\|_F^2$. The maximizer is easily found as the eigendirections corresponding to the $r$ largest singular values. We conclude that at the optimum

$$\frac{\langle \boldsymbol{\Gamma}_*, \boldsymbol{\Delta} \rangle^2}{\|\boldsymbol{\Gamma}_*\|_F^2} = \sum_{i=1}^r \lambda_i^2,$$

where the $\lambda_i$'s are the singular values of $\boldsymbol{\Delta}$ in descending order. Plugging this expression in Eq. (87) concludes the proof. $\qquad\square$

# D    Additional Experiments

For the sake of theoretical analysis, we focused on the case of quadratic activations for NT and NN in the main text. However, the phenomena we presented persist (qualitatively) even when other activation functions are used. For example, figures 3 and 4 examine the performance of our models when ReLU non-linearity is used. These experiments suggest that the when $d$ is larger than $N$ there is a significant performance gap between NN and NT. Moreover, similar to what was presented in the paper, we observe that the gap between $\mathsf{RF}(I)$ and NN does not vanish unless $\frac{N}{d} \to \infty$.

Figure 3: Left frame: Prediction (test) error of a two-layer neural networks in fitting a quadratic function in $d = 450$ dimensions, as a function of the number of neurons $N$. We consider the large sample (population) limit $n \to \infty$ and compare three training regimes: random features (RF), neural tangent (NT), and fully trained neural networks (NN). All models use **ReLU** activations. Right frame: Evolution of the risk for NT and NN with the number of samples.

Figure 4: Left frame: Prediction (test) error of a two-layer neural networks in fitting a mixture of Gaussians in $d = 450$ dimensions, as a function of the number of neurons $N$. We consider the large sample (population) limit $n \to \infty$ and compare three training regimes: random features (RF), neural tangent (NT), and fully trained neural networks (NN). All models use **ReLU** activations. Right frame: Evolution of the risk for NT and NN with the number of samples.