[Reviews · NeurIPS 2019]

Reviewer 1



See above. This work seems to be original and well written, and offers an interesting theoretical and experimental exploration of a simple version of several methods often proposed for computation-constrained applications. Edit: The authors have provided avenues for extending the results in the rebuttal. More importantly, they have provided evidence that the trends they observe hold for non-quadratic activations. I have thus incremented my score.

Reviewer 2



The analysis shows that the error of RF is always bounded away from zero unless the number of neural N goes to infinity. Both NT and NN achieve zero error if N is greater than or equal to the dimension d. It also shows that NN always achieves smaller errors than NT, because NN learns to fit the most significant direction, while NT can only fit the sub-space that is spanned by random directions. The results of the paper is quite intuitive, but non-trivial in my perspective. It provides a clear evidence that even for simple target functions, the neural network can hold advantage over random features models. This fact is often easy to be ignored, because for many simple tasks, random features work as well as the neural networks. I think the paper can become a reference point for future work when people want to talk about MLP versus random features. The comparison between the RF and the NT is not very meaningful. It is true that NT can achieve a zero error with a finite number of neurons while RF cannot, but that only holds for specific target functions (quadratic and mixture of Gaussians), not to mention that NT has much more parameters to learn than RF. That said, the comparison between RF/NT and NN is the main contribution of this paper.

Reviewer 3



In this article, the authors analyzed the performance of a single-hidden-layer neural network model under the random feature (RF) regime, the neural tangent (NT) regime, as well as the fully trained neural network (NN) regime. By considering the tasks of 1) learning a quadratic function of d-dimensional Gaussian data, and 2) classifying a two-class d-dimensional Gaussian mixture, the authors showed that, in the high dimensional regime where the number of neurons N and the data dimension p are both large and comparable, one has NN > NT > RF in the sense of prediction performance, in the infinite data limit. In this vein, this article improves/generalizes the analyses in [25] by covering the neural tangent model, which is a more involved model that is of more practical interest. This article provides solid analyses on an interesting problem and is already quite polished. I strongly recommend it for publication. Below are a few minor comments that the authors might consider addressing before publication. **After rebuttal**: This is a solid work on an interesting topic, I vote for accepting.

[Author Response · NeurIPS 2019]

## Response to Reviewer #1

The goal of this paper is to determine whether neural networks (NN) are equivalent to kernel methods or not. While at first sight one would guess that neural networks are more powerful than 'simple' kernel methods, the recently developed neural tangent (NT) kernel theory (with a dozen papers published by several groups) argue that SGD-trained networks are well approximated by the NT kernel, for very large networks.

In this paper, we argue in the opposite direction, namely NN is superior to NT. Our argument consists in rigorously proving that, in a concrete example, a significant gap exists between NT and NN. In view of this, *a simpler example results in a stronger conclusion.* This being said, a series of extensions are available:

**1.** RF model. The analysis of the random features model can be generalized to arbitrary target functions $f_*$, and activation function $\sigma$ (under mild technical conditions). Also, the analysis can be generalized to the case of $x$ and $w$ distributed uniformly over the $d$-dimensional sphere.

All of these generalizations are accessible to the same proof techniques developed in our paper. We focused on the quadratic case uniquely to make the comparison more transparent.

**2.** NT model. This analysis can be generalized to arbitrary $f_*$, if $x$ and $w$ are distributed uniformly on the sphere.

**3.** NN model. Generalizing the analysis of the NN model would require proving global convergence for gradient descent beyond quadratic activations. This has been an open problem for a long time, and it probably requires additional assumptions. Before our paper, this problem was open for quadratic activations as well.

**4.** We have extensive experimental results comparing NN and NT for RELU and Tanh activations (see figure). Our experiments indicate that these nonlinearities behave in general as predicted by the theory developed for the square nonlinearity.

Figure 1: Left: Test error of NN, NT and RF in fitting a quadratic function in d = 450 dimensions. Here the experiments are performed using **ReLU** nonlinearity. Similar to the case of square nonlinearity studied in the paper, there is a significant gap between NT and NN. Right: Evolution of the risk for NT and NN with the number of samples.

## Response to Reviewer #2

We agree that a direct comparison of RF and NT is not meaningful. We find the difference in behavior quite interesting, but not indicating the superiority of one method over the other.

## Response to Reviewer #3

**1.** We believe that the case of Gaussians with unequal means can also be analyzed within the RF and NT model (under certain technical assumptions) although at the cost of non-negligible technical complications.

**2.** We believe that the analysis of RF and NT can be generalized to certain other models beyond Gaussian covariates $x$. For instance, the case of $x$ and $w$ uniform over the $d$-dimensional sphere can be treated using similar techniques. Also certain derivations can be extended to $x = \Sigma^{1/2} z$ with $z$ having i.i.d. (non-Gaussian) components.

**3.** Our proof uses the gradient flow dynamics for studying the convergence of the NN. In the small learning rate regime, both SGD and mini-batch SGD dynamics converge to the gradient flow dynamics. We use the mini-batch SGD setting because (1) the convergence happens with larger learning rates and (2) experiments with minibatch SGD can be done in a computationally efficient, GPU-friendly manner.

**4.** We have performed an extensive set of experiments with different types of $\Delta$, $\Sigma$ and activation functions. Due to space constraints, we did not include these in the initial submission. We will definitely add these to the camera-ready version / appendix.

[Meta-Review · NeurIPS 2019]

The reviewers are positive about the paper and think that the feedback addressed their questions satisfactorily. The authors should incorporate that in the final submission.